# Antigenic drift and subtype interference shape A(H3N2) epidemic dynamics in the United States

Amanda C Perofsky[1,2]*, John Huddleston[3], Chelsea L Hansen[1,2], John R Barnes[4], Thomas Rowe[4], Xiyan Xu[4], Rebecca Kondor[4], David E Wentworth[4], Nicola Lewis[5], Lynne Whittaker[5], Burcu Ermetal[5], Ruth Harvey[5], Monica Galiano[5], Rodney Stuart Daniels[5], John W McCauley[5], Seiichiro Fujisaki[6], Kazuya Nakamura[6], Noriko Kishida[6], Shinji Watanabe[6], Hideki Hasegawa[6], Sheena G Sullivan[7], Ian G Barr[7], Kanta Subbarao[7], Florian Krammer[8,9], Trevor Bedford[2,3,10,11], Cécile Viboud[1]

[1]Fogarty International Center, National Institutes of Health, Bethesda, United States; [2]Brotman Baty Institute for Precision Medicine, University of Washington, Seattle, United States; [3]Vaccine and Infectious Disease Division, Fred Hutchinson Cancer Center, Seattle, United States; [4]Virology Surveillance and Diagnosis Branch, Influenza Division, National Center for Immunization and Respiratory Diseases (NCIRD), Centers for Disease Control and Prevention (CDC), Atlanta, United States; [5]WHO Collaborating Centre for Reference and Research on Influenza, Crick Worldwide Influenza Centre, The Francis Crick Institute, London, United Kingdom; [6]Influenza Virus Research Center, National Institute of Infectious Diseases, Tokyo, Japan; [7]WHO Collaborating Centre for Reference and Research on Influenza, The Peter Doherty Institute for Infection and Immunity, Department of Microbiology and Immunology, The University of Melbourne, The Peter Doherty Institute for Infection and Immunity, Melbourne, Australia; [8]Center for Vaccine Research and Pandemic Preparedness (C-VaRPP), Icahn School of Medicine at Mount Sinai, New York, United States; [9]Department of Pathology, Molecular and Cell-Based Medicine, Icahn School of Medicine at Mount Sinai, New York, United States; [10]Department of Genome Sciences, University of Washington, Seattle, United States; [11]Howard Hughes Medical Institute, Seattle, United States

*For correspondence: amanda.perofsky@nih.gov

**Abstract** Influenza viruses continually evolve new antigenic variants, through mutations in epitopes of their major surface proteins, hemagglutinin (HA) and neuraminidase (NA). Antigenic drift potentiates the reinfection of previously infected individuals, but the contribution of this process to variability in annual epidemics is not well understood. Here, we link influenza A(H3N2) virus evolution to regional epidemic dynamics in the United States during 1997—2019. We integrate phenotypic measures of HA antigenic drift and sequence-based measures of HA and NA fitness to infer antigenic and genetic distances between viruses circulating in successive seasons. We estimate the magnitude, severity, timing, transmission rate, age-specific patterns, and subtype dominance of each regional outbreak and find that genetic distance based on broad sets of epitope sites is the strongest evolutionary predictor of A(H3N2) virus epidemiology. Increased HA and NA epitope distance between seasons correlates with larger, more intense epidemics, higher transmission, greater A(H3N2) subtype dominance, and a greater proportion of cases in adults relative to children, consistent with increased population susceptibility. Based on random forest models, A(H1N1) incidence impacts A(H3N2) epidemics to a greater extent than viral evolution, suggesting that subtype

interference is a major driver of influenza A virus infection ynamics, presumably via heterosubtypic cross-immunity.

## eLife assessment

This paper explores the relationships among evolutionary and epidemiological quantities in influenza, and presents **fundamental** findings that substantially advance our understanding of the drivers of influenza epidemics. The authors use a rich set of data sources to gather and analyze **compelling** evidence on the roles of genetic distance, other influenza dynamics and epidemiological indicators in predicting influenza epidemics. The central findings highlight the significant influence of genetic distance on A(H3N2) virus epidemiology and emphasize the role of A(H1N1) virus incidence in shaping A(H3N2) epidemics, suggesting subtype interference as a key factor. This paper also makes relevant data available to the research community.

## Introduction

Influenza viruses continually accumulate genetic changes in epitopes of two major surface proteins, hemagglutinin (HA) and neuraminidase (NA), in a process known as 'antigenic drift'. Although individual hosts develop long-lasting immunity to specific influenza virus strains after infection, antigenic drift helps the virus to escape immune recognition, leaving previously exposed hosts susceptible to reinfection and necessitating regular updates to the antigens included in the influenza vaccine (*Gerdil, 2003*). While antigenic drift aids immune escape, prospective cohort studies and modeling of surveillance data also indicate that reinfection by antigenically homologous viruses occurs on average every 1–4 years, due to the waning of protection over time (*He et al., 2015*; *Wraith et al., 2022*).

Among the influenza virus types that routinely co-circulate in humans (A and B), type A viruses, particularly subtype A(H3N2), experience the fastest rates of antigenic evolution and cause the most substantial morbidity and mortality (*Bedford et al., 2015*; *Bedford et al., 2014*; *Ferguson et al., 2005*; *Hay et al., 2001*). Seasonal influenza A viruses (IAV) cause annual winter epidemics in temperate zones of the Northern and Southern Hemispheres and circulate year-round in tropical regions (*Simonsen, 1999*). Influenza A epidemic burden fluctuates substantially from year to year (*Viboud et al., 2004*), and there is much scientific interest in disentangling the relative roles of viral evolution, prior immunity, human behavior, and climatic factors in driving this seasonal variability. Climatic factors, such as humidity and temperature, have been implicated in the seasonality and timing of winter outbreaks in temperate regions (*Chattopadhyay et al., 2018*; *Kramer and Shaman, 2019*; *Lee et al., 2018*; *Shaman and Kohn, 2009*; *Shaman et al., 2010*), while contact and mobility patterns contribute to the seeding of new outbreaks and geographic spread (*Bedford et al., 2010*; *Bedford et al., 2015*; *Charu et al., 2017*; *Chattopadhyay et al., 2018*; *Geoghegan et al., 2018*; *Pei et al., 2018*; *Viboud et al., 2006*). A principal requirement for the recurrence of epidemics is a sufficient and continuous source of susceptible individuals, which is determined by the degree of cross-immunity between the surface antigens of currently circulating viruses and functional antibodies elicited by prior infection or vaccination in a population.

Because mutations to the HA1 region of the HA protein are considered to drive the majority of antigenic drift (*Nelson and Holmes, 2007*; *Wiley et al., 1981*), influenza virus genetic and antigenic surveillance have focused primarily on HA, and official influenza vaccine formulations prescribe the amount of HA (*Fiore et al., 2009*). Yet, evidence for the effect of HA drift on influenza epidemic dynamics remains conflicting. Theoretical and empirical studies have shown that HA drift between currently circulating viruses and the previous season's viruses is expected to cause earlier, larger, more severe, or more synchronized epidemics; however, the majority of these studies were limited to the pre 2009 influenza pandemic period (*Bedford et al., 2014*; *Boni et al., 2004*; *Geoghegan et al., 2018*; *Greene et al., 2006*; *Koelle et al., 2006*; *Koelle et al., 2009*; *Wolf et al., 2010*; *Wu et al., 2010*). Information on HA evolution has been shown to improve forecasts of seasonal influenza dynamics in Israel (*Axelsen et al., 2014*) and the United States (*Du et al., 2017*), but recent research has also found that HA evolution is not predictive of epidemic size in Australia (*Lam et al., 2020*) or epidemic timing in the United States (*Charu et al., 2017*). A caveat is that many of these studies used binary indicators to study seasonal antigenic change, defined as seasons in which circulating viruses

**eLife digest** Seasonal influenza (flu) viruses cause outbreaks every winter. People infected with influenza typically develop mild respiratory symptoms. But flu infections can cause serious illness in young children, older adults and people with chronic medical conditions. Infected or vaccinated individuals develop some immunity, but the viruses evolve quickly to evade these defenses in a process called antigenic drift. As the viruses change, they can re-infect previously immune people. Scientists update the flu vaccine yearly to keep up with this antigenic drift.

The immune system fights flu infections by recognizing two proteins, known as antigens, on the virus's surface, called hemagglutinin (HA) and neuraminidase (NA). However, mutations in the genes encoding these proteins can make them unrecognizable, letting the virus slip past the immune system. Scientists would like to know how these changes affect the size, severity and timing of annual influenza outbreaks.

Perofsky et al. show that tracking genetic changes in HA and NA may help improve flu season predictions. The experiments compared the severity of 22 flu seasons caused by the A(H3N2) subtype in the United States with how much HA and NA had evolved since the previous year. The A(H3N2) subtype experiences the fastest rates of antigenic drift and causes more cases and deaths than other seasonal flu viruses. Genetic changes in HA and NA were a better predictor of A(H3N2) outbreak severity than the blood tests for protective antibodies that epidemiologists traditionally use to track flu evolution. However, the prevalence of another subtype of influenza A circulating in the population, called A(H1N1), was an even better predictor of how severe A(H3N2) outbreaks would be.

Perofsky et al. are the first to show that genetic changes in NA contribute to the severity of flu seasons. Previous studies suggested a link between genetic changes in HA and flu season severity, and flu vaccines include the HA protein to help the body recognize new influenza strains. The results suggest that adding the NA protein to flu vaccines may improve their effectiveness. In the future, flu forecasters may want to analyze genetic changes in both NA and HA to make their outbreak predictions. Tracking how much of the A(H1N1) subtype is circulating may also be useful for predicting the severity of A(H3N2) outbreaks.

were antigenically distinct from the vaccine reference strain (*Charu et al., 2017*; *Geoghegan et al., 2018*; *Greene et al., 2006*; *Lam et al., 2020*; *Smith et al., 2004*). This may obscure epidemiologically relevant patterns, as positive selection in HA and NA is both episodic and continuous (*Bedford et al., 2011*; *Bedford et al., 2014*; *Bhatt et al., 2011*; *Huddleston et al., 2020*; *Shih et al., 2007*; *Smith et al., 2004*; *Suzuki, 2008*). Past research has also typically focused on serological and sequence-based measures of viral evolution in isolation, and the relative importance of these two approaches in predicting epidemic dynamics has not been systematically assessed. Further, to the best of our knowledge, the epidemiologic impact of NA evolution has not been explored.

There has been recent recognition of NA's role in virus inhibiting antibodies and its potential as a vaccine target (*Chen et al., 2018*; *Eichelberger et al., 2018*; *Wohlbold et al., 2015*). Although antibodies against NA do not prevent influenza infection, NA immunity attenuates the severity of infection by limiting viral replication (*Brett and Johansson, 2005*; *Couch et al., 1974*; *Johansson et al., 1993*; *Kilbourne, 1976*; *Murphy et al., 1972*; *Schulman et al., 1968*), and NA-specific antibody titers are an independent correlate of protection in both field studies and human challenge trials (*Couch et al., 2013*; *Memoli et al., 2016*; *Monto et al., 2015*). Lastly, the phenomenon of interference between influenza A subtypes, modulated by immunity to conserved T-cell epitopes (*Grebe et al., 2008*; *Sridhar et al., 2013*; *Ulmer et al., 1998*), has long been debated (*Epstein, 2006*; *Sonoguchi et al., 1985*). Interference effects are most pronounced during pandemic seasons, leading to troughs or even replacement of the resident subtype in some pandemics (*Ferguson et al., 2003*), but the contribution of heterosubtypic interference to annual dynamics is unclear (*Cowling et al., 2014*; *Gatti et al., 2022*; *Goldstein et al., 2011*; *He et al., 2015*; *Steinhoff et al., 1993*).

Here, we link A(H3N2) virus evolutionary dynamics to epidemiologic surveillance data in the United States over the course of 22 influenza seasons prior to the coronavirus disease 2019 (COVID-19) pandemic, considering the full diversity of viruses circulating in this period. We analyze a variety of antigenic and genetic markers of HA and NA evolution against multiple indicators characterizing the

epidemiology and disease burden of annual outbreaks. Rather than characterize in situ evolution of A(H3N2) lineages circulating in the U.S., we study the epidemiological impacts of antigenic drift once A(H3N2) variants have arrived on U.S. soil and managed to establish and circulate at relatively high levels. We find a signature of both HA and NA antigenic drift in surveillance data, with a more pronounced relationship in epitope change rather than the serology-based indicator, along with a major effect of subtype interference. Our study has implications for surveillance of evolutionary indicators that are most relevant for population impact and for the prediction of influenza burden on inter-annual timeframes.

## Methods

Our study focuses on the impact of A(H3N2) virus evolution on seasonal epidemics from seasons 1997–1998 to 2018–2019 in the U.S.; whenever possible, we make use of regionally disaggregated indicators and analyses. We start by identifying multiple indicators of influenza evolution each season based on changes in HA and NA. Next, we compile influenza virus subtype-specific incidence time series for U.S. Department of Health and Human Service (HHS) regions and estimate multiple indicators characterizing influenza A(H3N2) epidemic dynamics each season, including epidemic burden, severity, type/subtype dominance, timing, and the age distribution of cases. We then assess univariate relationships between national indicators of evolution and regional epidemic characteristics. Lastly, we use multivariable regression models and random forest models to measure the relative importance of viral evolution, heterosubtypic interference, and prior immunity in predicting regional A(H3N2) epidemic dynamics.

### Influenza epidemic timing and burden

Epidemiological data processing and analysis were performed using R version 4.3 (*R Development Core Team, 2023*).

#### Influenza-like illness and virological surveillance data

We obtained weekly epidemiological and virological data for influenza seasons 1997–1998 to 2018–2019, at the U.S. HHS region level. We defined influenza seasons as calendar week 40 in a given year to calendar week 20 in the following year, with the exception of the 2008–2009 season, which ended in 2009 week 16 due to the emergence of the A(H1N1)pdm09 virus (*Goldstein et al., 2011*).

We extracted syndromic surveillance data for the 10 HHS regions from the U.S. Outpatient Influenza-like Illness Surveillance Network (ILINet) (*Centers for Disease Control and Prevention, National Center for Immunization and Respiratory Diseases, 2023a*). ILINet consists of approximately 3200 sentinel outpatient healthcare providers throughout the U.S. that report the total number of consultations for any reason and the number of consultations for influenza-like illness (ILI) every week. ILI is defined as fever (temperature of 100 °F [37.8 °C] or greater) and a cough and/or a sore throat. ILI rates are based on the weekly proportion of outpatient consultations for influenza-like illness and are available weighted or unweighted by regional population size. The number of ILI encounters by age group are also provided (0–4, 5–24, 25–64, and ≥65), but these data are not weighted by total encounters or population size.

We obtained data on weekly influenza virus type and subtype circulation from the U.S. CDC's WHO Collaborating Center for Surveillance, Epidemiology and Control of Influenza (*World Health Organization, 2023*). Approximately 100 public health laboratories and 300 clinical laboratories located throughout the U.S. report influenza test results to the U.S. CDC, through either the U.S. WHO Collaborating Laboratories Systems or the National Respiratory and Enteric Virus Surveillance System (NREVSS). Clinical laboratories test respiratory specimens for diagnostic purposes whereas public health laboratories primarily test specimens to characterize influenza virus type, subtype, and lineage circulation. Public health laboratories often receive samples that have already tested positive for influenza at a clinical laboratory.

We estimated the weekly number of respiratory samples testing positive for influenza A(H3N2), A(H1N1), A(H1N1)pdm09, or B at the HHS region level. We combined pre-2009 seasonal A(H1N1) and A(H1N1)pdm09 as influenza A(H1N1) and the Victoria and Yamagata lineages of influenza B as influenza B. Beginning in the 2015/2016 season, reports from public health and clinical laboratories

are presented separately in the CDC's weekly influenza updates. From 2015 week 40 onwards, we used clinical laboratory data to estimate the proportion of respiratory samples testing positive for any influenza type/subtype and the proportion of samples testing positive for influenza A or B. We used public health laboratory data to estimate the proportion of influenza A isolates typed as A(H3N2) or A(H1N1) in each week. Untyped influenza A-positive isolates were assigned to either A(H3N2) or A(H1N1) according to their proportions among typed isolates.

We defined influenza A subtype dominance in each season based on the proportion of influenza A virus (IAV) positive samples typed as A(H3N2). Specifically, we categorized seasons as A(H3N2) or A(H1N1) dominant when ≥70% of IAV positive samples were typed as one IAV subtype and co-dominant when one IAV subtype comprised 50–69% of IAV positive samples. We applied a strict threshold for subtype dominance because seasons with <70% samples typed as one IAV subtype tended to have greater geographic heterogeneity in circulation, resulting in regions with dominant subtypes that were not nationally dominant.

For each HHS region, we estimated weekly incidences of influenza A(H3N2), A(H1N1), and B by multiplying the percentage of influenza-like illness among outpatient visits, weighted by regional population size, with the percentage of respiratory samples testing positive for each type/subtype (*Figure 1*, *Figure 1—figure supplement 1*). ILI × percent positive (ILI$^+$) is considered a robust estimate of influenza activity and has been used in multiple prior modeling studies (*Bedford et al., 2014*; *Goldstein et al., 2011*; *Pei et al., 2018*). We used linear interpolation to estimate missing values for time spans of up to 4 consecutive weeks.

The emergence of the A(H1N1)pdm09 virus in 2009 altered influenza testing and reporting patterns (*Figure 1—figure supplement 2*). Specifically, the U.S. CDC and WHO increased laboratory testing capacity and strengthened epidemiological networks, which led to substantial improvements to influenza surveillance that are still in place today (*Centers for Disease Control and Prevention, National Center for Immunization and Respiratory Diseases, 2023b*). For each HHS region, we adjusted weekly incidences for increases in reporting rates during the post-pandemic period – defined as the weeks after 2010 week 33 – by scaling pre-pandemic incidences by the ratio of mean weekly ILI$^+$ in the post-pandemic period to that of the pre-pandemic period (1997 week 40–2009 week 17). Incidences for HHS Region 10 were not adjusted for pre- and post-pandemic reporting because surveillance data for this region were not available prior to 2009. To account for differences in reporting rates across HHS regions, we next scaled each region's type/subtype incidences by its mean weekly ILI$^+$ for the entire study period. Scaled incidences were used in all downstream analyses of epidemic burden and timing.

## Characteristics of seasonal influenza epidemics
### Epidemic burden

We considered three complementary indicators of epidemic burden, separately for each influenza type/subtype, HHS region, and season. We defined *peak incidence* as the maximum weekly scaled incidence and *epidemic size* as the cumulative weekly scaled incidence. We estimated *epidemic intensity* based on a method previously developed to study variation in the shape (i.e. sharpness) of influenza epidemics across U.S. cities (*Dalziel et al., 2018*). Epidemic intensity increases when incidence is more concentrated in particular weeks and decreases when incidence is more evenly spread across weeks. Specifically, we defined the incidence distribution $p_{ij}$ as the fraction of influenza incidence in season $j$ that occurred during week $i$ in a given region, and epidemic intensity $v_j$ as the inverse of the Shannon entropy of the weekly incidence distribution:

$$v_j = \left( -\sum_i p_{ij} \ln p_{ij} \right)^{-1} \tag{1}$$

Epidemic intensity is intended to measure the shape and spread of an epidemic, regardless of the actual volume of cases in a given region or season. Following the methodology of Dalziel et al., epidemic intensity values were normalized to fall between 0 and 1 so that epidemic intensity is invariant to differences in reporting rates and/or attack rates across regions and seasons.

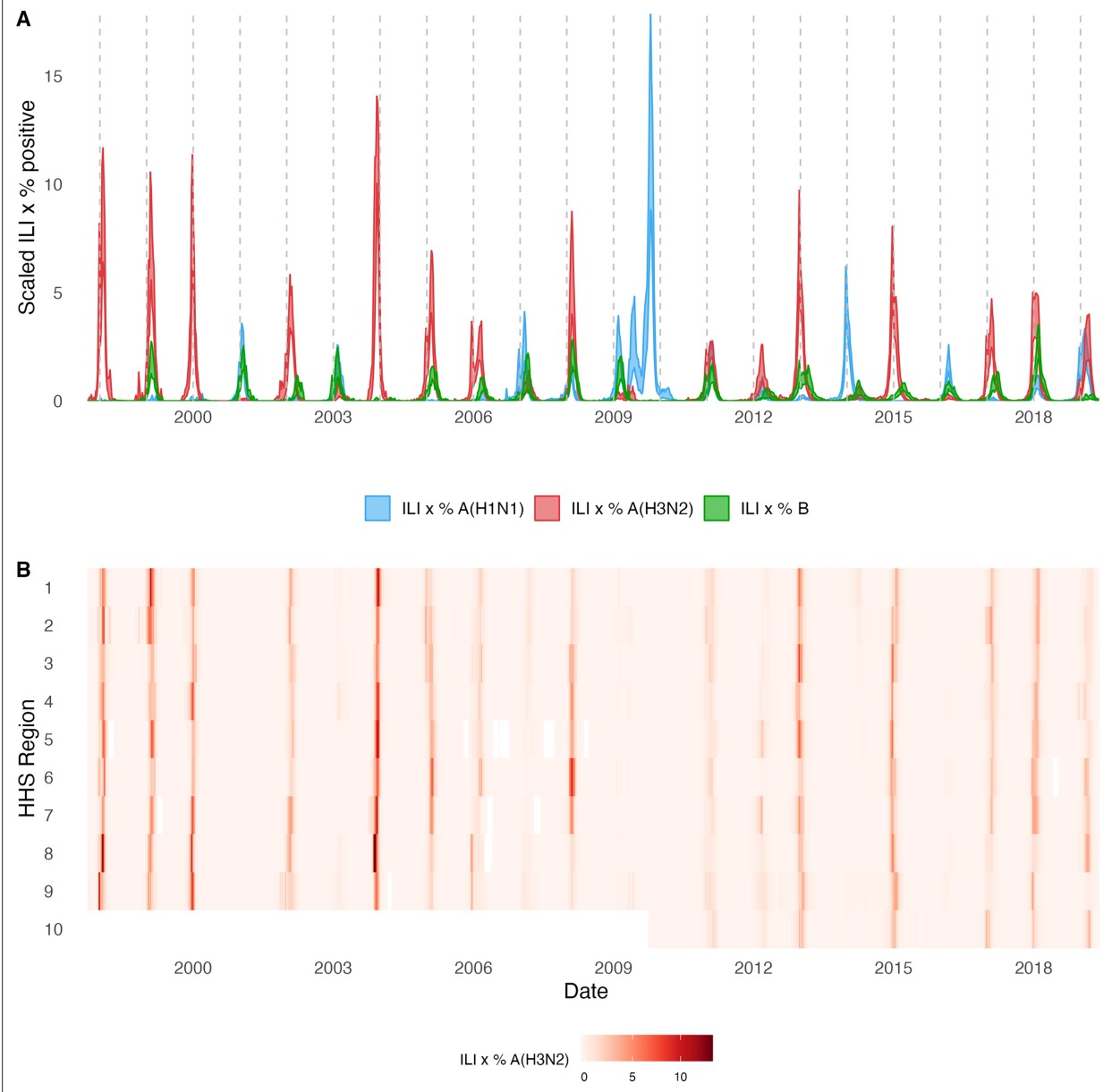

**Figure 1.** Annual influenza A(H3N2) epidemics in the United States, 1997 – 2019. (**A**) Weekly incidence of influenza A(H1N1) (blue), A(H3N2) (red), and B (green) averaged across 10 HHS regions (Region 1: Boston; Region 2: New York City; Region 3: Washington, DC; Region 4: Atlanta; Region 5: Chicago; Region 6: Dallas, Region 7: Kansas City; Region 8: Denver; Region 9: San Francisco; Region 10: Seattle). Incidences are the proportion of influenza-like illness (ILI) visits among all outpatient visits, multiplied by the proportion of respiratory samples testing positive for each influenza type/subtype. Time series are 95% confidence intervals of regional incidence estimates. Vertical dashed lines indicate January 1 of each year. (**B**) Intensity of weekly influenza A(H3N2) incidence in 10 HHS regions. White tiles indicate weeks when influenza-like-illness data or virological data were not reported. Data for Region 10 are not available in seasons prior to 2009.

The online version of this article includes the following figure supplement(s) for figure 1:

**Figure supplement 1.** Annual influenza A(H1N1) and influenza B epidemics in the United States, 1997 - 2019.

**Figure supplement 2.** Influenza test volume systematically increases in all HHS regions after the 2009 A(H1N1) pandemic.

**Figure supplement 3.** Pairwise correlations between seasonal influenza A(H3N2), A(H1N1), and B epidemic metrics.

## Transmission intensity

For each region in each season, we used semi-mechanistic epidemiological models to estimate A(H3N2) virus time-varying (instantaneous) reproduction numbers, $R_t$, by date of infection (Epidemia R package; *Bhatt et al., 2023*; *Scott et al., 2021*). Epidemia implements a Bayesian approach using the probabilistic programming language Stan (*Carpenter et al., 2017*). Prior to $R_t$ estimation, we computed daily A(H3N2) case counts by disaggregating weekly incidence rates to daily rates (temp-disagg R package; *Sax and Steiner, 2013*) and rounding the resultant values to integers.

### Model specifications

Formally, $R_t$ is modeled as:

$$R_t = \exp\left(\beta_o + \epsilon_t^1\right),$$  (2)

$$\beta_o \sim \text{Normal}\left(\log\left(R_o\right), 0.2\right),$$  (3)

$$\epsilon_t^1 \sim \text{Normal}\left(0, \sigma_\epsilon\right),$$  (4)

$$\sigma_\epsilon \sim \text{Half} - \text{Normal}\left(0, 0.01\right),$$  (5)

where exp is the exponential function, the mean of the prior for the intercept $\beta_o$ is the natural log of the basic reproduction number $R_o$ of A(H3N2) virus (1.3) (*Biggerstaff et al., 2014a*), and $\epsilon_t^1$ is a daily random walk process. The steps of the daily walks $\epsilon_t^1$ are independent and centered around 0 with standard deviation $\sigma_\epsilon$.

Instead of using a renewal process to propagate infections, we modeled new infections $i_t$ as unknown latent parameters $i_t'$, because the additional variance around infections can account for uncertainty in initial growth rates, as well as superspreading events (*Bhatt et al., 2023*; *Scott et al., 2021*):

$$i_t \sim \text{Normal}\left(i_t', d\right),$$  (6)

$$d \sim \text{Normal}\left(10, 2\right),$$  (7)

where $d$ is the coefficient of dispersion. This prior assumes that infections have conditional variance around 10 times the conditional mean (*Scott et al., 2021*).

The generation interval distribution $g_k$ is the probability that $s$ days separate the moment of infection in an index case and in an offspring case. For the generation interval, we assumed a discretized Weibull distribution with mean 3.6 days and s.d. 1.6 days (*Cowling et al., 2009*).

Given the generation interval distribution $g_k$, the number of new infections on day $t$ is given by the convolution function:

$$i_t' = R_t \sum_{s<t} i_s g_{t-s},$$  (8)

where $R_t$ is the non-negative instantaneous reproduction number. $R_t$ can be expressed as the number of new infections on day $t$ relative to the cumulative sum of individuals infected $s$ days before day $t$, weighted by the current infectiousness of those individuals (*Cori et al., 2013*; *Gostic et al., 2020*):

$$R_t = \frac{i_t'}{\sum_{s<t} i_s g_{t-s}}$$  (9)

The model is initialized with seeded infections $i_{v:\,0}, v < 0$, which are treated as unknown parameters (*Bhatt et al., 2023*; *Scott et al., 2021*). The prior on $i_{v:\,0}$ assumes that daily seeds are constant over a seeding period of 6 days:

$$i_{-6:\,0} \sim \text{Exponential}\left(\tau^{-1}\right),$$  (10)

$$\tau \sim \text{Exponential}\left(\lambda_0\right),$$  (11)

where $\lambda_0 > 0$ is a rate hyperparameter. $\lambda_0$ is given an uninformative prior (0.03) so that seeds are primarily determined by initial transmission rates and the chosen start date of the epidemic (*Bhatt et al., 2023*; *Scott et al., 2021*).

Daily case counts $Y_t$ are modeled as deriving from past new infections $i_s, s < t$, assuming a negative binomial observation model with mean $y_t$ and overdispersion parameter $\phi$ and a constant infection ascertainment rate $\alpha$ of 0.45 (*Biggerstaff et al., 2014b*). The expected number of observed cases on day $t$ was mapped to past infections by convolving over the time distribution of infection to case observation $\pi_k$:

$$Y_t \sim \text{NegativeBinomial}\left(y_t, \phi\right) \tag{12}$$

$$\phi \sim \text{Normal}\left(10, 5\right) \tag{13}$$

$$\text{logit}\left(y_t\right) = \alpha \left(\sum_{s \leq t} i_s \pi_{t-s}\right) \tag{14}$$

We estimated $\pi_k$ by summing the incubation period distribution and the reporting delay distribution (i.e. the time period from symptom onset to case observation), assuming a lognormal-distributed incubation period with mean 1.4 days and s.d. 1.5 days (*Lessler et al., 2009*) and a lognormal-distributed reporting delay with mean 2 days and s.d. 1.5 days (*Russell et al., 2018*). Thus, the time distribution for infection-to-case-observation was:

$$\pi \sim \text{lognormal}\left(1.4, 1.5\right) + \text{lognormal}\left(2, 1.5\right) \tag{15}$$

Epidemic trajectories for each region and season were fit independently using Stan's Hamiltonian Monte Carlo sampler (*Hoffman and Gelman, 2014*). For each model, we ran four chains, each for 10,000 iterations (including a burn-in period of 2000 iterations that was discarded), producing a total posterior sample size of 32,000. We verified convergence by confirming that all parameters had sufficiently low R-hat values (all R-hat <1.1) and sufficiently large effective sample sizes (>15% of the total sample size).

To generate seasonal indicators of transmission intensity, we extracted posterior draws of daily $R_t$ estimates for each region and season, calculated the median value for each day, and averaged daily median values by epidemic week. For each region and season, we averaged $R_t$ estimates from the weeks spanning epidemic onset to epidemic peak (*initial $R_t$*) and averaged the two highest $R_t$ estimates (*maximum $R_t$*). Initial $R_t$ and maximum $R_t$ produced qualitatively equivalent results in downstream analyses, so we opted to report results for maximum $R_t$.

## Excess pneumonia and influenza deaths attributable to A(H3N2)

To measure the epidemic severity each season, we obtained estimates of seasonal excess mortality attributable to influenza A(H3N2) infections (*Hansen et al., 2022*). Excess mortality is a measure of the mortality burden of a given pathogen in excess of a seasonally adjusted baseline, obtained by regressing weekly deaths from broad disease categories against indicators of influenza virus circulation. Hansen et al. used pneumonia and influenza (P&I) excess deaths, which are considered the most specific indicator of influenza burden (*Simonsen and Viboud, 2012*). Deaths with a mention of P&I (ICD-10 codes J00-J18) were aggregated by week and age group (<1, 1–4, 5–49, 50–64, and ≥65) for seasons 1998–1999 to 2017–2018. Age-specific generalized linear models were fit to observed weekly P&I death rates, while accounting for influenza and respiratory syncytial virus (RSV) activity and seasonal and temporal trends. The weekly national number of excess A(H3N2)-associated deaths were estimated by subtracting the baseline death rate expected in the absence of A(H3N2) virus circulation (A(H3N2) model terms set to zero) from the observed P&I death rate. We summed the number of excess A(H3N2) deaths per 100,000 people from October to May to obtain seasonal age-specific estimates.

## Epidemic timing

### Epidemic onset and peak timing

We estimated the regional onsets of A(H3N2) virus epidemics by detecting breakpoints in A(H3N2) incidence curves at the beginning of each season. The timing of the breakpoint in incidence represents

epidemic establishment (i.e. sustained transmission) rather than the timing of influenza introduction or arrival (*Charu et al., 2017*). We used two methods to estimate epidemic onsets: (1) piecewise regression, which models non-linear relationships with break points by iteratively fitting linear models to each segment (segmented R package; *Muggeo, 2008*; *Muggeo, 2003*), and (2) a Bayesian ensemble algorithm (BEAST – a Bayesian estimator of Abrupt change, Seasonal change, and Trend) that explicitly accounts for the time series nature of incidence data and allows for complex, non-linear trajectories interspersed with change points (Rbeast R package) (*Zhao et al., 2019*). For each region in each season, we limited the time period of breakpoint detection to epidemic week 40 to the first week of maximum incidence and did not estimate epidemic onsets for regions with insufficient signal, which we defined as fewer than three weeks of consecutive incidence and/or greater than 30% of weeks with missing data. We successfully estimated A(H3N2) onset timing for most seasons, except for three A(H1N1) dominant seasons: 2000–2001 (0 regions), 2002–2003 (3 regions), and 2009–2010 (0 regions). Estimates of epidemic onset weeks were similar when using piecewise regression versus the BEAST method, and downstream analyses of correlations between viral fitness indicators and onset timing produced equivalent results. We therefore report results from onsets estimated via piecewise regression. We defined epidemic peak timing as the first week of maximum incidence.

### Epidemic speed

To measure spatiotemporal synchrony of regional epidemic dynamics, we calculated the standard deviation (s.d.) of regional onset and peak timing in each season (*Viboud et al., 2006*; *Wolf et al., 2010*). To measure the speed of viral spread in each region in each season, we measured the number of days spanning onset and peak weeks and seasonal duration (the number of weeks of non-zero incidence). We used two-sided Wilcoxon rank-sum tests to compare the distributions of epidemic timing metrics between A(H3N2) and A(H1N1) dominant seasons.

### Wavelet analysis

As a sensitivity analysis, we used wavelets to estimate timing differences between A(H3N2), A(H1N1), and B epidemics in each HHS region. Incidence time series were square root transformed and normalized and then padded with zeros to reduce edge effects. Wavelet coherence was used to determine the degree of synchrony between A(H3N2) versus A(H1N1) incidence and A(H3N2) versus B incidence within each region at multi-year time scales. Statistical significance was assessed using 10,000 Monte Carlo simulations. Coherence measures time- and frequency-specific associations between two wavelet transforms, with high coherence indicating that two non-stationary signals (time series) are associated at a particular time and frequency (*Johansson et al., 2009*).

Following methodology developed for influenza and other viruses (*Grenfell et al., 2001*; *Johansson et al., 2009*; *Liebhold et al., 2004*; *Viboud et al., 2006*; *Weinberger et al., 2012*), we used continuous wavelet transformations (Morlet) to calculate the phase of seasonal A(H3N2), A(H1N1), and B epidemics. We reconstructed weekly time series of phase angles using wavelet reconstruction (*Torrence and Compo, 1998*; *Viboud et al., 2006*) and extracted the major one-year seasonal component (period 0.8–1.2 years) of the Morlet decomposition of A(H3N2), A(H1N1), and B time series. To estimate the relative timing of A(H3N2) and A(H1N1) incidence or A(H3N2) and B incidence in each region, phase angle differences were calculated as phase in A(H3N2) minus phase in A(H1N1) (or B), with a positive value indicating that A(H1N1) (or B) lags A(H3N2).

### Influenza-like illness age patterns

We calculated the seasonal proportion of ILI encounters in each age group (0–4 years, 5–24 years, 25–64 years, and ≥65 years). Data for more narrow age groups are available after 2009, but we chose these four categories to increase the number of seasons in our analysis.

### Influenza vaccination coverage and A(H3N2) vaccine effectiveness

Influenza vaccination coverage and effectiveness vary between years and would be expected to affect the population impact of seasonal outbreaks, and in turn our epidemiologic indicators. We obtained seasonal estimates of national vaccination coverage for adults 18–49 years and adults ≥65 years from studies utilizing vaccination questionnaire data collected by the National Health Interview Survey (*Centers for Disease Control and Prevention, National Center for Immunization and Respiratory*

*Diseases, 2023b*; *Centers for Disease Control and Prevention, National Center for Immunization and Respiratory Diseases, 2019*; *Jang and Kang, 2021*; *Lu et al., 2019*; *Lu et al., 2013*; *National Health Interview Survey, 2008*; *Ward et al., 2015*; *Ward et al., 2016*). We did not consider the effects of vaccination coverage in children, due to our inability to find published estimates for most influenza seasons in our study.

We obtained seasonal estimates of adjusted A(H3N2) vaccine effectiveness (VE) from 32 observational studies (*Belongia et al., 2011*; *Bridges et al., 2000*; *Castilla et al., 2016*; *Centers for Disease Control and Prevention, National Center for Immunization and Respiratory Diseases, 2023b*; *Centers for Disease Control and Prevention (CDC), 2004*; *Flannery et al., 2019*; *Flannery et al., 2020*; *Flannery et al., 2016*; *Jackson et al., 2017*; *Janjua et al., 2012*; *Kawai et al., 2003*; *Kissling et al., 2013*; *Lester et al., 2003*; *McLean et al., 2014*; *Ohmit et al., 2014*; *Pebody et al., 2017*; *Rolfes et al., 2019*; *Simpson et al., 2015*; *Public Health Agency of Canada, 2005*; *Skowronski et al., 2017a*; *Skowronski et al., 2016*; *Skowronski et al., 2017b*; *Skowronski et al., 2010*; *Skowronski et al., 2009*; *Skowronski et al., 2014a*; *Skowronski et al., 2012*; *Skowronski et al., 2014b*; *Skowronski et al., 2022*; *Skowronski et al., 2007*; *Treanor et al., 2012*; *Valenciano et al., 2018*; *van Doorn et al., 2017*; *Zimmerman et al., 2016*). Most studies had case-control test-negative designs (N=30) and took place in North America (N=25) or Europe (N=6). When possible, we limited VE estimates to those for healthy adults or general populations. When multiple VE studies were available for a given season, we calculated mean VE as the weighted average of $m$ different VE point estimates:

$$\frac{\sum_{i=1}^{m} \delta_{VE_i}^{-1/2} VE_i}{\sum_{i=1}^{m} \delta_{VE_i}^{-1/2}}, \tag{16}$$

wherein $\delta_{VE}$ denotes the width of the 95% confidence interval (CI) for $VE_i$ (*Ndifon et al., 2009*).

The 95% CI for the weighted mean VE was calculated as:

$$\frac{1}{m} \sqrt{\sum_{i=1}^{m} (\delta_{VE_i})^2} \tag{17}$$

## Correlations between seasonal epidemic metrics

We used Spearman's rank correlation coefficients to measure pairwise relationships between A(H3N2), A(H1N1), and B epidemiological indictors. We adjusted $p$-values for multiple testing using the Benjamini and Hochberg method (*Benjamini and Hochberg, 1995*).

## Indicators of influenza A(H3N2) evolution

We considered multiple indicators of influenza evolution based on genetic and phenotypic (serologic) data, separately for HA and NA (*Figure 2*, *Table 1*). Our choice of evolutionary indicators builds on earlier studies that found hemagglutination inhibition (HI) phenotype or HA sequence data beneficial in forecasting seasonal influenza virus evolution (*Huddleston et al., 2020*; *Luksza and Lässig, 2014*; *Neher et al., 2016*; *Neher et al., 2014*) or annual epidemic dynamics (*Axelsen et al., 2014*; *Du et al., 2017*; *Wolf et al., 2010*; *Table 1*).

### HA and NA sequence data

We downloaded all H3 sequences and associated metadata from the Global Initiative on Sharing All Influenza Data (GISAID) EpiFlu database (*Shu and McCauley, 2017*). We focused our analysis on complete H3 sequences that were sampled between January 1, 1997, and October 1, 2019. We prioritized viruses with corresponding HI titer measurements provided by the WHO Global Influenza Surveillance and Response System (GISRS) Collaborating Centers and excluded all egg-passaged viruses and sequences with ambiguous year, month, and day annotations. To account for variation in sequence availability across global regions, we subsampled the selected sequences five times to representative sets of no more than 50 viruses per month, with preferential sampling for North America. Each month up to 25 viruses were selected from North America (when available) and up to 25 viruses were selected from nine other global regions (when available), with even sampling across the other global regions (Africa, Europe, China, South Asia, Japan and Korea, Oceania, South America, Southeast Asia, and West Asia; *Figure 2—figure supplement 1*). To ensure proper topology early in the phylogeny,

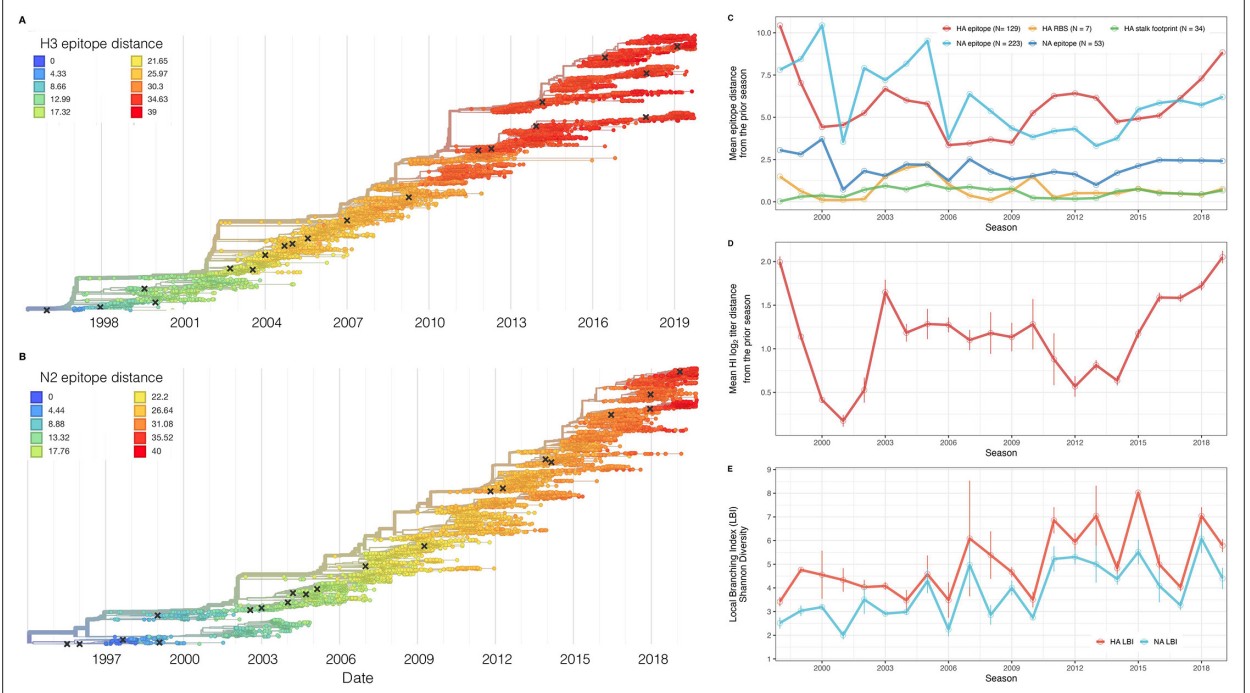

**Figure 2.** Antigenic and genetic evolution of seasonal influenza A(H3N2) viruses, 1997 – 2019. (**A–B**) Temporal phylogenies of (**A**) hemagglutinin (H3) and (**B**) neuraminidase (N2) gene segments. Tip color denotes the Hamming distance from the root of the tree, based on the number of substitutions at epitope sites in H3 (N=129 sites) and N2 (N=223 sites). Black 'X' marks indicate the phylogenetic positions of U.S. recommended vaccine strains. (**C–D**) Seasonal genetic and antigenic distances are the mean distance between A(H3N2) viruses circulating in the current season *t* and viruses circulating in the prior season (*t* – 1), measured by (**C**) five sequence-based metrics (HA epitope (N=129), HA receptor binding site (RBS) (N=7), HA stalk footprint (N=34), NA epitope (N=223 or N=53)) and (**D**) hemagglutination inhibition (HI) titer measurements. (**E**) The Shannon diversity of H3 and N2 local branching index (LBI) values in each season. Vertical bars in (**C**), (**D**), and (**E**) are 95% confidence intervals of seasonal estimates from five bootstrapped phylogenies.

The online version of this article includes the following source data and figure supplement(s) for figure 2:

**Source data 1.** A/H3 sequence counts in five subsampled datasets.

**Source data 2.** A/N2 sequence counts in five subsampled datasets.

**Figure supplement 1.** The number of A/H3 sequences in five subsampled datasets in each month and in each influenza season.

**Figure supplement 2.** The number of A/N2 sequences in five subsampled datasets in each month and in each influenza season.

**Figure supplement 3.** Comparison of seasonal antigenic drift measured by substitutions at H3 epitope sites and HI log₂ titer measurements, from seasons 1997–1998 to 2018–2019.

**Figure supplement 4.** Pairwise correlations between H3 and N2 evolutionary indicators (one-season lags).

**Figure supplement 5.** Pairwise correlations between H3 and N2 evolutionary indicators (two-season lags).

**Figure supplement 6.** Pairwise correlations between H3 and N2 evolutionary indicators (one- and two-season lags).

**Figure supplement 7.** Comparison of seasonal antigenic drift measured by substitutions at H3 and N2 epitope sites, from seasons 1997–1998 to 2018–2019.

we included reference strains that had been collected no earlier than 5 years prior to January 1, 1997. The resultant sets of H3 sequences included 10,060–10,062 sequences spanning December 25, 1995 – October 1, 2019 (*Figure 2—source data 1*). Although our subsampling scheme entailed selecting up to 50 viruses per month, with up to 25 viruses per month collected in North America, each replicate dataset was comprised of approximately 40% North American sequences across all seasons combined (*Figure 2—source data 1*), due to low sequence volumes in the early years of our study.

As with the H3 analysis, we downloaded all N2 sequences and associated metadata from GISAID and selected complete N2 sequences that were sampled between January 1, 1997, and October 1, 2019. We excluded all sequences with ambiguous year, month, and day annotations, forced the inclusion of reference strains collected no earlier than 5 years prior to January 1, 1997, and compiled five

**Table 1.** Evolutionary indicators of seasonal viral fitness.

Evolutionary indicators are labeled by the influenza gene for which data are available (hemagglutinin, HA or neuraminidase, NA), the type of data they are based on, and the component of influenza fitness they represent.

| Evolutionary indicator | Influenza gene | Data type | Fitness category | Citations |
|---|---|---|---|---|
| HI $log_2$ titer distance from the prior season | HA | Hemagglutination inhibition measurements using ferret sera | Antigenic drift | *Huddleston et al., 2020*; *Neher et al., 2016* |
| Epitope distance from the prior season | HA and NA | Sequences | Antigenic drift | *Bhatt et al., 2011*; *Bush et al., 1999*; *Krammer, 2023*; *Webster and Laver, 1980*; *Wiley et al., 1981*; *Wilson and Cox, 1990*; *Wolf et al., 2010* |
| Receptor binding site distance from the prior season | HA | Sequences | Antigenic drift | *Koel et al., 2013* |
| Mutational load (non-epitope distance from the prior season) | HA and NA | Sequences | Functional constraint | *Luksza and Lässig, 2014* |
| Stalk 'footprint' distance from the prior season | HA | Sequences | Negative control | *Kirkpatrick et al., 2018* |
| Local branching index | HA and NA | Sequences | Rate of recent phylogenetic branching | *Huddleston et al., 2020*; *Neher et al., 2014* |

Table format is adapted from *Huddleston et al., 2020*.

replicate subsampled datasets with preferential sampling for North America (8815–8816 sequences; June 8, 1995 – October 1, 2019; *Figure 2—figure supplement 2*, *Figure 2—source data 2*). Similar to the H3 sequence datasets, each replicate dataset was comprised of approximately 40% North American sequences across all seasons combined (*Figure 2—source data 2*).

## HA serologic data

Hemagglutination inhibition (HI) measurements from ferret sera were provided by WHO GISRS Collaborating Centers in London, Melbourne, Atlanta, and Tokyo. We converted raw two-fold dilution measurements to $log_2$ titer drops normalized by the corresponding $log_2$ autologous measurements (*Huddleston et al., 2020*; *Neher et al., 2016*).

Although a phenotypic assay exists for NA, NA inhibiting antibody titers are not routinely measured for influenza surveillance. Therefore, we could not include a phenotypic marker of NA evolution in our study.

## Phylogenetic inference

For each set of H3 and N2 sequences, we aligned sequences with the augur align command (*Hadfield et al., 2018*) and MAFFT v7.407 (*Katoh et al., 2002*). We inferred initial phylogenies with IQ-TREE v1.6.10 (*Nguyen et al., 2015*). To reconstruct time-resolved phylogenies, we applied TreeTime v0.5.6 (*Sagulenko et al., 2018*) with the augur refine command (*Huddleston et al., 2021*).

## Viral fitness metrics

We defined the following fitness metrics for each influenza season:

### Antigenic drift

We estimated antigenic drift of each H3 sequence using either serologic or genetic data.

Historically, HI serological assays were considered the 'gold standard' for measuring immune cross-reactivity between viruses, yet measurements are available for only a subset of viruses. To overcome this limitation, we used a computational approach that maps HI titer measurements onto the HA phylogenetic tree to infer antigenic phenotypes (*Huddleston et al., 2020*; *Neher et al., 2016*). Importantly, this model infers the antigenicity of virus isolates that lack HI titer measurements, which

comprise the majority of HA sequences in GISAID. To estimate antigenic drift with hemagglutination inhibition (HI) titer data, hereon *HI log$_2$ titer distance*, we applied the phylogenetic tree model from *Neher et al., 2016* to the H3 phylogeny and the available HI data for its sequences. The tree model estimates the antigenic drift per branch in units of *log$_2$* titer change.

Our sequence-based measures of drift counted substitutions at putative epitope sites in the globular head domains of HA and NA, identified through monoclonal antibody escape or protein crystal structure: 129 sites in HA epitope regions A to E (*Bush et al., 1999*; *Webster and Laver, 1980*; *Wiley et al., 1981*; *Wilson and Cox, 1990*; *Wolf et al., 2006*) (*HA epitope distance*), 7 sites adjacent to the HA receptor binding site (RBS) (*Koel et al., 2013*) (*HA RBS distance*), and 223 or 53 sites in NA epitope regions A to C (*Bhatt et al., 2011*; *Krammer, 2023*) (*NA epitope distance*). We also counted the number of substitutions at epitope sites in the HA stalk domain (*HA stalk footprint distance*) (*Kirkpatrick et al., 2018*). Although the majority of the antibody-mediated response to HA is directed to the immunodominant HA head, antibodies towards the highly conserved immunosubdominant stalk domain of HA are widely prevalent in older individuals, although at low levels (*Krammer, 2019*; *Margine et al., 2013*; *Nachbagauer et al., 2016*). We considered stalk footprint distance to be our 'control' metric for drift, given the HA stalk evolves at a significantly slower rate than the HA head (*Kirkpatrick et al., 2018*).

### Mutational load

To estimate mutational load for each H3 and N2 sequence, an inverse proxy of viral fitness (*Huddleston et al., 2020*; *Luksza and Lässig, 2014*), we implemented metrics that count substitutions at putative non-epitope sites in HA (N=200) and NA (N=246), hereon *HA non-epitope distance* and *NA non-epitope distance*. Mutational load produces higher values for viruses that are less fit compared to previously circulating strains.

### Clade growth

The local branching index (LBI) measures the relative fitness of co-circulating clades, with high LBI values indicating recent rapid phylogenetic branching (*Huddleston et al., 2020*; *Neher et al., 2014*). To calculate LBI for each H3 and N2 sequence, we applied the LBI heuristic algorithm as originally described by *Neher et al., 2014* to H3 and N2 phylogenetic trees, respectively. We set the neighborhood parameter $\tau$ to 0.4 and only considered viruses sampled between the current season $t$ and the previous season $t − 1$ as contributing to recent clade growth in the current season $t$.

Variation in the phylogenetic branching rates of co-circulating A(H3N2) clades may affect the magnitude, intensity, onset, or duration of seasonal epidemics. For example, we expected that seasons dominated by a single variant with high fitness might have different epidemiological dynamics than seasons with multiple co-circulating clades with varying seeding and establishment times. We measured the diversity of clade growth rates of viruses circulating in each season by measuring the standard deviation (s.d.) and Shannon diversity of LBI values in each season. Given that LBI measures *relative* fitness among co-circulating clades, we did not compare overall clade growth rates (e.g. mean LBI) across seasons.

Each season's distribution of LBI values is right-skewed and does not follow a normal distribution. We therefore bootstrapped the LBI values of each season in each replicate dataset 1000 times (1000 samples with replacement) and estimated the seasonal standard deviation of LBI from resamples, rather than directly from observed LBI values. We also tested the seasonal standard deviation of LBI from log transformed LBI values, which produced qualitatively equivalent results to bootstrapped LBI values in downstream analyses.

As an alternative measure of seasonal LBI diversity, we binned raw H3 and N2 LBI values into categories based on their integer values (e.g. an LBI value of 0.5 is assigned to the (0,1] bin) and estimated the exponential of the Shannon entropy (*Shannon diversity*) of LBI categories (*Hill, 1973*; *Shannon, 1948*). The Shannon diversity of LBI considers both the richness and relative abundance of viral clades with different growth rates in each season and is calculated as follows:

$$^1D = \exp\left(-\sum_{i=1}^{R} p_i \ln p_i\right),$$

(18)

where $^qD$ is the effective number of categories or Hill numbers of order $q$ (here, clades with different growth rates), with $q$ defining the sensitivity of the true diversity to rare versus abundant categories (*Hill, 1973*). exp is the exponential function, $p_i$ is the proportion of LBI values belonging to the $i$th category, and $R$ is richness (the total number of categories). Shannon diversity $^1D$ ($q = 1$) estimates the effective number of categories in an assemblage using the geometric mean of their proportional abundances (*Hill, 1973*).

Because ecological diversity metrics are sensitive to sampling effort, we rarefied H3 and N2 sequence datasets prior to estimating Shannon diversity so that seasons had the same sample size. For each season in each replicate dataset, we constructed rarefaction and extrapolation curves of LBI Shannon diversity and extracted the Shannon diversity estimate of the sample size that was twice the size of the reference sample size (the smallest number of sequences obtained in any season during the study) (iNEXT R package; *Chao et al., 2014*). Chao et al. found that their diversity estimators work well for rarefaction and short-range extrapolation when the extrapolated sample size is up to twice the reference sample size. For H3, we estimated seasonal diversity using replicate datasets subsampled to 360 sequences/season; For N2, datasets were subsampled to 230 sequences/season.

## Antigenic and genetic distance relative to prior seasons

For each replicate dataset, we estimated national-level genetic and antigenic distances between influenza viruses circulating in consecutive seasons by calculating the mean distance between viruses circulating in the current season $t$ and viruses circulating during the prior season ($t - 1$ year; one-season lag) or two prior seasons ago ($t - 2$ years; two-season lag). We then averaged seasonal mean distances across the five replicate datasets. Seasonal genetic and antigenic distances are greater when currently circulating strains are more antigenically distinct from previously circulating strains. We used Spearman's rank correlation coefficients to measure pairwise relationships between scaled H3 and N2 evolutionary indicators. We adjusted $p$-values for multiple testing using the Benjamini and Hochberg method (*Benjamini and Hochberg, 1995*).

## Univariate relationships between viral fitness, (sub)type interference and A(H3N2) epidemic impact

We measured univariate associations between national indicators of A(H3N2) viral fitness and regional A(H3N2) epidemic parameters: peak incidence, epidemic size, transmissibility (effective $R_t$), epidemic intensity, subtype dominance, excess P&I deaths, onset timing, peak timing, spatiotemporal synchrony, the number of weeks from onset to peak, and seasonal duration. All predictors were centered and scaled prior to measuring correlations or fitting regression models.

We first measured Spearman's rank correlation coefficients between pairs of scaled evolutionary indicators and epidemic metrics using 1000 bootstrap replicates of the original dataset (1000 samples with replacement). Next, we fit regression models with different distribution families (Gaussian or Gamma) and link functions (identity, log, or inverse) to observed data and used Bayesian information criterion (BIC) to select the best fit model, with lower BIC values indicating a better fit to the data. For subtype dominance, epidemic intensity, and age-specific proportions of ILI cases, we fit Beta regression models with logit links. Beta regression models are appropriate when the variable of interest is continuous and restricted to the interval (0, 1) (*Ferrari and Cribari-Neto, 2004*). For each epidemic metric, we fit the best-performing regression model to 1000 bootstrap replicates of the original dataset.

To measure the effects of sub(type) interference on A(H3N2) epidemics, the same approach was applied to measure the univariate relationships between A(H1N1) or B epidemic size and A(H3N2) peak incidence, epidemic size, effective $R_t$, epidemic intensity, and excess mortality. As a sensitivity analysis, we evaluated univariate relationships between A(H3N2) epidemic metrics and A(H1N1) epidemic size during pre-2009 seasons (seasonal A(H1N1) viruses) and post-2009 seasons (A(H1N1) pdm09 viruses) separately.

## Selecting relevant predictors of A(H3N2) epidemic impact

Next, we explored multivariable approaches that would shed light on the potential mechanisms driving annual epidemic impact. Considering that we had many predictors and relatively few observations (22 seasons × 9–10 HHS regions), several covariates were collinear, and our goal was explicative

rather than predictive, we settled on methods that tend to select few covariates: conditional inference random forests and LASSO (least absolute shrinkage and selection operator) regression models. All predictors were centered and scaled prior to fitting models.

## Preprocessing of predictor data

The starting set of candidate predictors included all viral fitness metrics: genetic and antigenic distances between current and previously circulating viruses and the standard deviation and Shannon diversity of H3 and N2 LBI values in the current season. To account for potential type or subtype interference, we included A(H1N1) or A(H1N1)pdm09 epidemic size and B epidemic size in the current and prior season and the dominant IAV subtype in the prior season (*Lee et al., 2018*). We included A(H3N2) epidemic size in the prior season as a proxy for prior natural immunity to A(H3N2). To account for vaccine-induced immunity, we considered four categories of predictors and included estimates for the current and prior seasons: national vaccination coverage among adults (18–49 years coverage × ≥65 years coverage), adjusted A(H3N2) vaccine effectiveness (VE), a combined metric of vaccination coverage and A(H3N2) VE (18–49 years coverage × ≥65 years coverage × VE), and H3 and N2 epitope distances between naturally circulating A(H3N2) viruses and the U.S. A(H3N2) vaccine strain in each season. We could not include a predictor for vaccination coverage in children or consider clade-specific VE estimates because these data were not available for most seasons in our study.

Random forest and LASSO regression models are not sensitive to redundant (highly collinear) features (*Kuhn and Johnson, 2019*), but we chose to downsize the original set of candidate predictors to minimize the impact of multicollinearity on variable importance scores. For both types of models, if there are highly collinear variables that are useful for predicting the target variable, the predictor chosen by the model becomes a random selection (*Kuhn and Johnson, 2019*). In random forest models, these highly collinear variables will be used in all splits across the forest of decision trees, and this redundancy dilutes variable importance scores (*Kuhn and Johnson, 2019*). We first confirmed that none of the candidate predictors had zero variance or near-zero variance. Because seasonal lags of each viral fitness metric are highly collinear, we included only one lag of each evolutionary predictor, with a preference for the lag that had the strongest univariate correlations with various epidemic metrics. We checked for multicollinearity among the remaining predictors by examining Spearman's rank correlation coefficients between all pairs of predictors. If a particular pair of predictors was highly correlated (Spearman's $\rho > 0.8$), we retained only one predictor from that pair, with a preference for the predictor that had the strongest univariate correlations with various epidemic metrics. Lastly, we performed QR decomposition of the matrix of remaining predictors to determine if the matrix is full rank and identify sets of columns involved in linear dependencies. This step did not eliminate any additional predictors, given that we had already removed pairs of highly collinear variables based on Spearman correlation coefficients.

After these preprocessing steps, our final set of model predictors included 21 variables, including 8 viral evolutionary indicators: H3 epitope distance ($t − 2$), HI log$_2$ titer distance ($t − 2$), H3 RBS distance ($t − 2$), H3 non-epitope distance ($t − 2$), N2 epitope distance ($t − 1$), N2 non-epitope distance ($t − 1$), and H3 and N2 LBI diversity (s.d.) in the current season; 6 proxies for type/subtype interference and prior immunity: A(H1N1) and B epidemic sizes in the current and prior season, A(H3N2) epidemic size in the prior season, and the dominant IAV subtype in the prior season; and 7 proxies for vaccine-induced immunity: A(H3N2) VE in the current and prior season, H3 and N2 epitope distances between circulating viruses and the vaccine strain in each season, the combined metric of adult vaccination coverage × VE in the current and prior season, and adult vaccination coverage in the prior season.

## Random forest models

We used conditional inference random forest models to select relevant predictors of A(H3N2) epidemic size, peak incidence, transmissibility (effective $R_t$), epidemic intensity, and subtype dominance (party and caret R packages; *Hothorn et al., 2006*; *Kuhn, 2008*; *Strobl et al., 2008*; *Strobl et al., 2007*). We did not conduct variable selection analysis for excess A(H3N2) mortality due to data limitations (one national estimate per season). Metrics related to epidemic timing were also excluded from this analysis because we found weak or non-statistically significant associations with most viral fitness metrics in univariate analyses. Lastly, we could not separate our analysis into pre- and post-2009 pandemic periods due to small sample sizes.

We created each forest by generating 3000 regression trees. To determine the best performing model for each epidemic metric, we used leave-one-season-out (jackknife) cross-validation to train models and measure model performance, wherein each 'assessment' set is one season of data predicted by the model, and the corresponding 'analysis' set contains the remaining seasons. This approach is roughly analogous to splitting data into training and test sets, but all seasons are used at some point in the training of each model (*Kuhn and Johnson, 2019*). Due to the small size of our dataset (~20 seasons), evaluating the predictive accuracy of random forest models on a quasi-independent test set of 2–3 seasons produced unstable estimates. Instead of testing model performance on an independent test set, we generated 10 bootstrap resamples ('repeats') of each analysis set ('fold') and averaged the predictions of models trained on resamples (*Kuhn and Johnson, 2013*; *Kuhn and Johnson, 2019*). For each epidemic metric, we report the mean root mean squared error (RMSE) and $R^2$ of predictions from the best tuned model. We used permutation importance (N=50 permutations) to estimate the relative importance of each predictor in determining target outcomes. Permutation importance is the decrease in prediction accuracy when a single feature (predictor) is randomly permuted, with larger values indicating more important variables. Because many features were collinear, we used conditional permutation importance to compute feature importance scores, rather than the standard marginal procedure (*Altmann et al., 2010*; *Debeer and Strobl, 2020*; *Strobl et al., 2008*; *Strobl et al., 2007*).

## Regression models

As an alternative method for variable selection, we performed LASSO regression on the same cross-validated dataset and report the mean RMSE and $R^2$ of predictions from the best tuned model (glmnet and caret R packages; *Friedman et al., 2010*; *Kuhn, 2008*). Unlike random forest models, this modeling approach assumes linear relationships between predictors and the target variable. LASSO models (L1 penalty) are more restrictive than ridge models (L2 penalty) and elastic net models (combination of L1 and L2 penalties) and will arbitrarily retain one variable from a set of collinear variables.

To further reduce the set of predictors for each epidemic metric, we performed model selection with linear regression models that considered all combinations of the top 10 ranked predictors from conditional inference random forest models. Candidate models could include up to three predictors, and models were compared using BIC. We did not include HHS region or season as fixed or random effects because these variables either did not improve model fit (region) or caused overfitting and convergence issues (season).

## Results

### Indicators of influenza A(H3N2) evolution

We characterized seasonal patterns of genetic and antigenic evolution among A(H3N2) viruses circulating during 1997–2019, using HA and NA sequence data shared via the GISAID EpiFlu database (*Shu and McCauley, 2017*) and ferret hemagglutination inhibition (HI) assay data shared by WHO GISRS Collaborating Centers in London, Melbourne, Atlanta, and Tokyo. Time-resolved phylogenies of HA and NA genes are shown in *Figure 2*. Although our study is U.S.-focused, we used a global dataset because U.S.-collected sequences and HI titers were sometimes sparse during the earlier seasons of the study (*Figure 2—figure supplements 1 and 2*).

To measure antigenic distances between consecutive seasons, we calculated mean genetic distances at epitope sites or mean $\log_2$ titer distances from HI titer measurements (*Figure 2*), between viruses circulating in the current season $t$ and the prior season $t - 1$ year (one-season lag) or two prior seasons ago $t - 2$ years (two-season lag). These time windows generated seasonal antigenic distances consistent with empirical and theoretical studies characterizing transitions between H3 or N2 antigenic clusters (*Bedford et al., 2014*; *Ferguson et al., 2003*; *Huddleston et al., 2020*; *Neher et al., 2014*; *Sandbulte et al., 2011*; *Smith et al., 2004*), with H3 epitope distance and HI $\log_2$ titer distance, at two-season lags, and N2 epitope distance, at one-season lags, capturing expected 'jumps' in antigenic drift during key seasons that have been previously associated with major antigenic transitions (*Smith et al., 2004*), such as the seasons dominated by A/Sydney/5/1997-like strains (SY97) (1997–1998, 1998–1999, 1999–2000) and the 2003–2004 season dominated by A/Fujian/411/2002-like strains (FU02) (*Figure 2—figure supplements 3 and 7*). Prior studies explicitly linking antigenic

**Table 2.** Seasonal metrics of A(H3N2) epidemic dynamics.
Epidemic metrics are defined and labeled by which outcome category they represent.

| Epidemic Outcome | Definition | Outcome category | Citations |
|---|---|---|---|
| Epidemic size | Cumulative weekly incidence | Burden | |
| Peak incidence | Maximum weekly incidence | Burden | |
| Maximum time-varying effective reproduction number, $R_t$ | The number of secondary cases arising from a symptomatic index case, assuming conditions remain the same | Transmissibility | *Scott et al., 2021*; *Bhatt et al., 2023* |
| Epidemic intensity | Inverse Shannon entropy of the weekly incidence distribution (i.e. the spread of incidence across the season) | Sharpness of the epidemic curve | *Dalziel et al., 2018* |
| Subtype dominance | The proportion of influenza positive samples typed as A(H3N2) | Viral activity | |
| Excess pneumonia and influenza mortality attributable to A(H3N2) virus | Mortality burden in excess of a seasonally adjusted baseline | Severity | *Hansen et al., 2022*; *Simonsen and Viboud, 2012* |
| Onset week | Winter changepoint in incidence | Timing | *Charu et al., 2017* |
| Peak week | First week of maximum incidence | Timing | |
| Spatiotemporal synchrony | Regional variation (s.d.) in onset or peak timing | Speed | *Viboud et al., 2006* |
| Onset to peak | Number of days between onset week and peak week | Speed | |
| Seasonal duration | Number of weeks with non-zero incidence | Speed | |

drift to epidemic size or severity also support a 1-year (*Bedford et al., 2014*) or 2-year time window of drift (*Koelle et al., 2006*; *Wolf et al., 2010*). Given that protective immunity to homologous strains wanes after 1–4 years (*He et al., 2015*; *Wraith et al., 2022*), we would also expect these timeframes to return the greatest signal in epidemiological surveillance data.

We measured pairwise correlations between seasonal indicators of HA and NA evolution to assess their degree of concordance. As expected, we found moderate-to-strong associations between HA epitope distance and HI log$_2$ titer distance (*Figure 2—figure supplements 3–6*) and HA RBS distance and HI log$_2$ titer distance (*Figure 2—figure supplements 4–6*). Consistent with prior serological studies (*Eichelberger et al., 2018*; *Kilbourne et al., 1990*; *Schulman and Kilbourne, 1969*), epitope distances in HA and NA were not correlated at one-season lags (Spearman's $\rho$=0.25, $p$=0.3) or two-season lags ($\rho$=0.15, $p$=0.5) (*Figure 2—figure supplements 4–7*). The seasonal diversity of HA and NA LBI values was negatively correlated with NA epitope distance (*Figure 2—figure supplements 5 and 6*), with high antigenic novelty coinciding with low genealogical diversity. This association suggests that selective sweeps tend to follow the emergence of drifted variants with high fitness, resulting in seasons dominated by a single A(H3N2) variant rather than multiple co-circulating clades.

## Associations between A(H3N2) evolution and epidemic dynamics

We explored relationships between viral evolution and variation in A(H3N2) epidemic dynamics from seasons 1997–1998 to 2018–2019, excluding the 2009 A(H1N1) pandemic, using syndromic and virologic surveillance data collected by the U.S. CDC and WHO. We estimated weekly incidences of influenza A(H3N2), A(H1N1), and B in 10 HHS regions by multiplying the influenza-like illness (ILI) rate – the proportion of outpatient encounters for ILI, weighted by regional population size – by the regional proportion of respiratory samples testing positive for each influenza type/subtype (percent positive). *Figure 1* and *Figure 1—figure supplement 1* show variability in the timing and intensity of annual epidemics of A(H3N2), A(H1N1), and B viruses. Based on these incidence time series, we measured indicators of epidemic burden, intensity, severity, subtype dominance, timing, and age-specific patterns during each non-pandemic season (*Table 2*) and assessed their univariate relationships with each indicator of HA and NA evolution. *Figure 1—figure supplement 3* shows pairwise correlations between epidemic metrics.

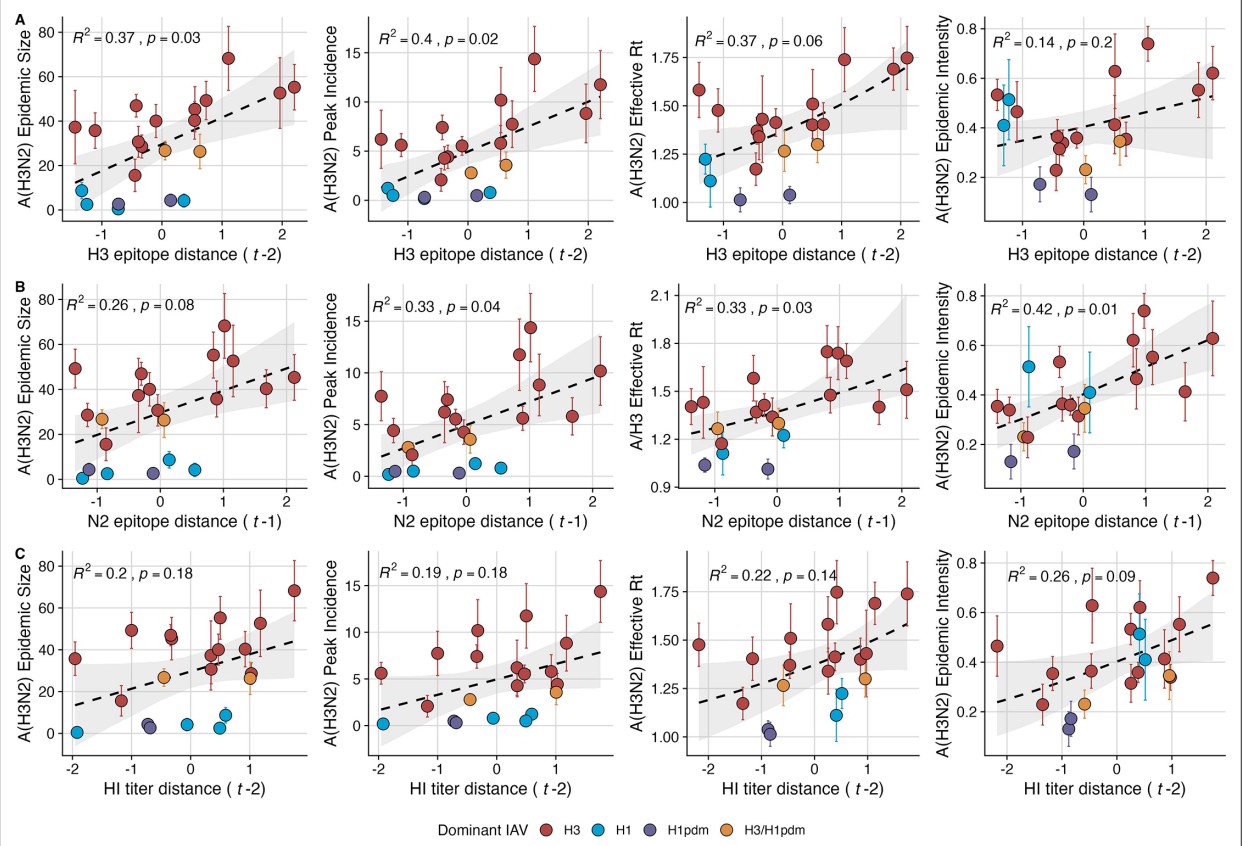

**Figure 3.** Influenza A(H3N2) antigenic drift correlates with larger, more intense annual epidemics. A(H3N2) epidemic size, peak incidence, transmissibility (effective reproduction number, $R_t$), and epidemic intensity increase with antigenic drift, measured by (**A**) hemagglutinin (H3) epitope distance, (**B**) neuraminidase (N2) epitope distance, and (**C**) hemagglutination inhibition (HI) $\log_2$ titer distance. Seasonal antigenic drift is the mean titer distance or epitope distance between viruses circulating in the current season $t$ and viruses circulating in the prior season ($t – 1$) or two prior seasons ago ($t – 2$). Distances are scaled to aid in direct comparison of evolutionary indicators. Point color indicates the dominant influenza A virus (IAV) subtype based on CDC influenza season summary reports (red: A(H3N2), blue: A(H1N1), purple: A(H1N1)pdm09, orange: A(H3N2)/A(H1N1)pdm09 co-dominant), and vertical bars are 95% confidence intervals of regional estimates (pre-2009 seasons: 9 regions; post-2009 seasons: 10 regions). Seasonal mean A(H3N2) epidemic metric values were fit as a function of antigenic or genetic distance using LMs (epidemic size, peak incidence), Gaussian GLMs (effective $R_t$: inverse link), or Beta GLMs (epidemic intensity: logit link) with 1000 bootstrap resamples. In each plot, the black dashed line represents the mean regression fit, and the gray shaded band shows the 95% confidence interval, based on 1000 bootstrap resamples. The $R^2$ and associated $p$-value from the mean regression fit are in the top left section of each plot.

The online version of this article includes the following figure supplement(s) for figure 3:

**Figure supplement 1.** Univariate correlations between influenza A(H3N2) evolutionary indictors and epidemic impact.

**Figure supplement 2.** Excess influenza A(H3N2) mortality increases with H3 and N2 epitope distance, but correlations are not statistically significant.

**Figure supplement 3.** Low seasonal diversity in the clade growth rates of circulating A(H3N2) viruses, as measured by the standard deviation of local branching index values, correlates with higher transmissibility and greater epidemic intensity.

**Figure supplement 4.** Low seasonal diversity in the clade growth rates of circulating A(H3N2) viruses, as measured by the Shannon diversity of local branching index values, correlates with higher transmissibility and greater epidemic intensity.

Two sequence-based measures based on broad sets of epitope sites exhibited stronger relationships with seasonal A(H3N2) epidemic burden and transmissibility than the serology-based measure, HI $\log_2$ titer distance. Both H3 epitope distance ($t – 2$) and N2 epitope distance ($t – 1$) correlated with increased epidemic size (H3, adjusted $R^2$=0.37, $p$=0.03; N2: $R^2$=0.26, $p$=0.08) and peak incidence (H3: $R^2$=0.4, $p$=0.02; N2: $R^2$=0.33, $p$=0.04) and higher effective reproduction numbers, $R_t$ (H3, $R^2$=0.37, $p$=0.06; N2, $R^2$=0.33, $p$=0.03; regression results: **Figure 3**; Spearman correlations: **Figure 3—figure supplement 1**). Excess pneumonia and influenza mortality attributable to A(H3N2) increased with H3 epitope distance, though this relationship was not statistically significant (**Figure 3—figure supplement 2**). HI $\log_2$ titer distance ($t – 2$) exhibited positive but non-significant associations with different measures of epidemic

impact (*Figure 3*, *Figure 3—figure supplement 1*). Effective $R_t$ and epidemic intensity were greater in seasons with low LBI diversity (*Figure 3—figure supplement 1*; *Figure 3—figure supplement 3* and *Figure 3—figure supplement 4*). The remaining indicators of viral evolution, including H3 and N2 non-epitope distance (mutational load), H3 RBS distance, and H3 stalk footprint distance had weaker, non-statistically significant correlations with epidemic impact (*Figure 3—figure supplement 1*).

We explored whether evolutionary changes in A(H3N2) may predispose this subtype to dominate influenza virus circulation in a given season. A(H3N2) subtype dominance – the proportion of influenza positive samples typed as A(H3N2) – increased with H3 epitope distance ($t - 2$) ($R^2$=0.32, $p$=0.05) and N2 epitope distance ($t - 1$) ($R^2$=0.34, $p$=0.03) (regression results: *Figure 4*; Spearman correlations: *Figure 3—figure supplement 1*). *Figure 4* illustrates this relationship at the regional level across two seasons in which A(H3N2) was nationally dominant, but where antigenic change differed. In 2003–2004, we observed widespread dominance of A(H3N2) viruses after the emergence of the novel antigenic cluster, FU02 (A/Fujian/411/2002-like strains). In contrast, there was substantial regional heterogeneity in subtype circulation during 2007–2008, a season in which A(H3N2) viruses were antigenically similar to those circulating in the previous season. Patterns in type/subtype circulation across all influenza seasons in our study period are shown in *Figure 4—figure supplement 1*. As observed for the 2003–2004 season, widespread A(H3N2) dominance tended to coincide with major antigenic transitions (e.g. A/Sydney/5/1997 (SY97) seasons, 1997–1998 to 1999–2000; A/California/7/2004 (CA04) season, 2004–2005), although this was not universally the case (e.g. A/Perth/16/2009 (PE09) season, 2010–2011).

After the 2009 A(H1N1) pandemic, A(H3N2) dominant seasons still occurred more frequently than A(H1N1) dominant seasons, but the mean fraction of influenza positive cases typed as A(H3N2) in A(H3N2) dominant seasons was lower compared to A(H3N2) dominant seasons prior to 2009 (*Figure 4—figure supplement 1*). Antigenically distinct 3 c.2a and 3 c.3a viruses began to co-circulate in 2012 and underwent further diversification during subsequent seasons in our study (https://nextstrain.org/seasonal-flu/h3n2/ha/12y@2024-05-13; *Dhanasekaran et al., 2022*; *Huddleston et al., 2020*; *Yan et al., 2019*). The decline in A(H3N2) predominance during the post-2009 period may be linked to the genetic and antigenic diversification of A(H3N2) viruses, wherein multiple lineages with similar fitness co-circulated in each season.

Next, we tested for associations between A(H3N2) evolution and various measures of epidemic timing (*Table 2*). Seasonal duration increased with H3 and N2 LBI diversity in the current season (H3, LBI Shannon diversity: $R^2$=0.37; $p$=0.04; LBI s.d.: $R^2$=0.3; $p$=0.09; N2, LBI Shannon diversity: $R^2$=0.38; $p$=0.04; LBI s.d.: $R^2$=0.36; $p$=0.06; regression results: *Figure 5*; Spearman correlations: *Figure 5—figure supplement 1*), while the number of days from epidemic onset to peak incidence shortened with increasing N2 epitope distance ($t - 1$) ($R^2$=0.38, $p$=0.03; *Figure 5—figure supplement 2*). Onset and peak timing tended to be earlier in seasons with increased H3 and N2 antigenic novelty, but correlations between antigenic change and epidemic timing were not statistically significant (*Figure 5—figure supplement 3*). A(H3N2) evolution did not correlate with the degree of spatiotemporal synchrony across HHS regions (*Figure 5—figure supplement 1*).

Lastly, we considered the effects of antigenic change on the age distribution of outpatient ILI cases, with the expectation that the proportion of cases in children would decrease in seasons with greater antigenic novelty, due to drifted variants' increased ability to infect more immunologically experienced adults (*Bedford et al., 2015*; *Gostic et al., 2019*). Consistent with this hypothesis, N2 epitope distance was negatively correlated with the fraction of cases in children aged <5 years (one-season lag: $R^2$=0.29, $p$=0.1; two-season lag: $R^2$=0.59, $p$=0.003) and individuals aged 5–24 years (one-season lag: $R^2$=0.38, $p$=0.04; two-season lag: $R^2$=0.17, $p$=0.18) and positively correlated with the fraction of cases in adults aged 25–64 years (one-season lag: $R^2$=0.36, $p$=0.05; two-season lag: $R^2$=0.49, $p$=0.01) and ≥65 years (one-season lag: $R^2$=0.39, $p$=0.01; two-season lag: $R^2$=0.33, $p$=0.05) (regression results: *Figure 6*; Spearman correlations: *Figure 6—figure supplement 1*). Antigenic drift in H3 exhibited similar associations with age patterns of ILI cases, but correlations were weaker and non-significant (*Figure 6*, *Figure 6—figure supplement 1*).

## Effects of heterosubtypic viral interference on A(H3N2) epidemic burden and timing

We investigated the effects of influenza type/subtype interference – proxied by influenza A(H1N1) and B epidemic size – on A(H3N2) incidence during annual outbreaks. Across the entire study period, we

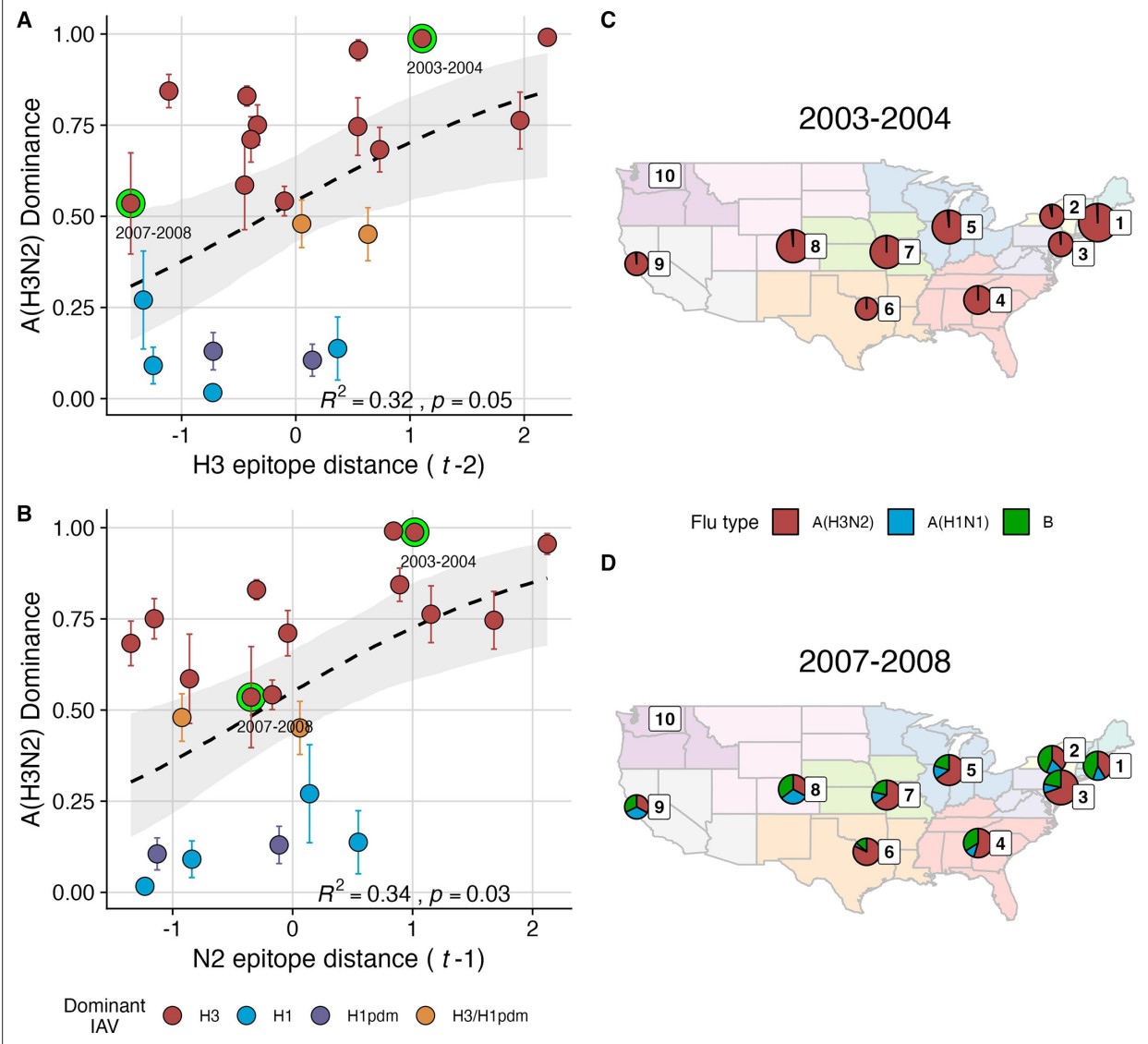

**Figure 4.** The proportion of influenza positive samples typed as A(H3N2) increases with antigenic drift. (**A-B**) Seasonal A(H3N2) subtype dominance increases with (**A**) hemagglutinin (H3) and (**B**) neuraminidase (N2) epitope distance. Seasonal epitope distance is the mean epitope distance between viruses circulating in the current season $t$ and viruses circulating in the prior season ($t$ - 1) or two prior seasons ago ($t$ - 2). Distances were scaled to aid in direct comparison of evolutionary indicators. Point color indicates the dominant influenza A virus (IAV) subtype based on CDC influenza season summary reports (red: A(H3N2), blue: A(H1N1), purple: A(H1N1)pdm09, orange: A(H3N2)/A(H1N1)pdm09 co-dominant), and vertical bars are 95% confidence intervals of regional estimates (pre-2009 seasons: 9 regions; post-2009 seasons: 10 regions). Seasonal mean A(H3N2) dominance was fit as a function of H3 or N2 epitope distance using Beta GLMs with 1000 bootstrap resamples. In (**A**) and (**B**), the dashed black line represents the mean regression fit, and the gray shaded band shows the 95% confidence interval, based on 1000 bootstrap resamples. The $R^2$ and associated $p$-value from the mean regression fit are in the bottom right section of each plot. (**C–D**) Regional patterns of influenza type and subtype incidence during two seasons when A(H3N2) was nationally dominant. Pie charts represent the proportion of influenza positive samples typed as A(H3N2) (red), A(H1N1) (blue), or B (green) in each HHS region. The sizes of regional pie charts are proportional to the total number of influenza positive samples. Data for Region 10 (purple) are not available for seasons prior to 2009. (**C**) Widespread A(H3N2) dominance during 2003–2004 after the emergence of a novel antigenic cluster, FU02 (A/Fujian/411/2002-like strains). (**D**) Spatial heterogeneity in subtype circulation during 2007–2008, a season with low A(H3N2) antigenic novelty relative to the prior season.

The online version of this article includes the following figure supplement(s) for figure 4:

**Figure supplement 1.** Regional patterns of influenza type and subtype circulation during seasons 1997–1998 to 2018–2019.

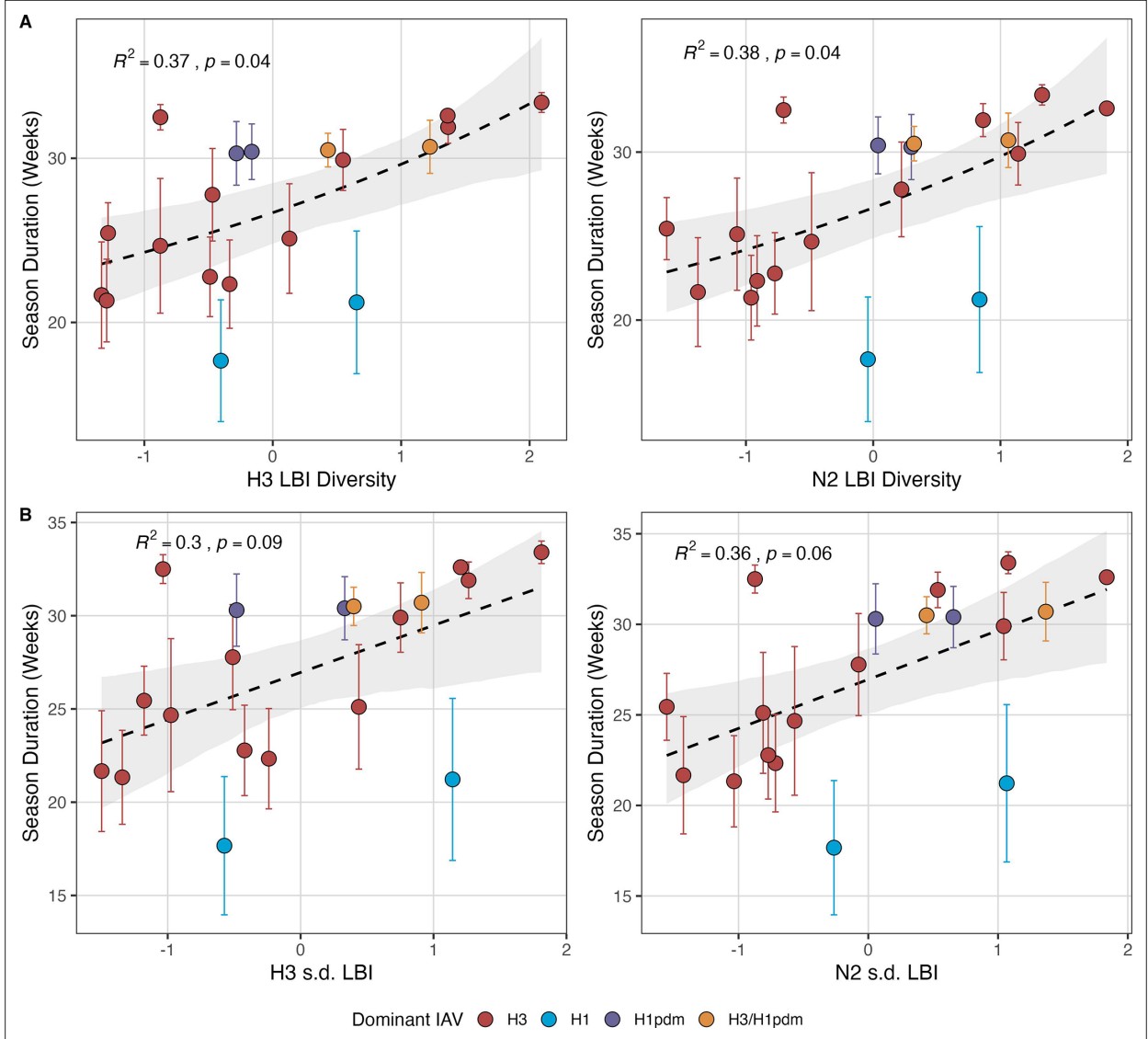

**Figure 5.** Influenza A(H3N2) seasonal duration increases with the diversity of hemagglutinin (H3) and neuraminidase (N2) clade growth rates in each season. Seasonal diversity of clade growth rates is measured as the (**A**) Shannon diversity or (**B**) standard deviation (s.d.) of H3 and N2 local branching index (LBI) values of viruses circulating in each season. LBI values are scaled to aid in direct comparisons of different LBI diversity metrics. Point color indicates the dominant influenza A subtype based on CDC influenza season summary reports (red: A(H3N2), blue: A(H1N1), purple: A(H1N1)pdm09, orange: A(H3N2)/A(H1N1)pdm09 co-dominant), and vertical bars are 95% confidence intervals of regional estimates (pre-2009 seasons: 9 regions; post-2009 seasons: 10 regions). Mean seasonal duration was fit as a function of H3 or N2 LBI diversity using Gaussian GLMs (inverse link) with 1000 bootstrap resamples. In each plot, the black dashed line represents the mean regression fit, and the gray shaded band shows the 95% confidence interval, based on 1000 bootstrap resamples. The R$^2$ and associated *p*-value from the mean regression fit are in the top left section of each plot.

The online version of this article includes the following figure supplement(s) for figure 5:

**Figure supplement 1.** Univariate correlations between influenza A(H3N2) evolutionary indicators and epidemic timing.

**Figure supplement 2.** Epidemic speed increases with N2 antigenic drift.

**Figure supplement 3.** Influenza A(H3N2) epidemic onsets and peaks are earlier in seasons with high antigenic novelty, but correlations are not statistically significant.

observed moderate-to-strong, non-linear relationships between A(H1N1) epidemic size and A(H3N2) epidemic size (R$^2$=0.65, p=0.01; *Figure 7*), peak incidence (R$^2$=0.66, p=0.02; *Figure 7*), and excess mortality (R$^2$=0.57, p=0.01; *Figure 7—figure supplement 1*), wherein A(H3N2) epidemic burden and excess mortality decreased as A(H1N1) incidence increased. A(H1N1) epidemic size was also significantly correlated with A(H3N2) transmissibility (effective $R_t$), exhibiting a negative, approximately

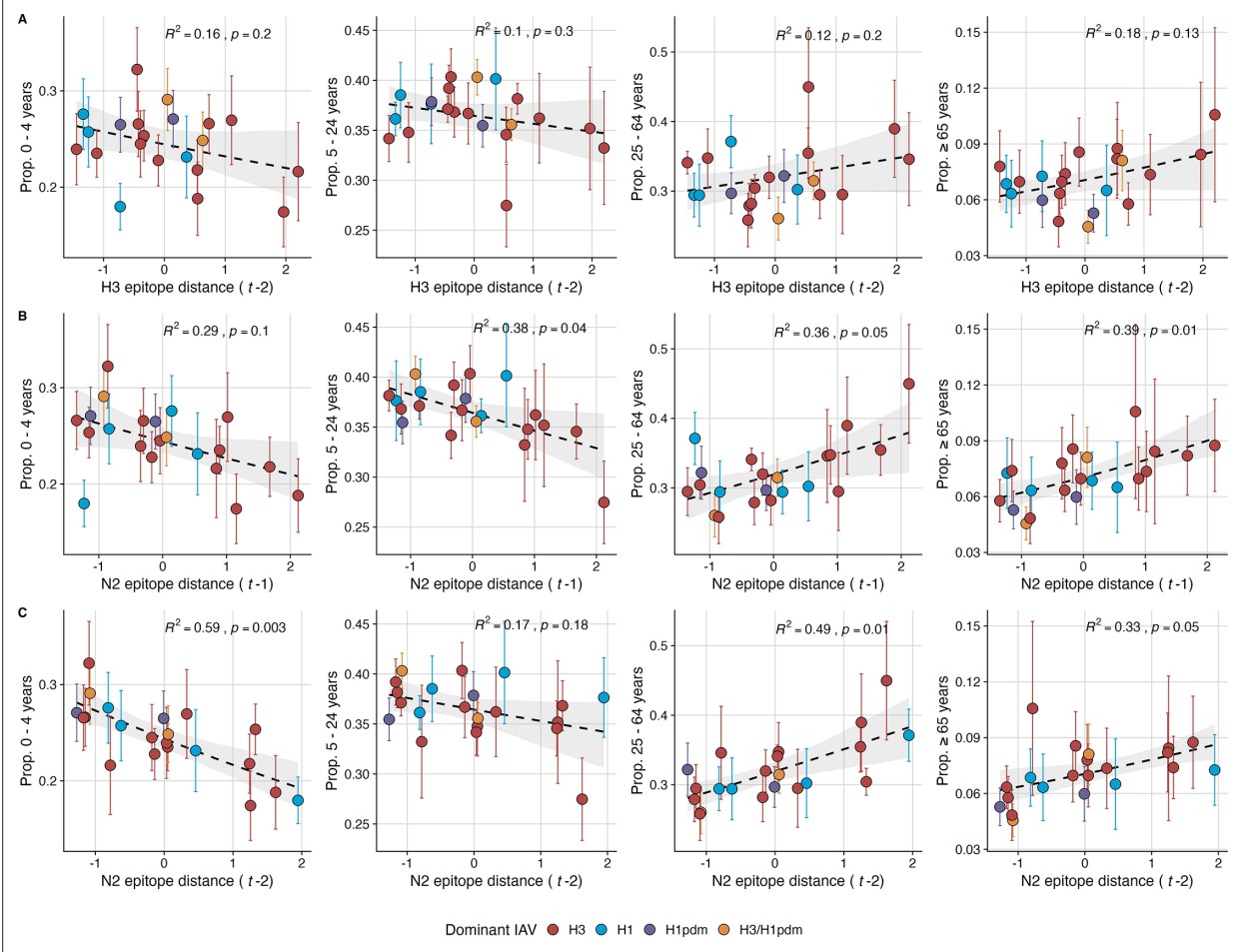

**Figure 6.** The proportion of outpatient influenza-like illness (ILI) cases in adults increases with neuraminidase (N2) antigenic novelty. N2 epitope distance, but not H3 epitope distance, significantly correlates with the age distribution of outpatient ILI cases. Seasonal epitope distance is the mean distance between viruses circulating in current season $t$ and viruses circulating in the prior season ($t − 1$) or two prior seasons ago ($t − 2$). Distances are scaled to aid in direct comparison of evolutionary indicators. Point color indicates the dominant influenza A subtype based on CDC influenza season summary reports (red: A(H3N2), blue: A(H1N1), purple: A(H1N1)pdm09, orange: A(H3N2)/A(H1N1)pdm09 co-dominant), and vertical bars are 95% confidence intervals of regional age distribution estimates (pre-2009 seasons: 9 regions; post-2009 seasons: 10 regions). The seasonal mean fraction of cases in each age group were fit as a function of H3 or N2 epitope distance using Beta GLMs (logit link) with 1000 bootstrap resamples. In each plot, the black dashed line represents the mean regression fit, and the gray shaded band shows the 95% confidence interval, based on 1000 bootstrap resamples. The $R^2$ and associated $p$-value from the mean regression fit are in the top right section of each plot.

The online version of this article includes the following figure supplement(s) for figure 6:

**Figure supplement 1.** Univariate correlations between A(H3N2) antigenic change and the age distribution of outpatient influenza-like illness (ILI) cases.

linear relationship ($R^2$=0.46, $p$=0.01; *Figure 7*). A(H3N2) epidemic intensity was negatively associated with A(H1N1) epidemic size, but this relationship was not statistically significant ($R^2$=0.21, $p$=0.15; *Figure 7*). Influenza B epidemic size was not significantly correlated with any A(H3N2) epidemic metrics (*Figure 7*, *Figure 7—figure supplement 1*).

The internal gene segments NS, M, NP, PA, and PB2 of A(H3N2) viruses and pre-2009 seasonal A(H1N1) viruses share a common ancestor (*Webster et al., 1992*) whereas A(H1N1)pdm09 viruses have a combination of gene segments derived from swine and avian reservoirs that were not reported prior to the 2009 pandemic (*Garten et al., 2009*; *Smith et al., 2009*). Non-glycoprotein genes are highly conserved between influenza A viruses and elicit cross-reactive antibody and T cell responses (*Grebe et al., 2008*; *Sridhar, 2016*). Because pre-2009 seasonal A(H1N1) viruses and A(H3N2) are more closely related, we hypothesized that seasonal A(H1N1) viruses could potentially limit the circulation of A(H3N2) viruses to a greater extent than A(H1N1)pdm09 viruses, due to greater T cell-mediated cross-protective immunity. As a sensitivity analysis, we measured correlations between

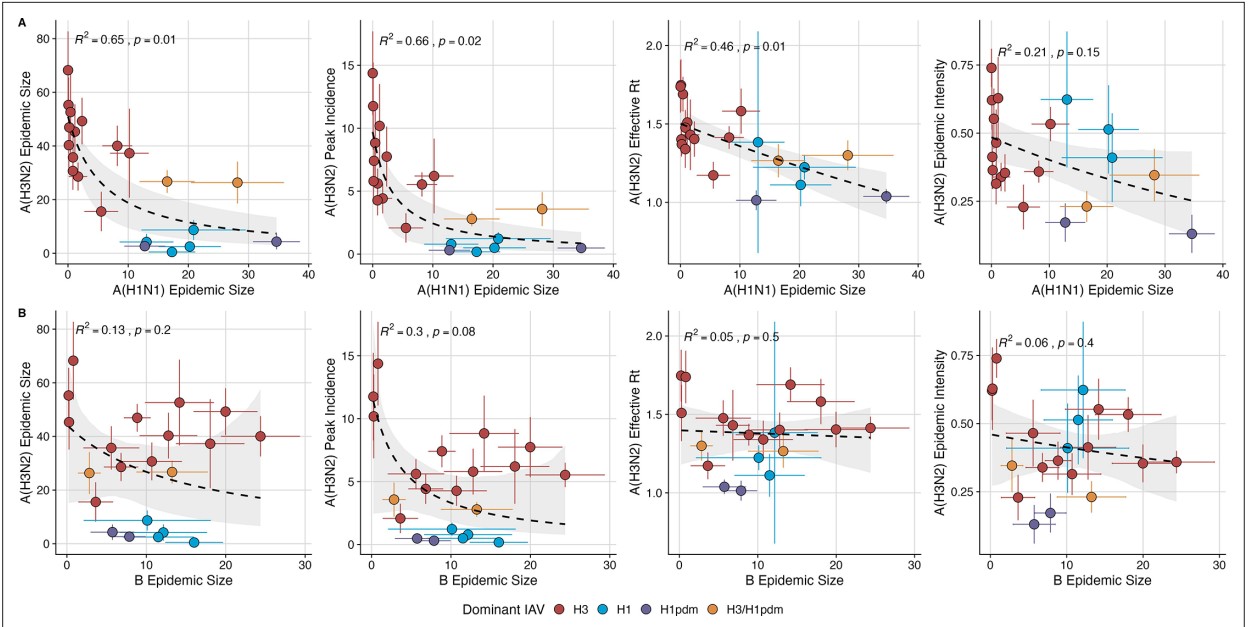

**Figure 7.** The effects of influenza A(H1N1) and B epidemic size on A(H3N2) epidemic burden. (**A**) Influenza A(H1N1) epidemic size negatively correlates with A(H3N2) epidemic size, peak incidence, transmissibility (effective reproduction number, $R_t$), and epidemic intensity. (**B**) Influenza B epidemic size does not significantly correlate with A(H3N2) epidemic metrics. Point color indicates the dominant influenza A virus (IAV) subtype based on CDC influenza season summary reports (red: A(H3N2), blue: A(H1N1), purple: A(H1N1)pdm09, orange: A(H3N2)/A(H1N1)pdm09 co-dominant), and vertical and horizontal bars are 95% confidence intervals of regional estimates (pre-2009 seasons: 9 regions; post-2009 seasons: 10 regions). Seasonal mean A(H3N2) epidemic metrics were fit as a function of mean A(H1N1) or B epidemic size using Gaussian GLMs (epidemic size and peak incidence: inverse link; effective $R_t$: log link) or Beta GLMs (epidemic intensity: logit link) with 1000 bootstrap resamples. In each plot, the black dashed line represents the mean regression fit, and the gray shaded band shows the 95% confidence interval, based on 1000 bootstrap resamples. The $R^2$ and associated *p*-value from the mean regression fit are in the top left section of each plot.

The online version of this article includes the following figure supplement(s) for figure 7:

**Figure supplement 1.** National excess influenza A(H3N2) mortality decreases with A(H1N1) epidemic size but not B epidemic size.

**Figure supplement 2.** The effect of influenza A(H1N1) epidemic size on A(H3N2) epidemic burden during the entire study period, pre-2009 seasons, and post-2009 seasons.

**Figure supplement 3.** Wavelet analysis of influenza A(H3N2), A(H1N1), and B epidemic timing.

A(H1N1) incidence and A(H3N2) epidemic metrics separately for pre- and post-2009 pandemic time periods. Relationships between different A(H3N2) epidemic metrics and A(H1N1) epidemic size were broadly similar for both periods, with slightly stronger correlations observed during the pre-2009 period (***Figure 7—figure supplement 2***).

We compared A(H3N2) epidemic timing across A(H3N2) and A(H1N1) dominant seasons, which we defined as when ≥70% of influenza A positive samples are typed as A(H3N2) or A(H1N1), respectively. A(H3N2) epidemic onsets and peaks occurred, on average, 3–4 weeks earlier in A(H3N2) dominant seasons (Wilcoxon test, *p*<0.0001; ***Table 3***). In A(H1N1) dominant seasons, regional A(H3N2) epidemics exhibited greater heterogeneity in epidemic timing (Wilcoxon tests, *p*<0.0001; ***Table 3***) and were shorter in duration compared to A(H3N2) dominant seasons (median duration: 21.5 weeks versus 28 weeks; Wilcoxon test, *p*<0.0001; ***Table 3***).

We applied a wavelet approach to weekly time series of incidences to measure more fine-scale differences in the relative timing of type/subtype circulation (***Figure 7—figure supplement 3***). A(H3N2) incidence preceded A(H1N1) incidence during most seasons prior to 2009 and during the two seasons in which A(H1N1)pdm09 was dominant, potentially because A(H3N2) viruses are more globally prevalent and migrate between regions more frequently than A(H1N1) viruses (***Bedford et al., 2015***). There was not a clear relationship between the direction of seasonal phase lags and A(H1N1) epidemic size ($R^2$=0.23, *p*=0.1; ***Figure 7—figure supplement 3***). A(H3N2) incidence led influenza B incidence in all influenza seasons (positive phase lag), irrespective of influenza B epidemic size ($R^2$=0.05, *p*=0.5; ***Figure 7—figure supplement 3***).

**Table 3.** Comparison of influenza A(H3N2) epidemic timing between A(H3N2) and A(H1N1) dominant seasons.
We used two-sided Wilcoxon rank-sum tests to compare the distributions of epidemic timing metrics between A(H3N2) and A(H1N1) dominant seasons. We categorized seasons as A(H3N2) or A(H1N1) dominant when ≥70% of IAV positive samples were typed as one IAV subtype.

| A(H3N2) timing metric | Dominant IAV subtype | | Wilcoxon test | |
| --- | --- | --- | --- | --- |
| | H3N2 | H1N1 | W | *p*-value |
| Median onset week (from EW40) | 8 | 11 | 3590 | $2.95 \times 10^{-7}$ |
| Median peak week (from EW40) | 17 | 20.5 | 5294.5 | $3.5 \times 10^{-9}$ |
| Regional variation (s.d.) in onset timing | 9.6 | 16.3 | 4095 | $1.61 \times 10^{-5}$ |
| Regional variation (s.d.) in peak timing | 12 | 22.6 | 6166 | $6.43 \times 10^{-18}$ |
| Seasonal duration | 28 | 21.5 | 1977.5 | $6.25 \times 10^{-6}$ |

Abbreviations: IAV, influenza A virus; EW40, epidemic week 40 (the start of the influenza season); s.d., standard deviation.

## The relative impacts of viral evolution, heterosubtypic interference, and prior immunity on A(H3N2) epidemic dynamics

We implemented conditional inference random forest models to assess the relative importance of viral evolution, type/subtype co-circulation, prior population immunity, and vaccine-related parameters in predicting regional A(H3N2) epidemic metrics (*Figure 8*).

Based on variable importance scores, A(H1N1) epidemic size in the current season was the most informative predictor of A(H3N2) epidemic size and peak incidence, followed by H3 epitope distance ($t - 2$) and the dominant IAV subtype in the previous season or N2 epitope distance ($t - 1$) (*Figure 8*). For A(H3N2) subtype dominance, the highest ranked predictors were N2 epitope distance ($t - 1$), the dominant IAV subtype in the previous season, and H3 epitope distance ($t - 2$) (*Figure 8*). We note that we did not include A(H1N1) epidemic size as a predictor in this model, due to its confounding with the target outcome. For models of A(H3N2) transmissibility (effective $R_t$) and epidemic intensity, we observed less discernable differences in variable importance scores across the set of candidate predictors (*Figure 8*). For the model of effective $R_t$, A(H1N1) epidemic size in the current season, adult vaccination coverage in the previous season, and N2 epitope distance between circulating viruses and the vaccine strain were the highest ranked variables, while the most important predictors of epidemic intensity were vaccination coverage in the previous season, N2 epitope distance between circulating viruses and the vaccine strain, and N2 epitope distance ($t - 1$). Variable importance rankings from LASSO models were qualitatively similar to those from random forest models, with A(H1N1) epidemic size in the current season, H3 and N2 epitope distance, and the dominant IAV subtype in the previous season consistently retained across the best-tuned models of epidemic size, peak incidence, and subtype dominance (*Figure 8—figure supplement 1*). Vaccine-related parameters and H3 antigenic drift (either H3 epitope distance or HI $\log_2$ titer distance) were retained in the best-tuned LASSO models of effective $R_t$ and epidemic intensity (*Figure 8—figure supplement 1*).

We measured correlations between observed values and model-predicted values at the HHS region level. Among the various epidemic metrics, random forest models produced the most accurate predictions of A(H3N2) subtype dominance (Spearman's $\rho$=0.95, regional range = 0.85–0.97), peak incidence ($\rho$=0.91, regional range = 0.72–0.95), and epidemic size ($\rho$=0.9, regional range = 0.74–0.95), while predictions of effective $R_t$ and epidemic intensity were less accurate ($\rho$=0.81, regional range = 0.65–0.91; $\rho$=0.78, regional range = 0.63–0.92, respectively) (*Figure 9*). Random forest models tended to underpredict most epidemic targets in seasons with substantial H3 antigenic transitions, in particular the SY97 cluster seasons (1998–1999, 1999–2000) and the FU02 cluster season (2003–2004) (*Figure 9*).

For epidemic size and peak incidence, seasonal predictive error – the root-mean-square error (RMSE) across all regional predictions in a season – increased with H3 epitope distance (epidemic size, Spearman's $\rho$=0.51, p=0.02; peak incidence, $\rho$=0.63, $p$=0.004) and N2 epitope distance (epidemic

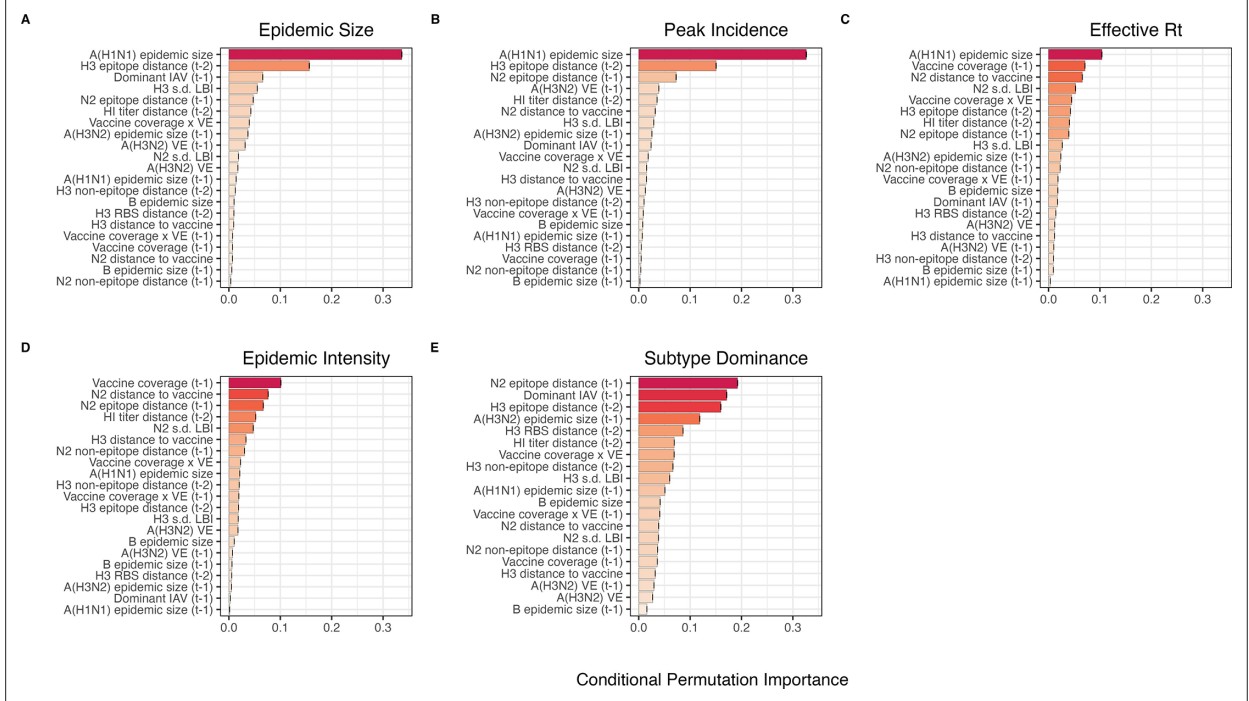

**Figure 8.** Variable importance rankings from conditional inference random forest models predicting seasonal region-specific influenza A(H3N2) epidemic dynamics. Ranking of variables in predicting regional A(H3N2) (**A**) epidemic size, (**B**) peak incidence, (**C**) transmissibility (maximum effective reproduction number, $R_t$), (**D**) epidemic intensity, and (**E**) subtype dominance. Each forest was created by generating 3000 regression trees from a repeated leave-one-season-out cross-validated sample of the data. Variables are ranked by their conditional permutation importance, with differences in prediction accuracy scaled by the total (null model) error. Black error bars are 95% confidence intervals of conditional permutation scores (N=50 permutations). Abbreviations: $t – 1$, one-season lag; $t – 2$, two-season lag; IAV, influenza A virus subtype; s.d., standard deviation; HI, hemagglutination inhibition; LBI, local branching index; distance to vaccine, epitope distance between currently circulating viruses and the recommended vaccine strain; VE, vaccine effectiveness.

The online version of this article includes the following figure supplement(s) for figure 8:

**Figure supplement 1.** Variable importance rankings from LASSO regression models predicting seasonal region-specific influenza A(H3N2) epidemic dynamics.

size, $\rho$=0.48, p=0.04; peak incidence, $\rho$=0.48, p=0.03) (**Figure 9—figure supplements 1 and 2**). For models of epidemic intensity, seasonal RMSE increased with N2 epitope distance ($\rho$=0.64, p=0.004) but not H3 epitope distance ($\rho$=0.06, p=0.8) (**Figure 9—figure supplements 1 and 2**). Seasonal RMSE of effective $R_t$ and subtype dominance predictions did not correlate with H3 or N2 epitope distance (**Figure 9—figure supplements 1 and 2**).

To further refine our set of informative predictors, we performed multivariable regression with the top 10 ranked predictors from each random forest model and used BIC to select the best fit model for each epidemic metric, allowing each metric's regression model to include up to three independent variables. This additional step of variable selection demonstrated that models with few predictors fit the observed data relatively well (epidemic size, adjusted $R^2$=0.69; peak incidence, $R^2$=0.63; effective $R_t$, $R^2$=0.63; epidemic intensity, $R^2$=0.75), except for subtype dominance ($R^2$=0.48) (**Table 4**). The set of variables retained after model selection were similar to those with high importance rankings in random forest models and LASSO regression models, with the exception that HI $\log_2$ titer distance, rather than H3 epitope distance, was included in the minimal models of effective $R_t$ and epidemic intensity.

## Discussion

Antigenic drift between currently circulating influenza viruses and the previous season's viruses is expected to confer increased viral fitness, leading to earlier, larger, or more severe epidemics. However, prior evidence for the impact of antigenic drift on seasonal influenza outbreaks is mixed.

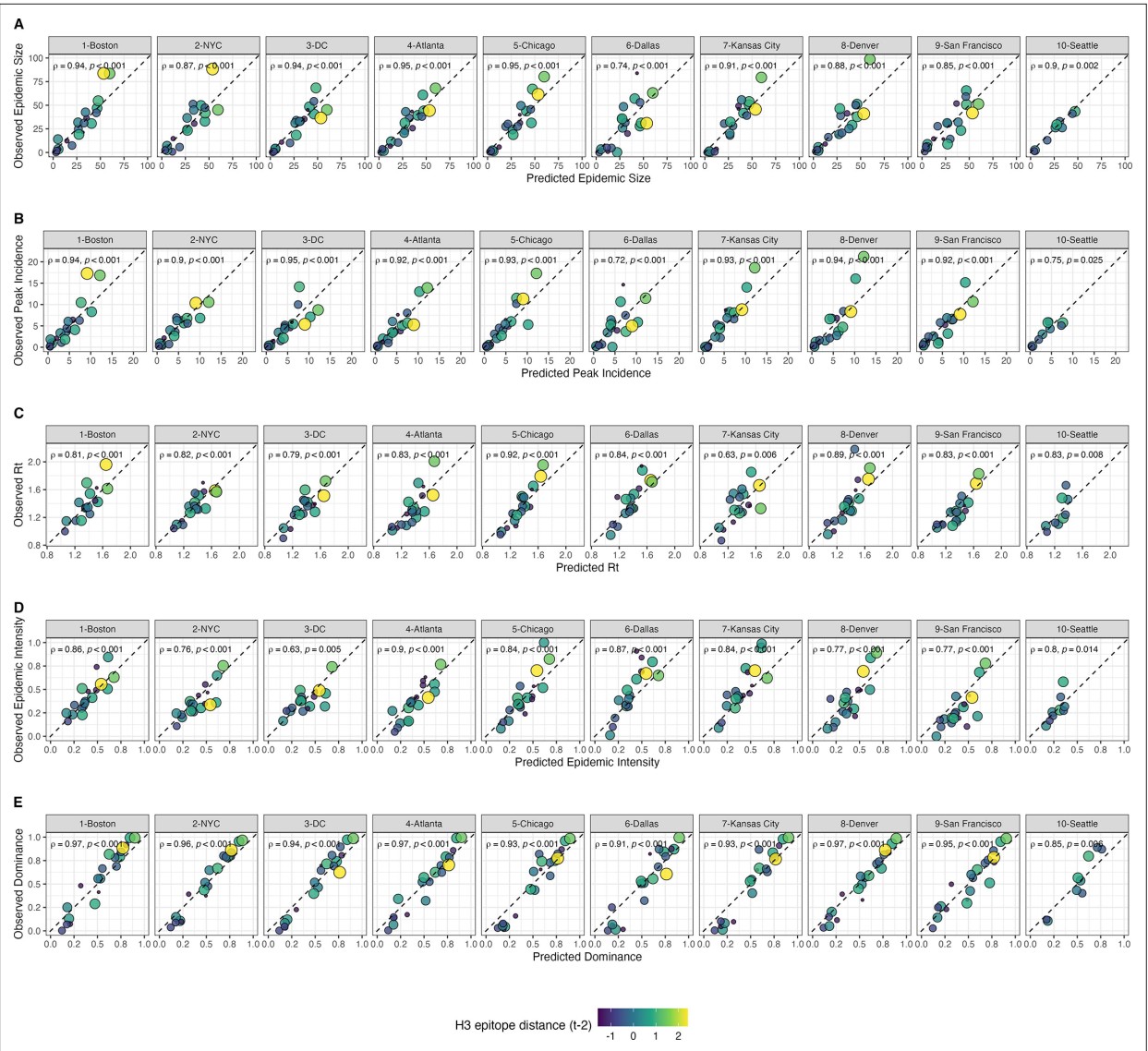

**Figure 9.** Observed versus predicted values of seasonal region-specific influenza A(H3N2) epidemic metrics from conditional inference random forest models. (**A**) Epidemic size, (**B**) peak incidence, (**C**) transmissibility (maximum effective reproduction number, $R_t$), (**D**) epidemic intensity, and (**E**) subtype dominance. Results are facetted by HHS region and epidemic metric. Point color and size corresponds to the mean H3 epitope distance between viruses circulating in the current season $t$ and viruses circulating two prior seasons ago ($t − 2$). Large, yellow points indicate seasons with high antigenic novelty, and small blue points indicate seasons with low antigenic novelty. In each facet, the Spearman's rank correlation coefficient and associated $p$-value are in the top left section, and the black dashed line shows $y = x$.

The online version of this article includes the following figure supplement(s) for figure 9:

**Figure supplement 1.** Relationships between the predictive accuracy of random forest models and seasonal H3 epitope distance.

**Figure supplement 2.** Relationships between the predictive accuracy of random forest models and seasonal N2 epitope distance.

Here, we systematically compare experimental and sequence-based measures of A(H3N2) evolution in predicting regional epidemic dynamics in the United States across 22 seasons, from 1997 to 2019. We also consider the effects of other co-circulating influenza viruses, prior immunity, and vaccine-related parameters, including vaccination coverage and effectiveness, on A(H3N2) incidence. Our findings indicate that evolution in both major surface proteins – hemagglutinin (HA) and neuraminidase (NA) – contributes to variability in epidemic magnitude across seasons, though viral fitness appears to be secondary to subtype interference in shaping annual outbreaks.

The first question of this study sought to determine which metrics of viral fitness have the strongest relationships with A(H3N2) epidemic burden and timing. Among our set of candidate evolutionary

**Table 4.** Predictors of seasonal A(H3N2) epidemic size, peak incidence, transmissibility, epidemic intensity, and subtype dominance.

Variables retained in the best fit model for each epidemic outcome were determined by BIC.

| Outcome | Best Minimal Model[1] | $R^2$ | Adj. $R^2$ | RMSE |
|---|---|---|---|---|
| Epidemic Size | H3 epitope distance $(t-2)$ + <br> H1 epidemic size + <br> H3 epidemic size $(t-1)$ | 0.74 | 0.69 | 9.88 |
| Peak Incidence | H3 epitope distance $(t-2)$ + <br> H1 epidemic size + <br> Dominant IAV Subtype $(t-1)$ | 0.69 | 0.63 | 2.09 |
| Effective $R_t$ | HI $\log_2$ titer distance $(t-2)$ + <br> H1 epidemic size + <br> N2 distance to vaccine strain | 0.69 | 0.63 | 0.11 |
| Epidemic Intensity | HI $\log_2$ titer distance $(t-2)$ + <br> N2 distance to vaccine strain + <br> vaccination coverage $(t-1)$ | 0.79 | 0.75 | 0.07 |
| Subtype Dominance | H3 epitope distance $(t-2)$ + <br> N2 epitope distance $(t-1)$ + <br> Dominant IAV Subtype $(t-1)$ | 0.56 | 0.48 | 0.2 |

[1]Candidate models were limited to three independent variables and considered all combinations of the top 10 ranked predictors from conditional inference random forest models (**Figure 8**).

predictors, genetic distances based on broad sets of epitope sites (HA = 129 sites; NA = 223 epitope sites) had the strongest, most consistent associations with A(H3N2) epidemic size, transmission rate, severity, subtype dominance, and age-specific patterns. Increased epitope distance in both H3 and N2 correlated with larger epidemics and increased transmissibility, with univariate analyses finding H3 distance more strongly correlated with epidemic size, peak incidence, transmissibility, and excess mortality, and N2 distance more strongly correlated with epidemic intensity (i.e. the 'sharpness' of the epidemic curve) and subtype dominance patterns. However, we note that minor differences in correlative strength between H3 and N2 epitope distance are not necessarily biologically relevant and could be attributed to noise in epidemiological or virological data or the limited number of influenza seasons in our study. The fraction of ILI cases in children relative to adults was negatively correlated with N2 epitope distance, consistent with the expectation that cases are more restricted to immunologically naive children in seasons with low antigenic novelty (**Bedford et al., 2015**; **Gostic et al., 2019**). Regarding epidemic timing, the number of days from epidemic onset to peak (a proxy for epidemic speed) decreased with N2 epitope distance, but other measures of epidemic timing, such as peak week, onset week, and spatiotemporal synchrony across HHS regions, were not significantly correlated with H3 or N2 antigenic change.

The local branching index (LBI) is traditionally used to predict the success of individual clades, with high LBI values indicating high viral fitness (**Huddleston et al., 2020**; **Neher et al., 2014**). In our epidemiological analysis, low diversity of H3 or N2 LBI in the current season correlated with greater epidemic intensity, higher transmission rates, and shorter seasonal duration. These associations suggest that low LBI diversity is indicative of a rapid selective sweep by one successful clade, while high LBI diversity is indicative of multiple co-circulating clades with variable seeding and establishment times over the course of an epidemic. A caveat is that LBI estimation is more sensitive to sequence sub-sampling schemes than strain-level measures. If an epidemic is short and intense (e.g. 1–2 months), a phylogenetic tree with our sub-sampling scheme (50 sequences per month) may not incorporate enough sequences to capture the true diversity of LBI values in that season.

Positive associations between H3 antigenic drift and population-level epidemic burden are consistent with previous observations from theoretical models (**Bedford et al., 2012**; **Koelle et al., 2006**; **Koelle et al., 2009**). For example, phylodynamic models of punctuated antigenic evolution have reproduced key features of A(H3N2) phylogenetic patterns and case dynamics, such as the sequential replacement of antigenic clusters, the limited standing diversity in HA after a cluster transition, and higher incidence and attack rates in cluster transition years (**Bedford et al., 2012**; **Koelle et al., 2006**; **Koelle et al., 2009**). Our results also corroborate empirical analyses of surveillance data

(*Bedford et al., 2014*; *Wilson and Cox, 1990*; *Wolf et al., 2010*; *Wu et al., 2010*) and forecasting models of annual epidemics (*Axelsen et al., 2014*; *Du et al., 2017*) that found direct, quantitative links between HA antigenic novelty and the number of influenza cases or deaths in a season. Moving beyond the paradigm of antigenic clusters, *Wolf et al., 2010* and *Bedford et al., 2014* demonstrated that smaller, year-to-year changes in H3 antigenic drift also correlate with seasonal severity and incidence (*Bedford et al., 2014*; *Wolf et al., 2010*). A more recent study did not detect an association between antigenic drift and city-level epidemic size in Australia (*Lam et al., 2020*), though the authors used a binary indicator to signify seasons with major HA antigenic transitions and did not consider smaller, more gradual changes in antigenicity. While Lam and colleagues did not observe a consistent effect of antigenic change on epidemic magnitude, they found a negative relationship between the cumulative prior incidence of an antigenic variant and its probability of successful epidemic initiation in a city.

We did not observe a clear relationship between H3 receptor binding site (RBS) distance and epidemic burden, even though single substitutions at these seven amino acid positions are implicated in major antigenic transitions (*Koel et al., 2013*; *Petrova and Russell, 2018*). The outperformance of the RBS distance metric by a broader set of epitope sites could be attributed to the tempo of antigenic cluster changes. A(H3N2) viruses are characterized by both continuous and punctuated antigenic evolution, with transitions between antigenic clusters occurring every 2–8 years (*Bedford et al., 2011*; *Bedford et al., 2014*; *Koel et al., 2013*; *Koelle et al., 2006*; *Koelle and Rasmussen, 2015*; *Shih et al., 2007*; *Smith et al., 2004*; *Suzuki, 2008*; *Wolf et al., 2006*). Counting substitutions at only a few sites may fail to capture more modest, gradual changes in antigenicity that are on a time scale congruent with annual outbreaks. Further, a broader set of epitope sites may better capture the epistatic interactions that underpin antigenic change in HA (*Kryazhimskiy et al., 2011*). Although the seven RBS sites were responsible for the majority of antigenic phenotype in Koel and colleagues' experimental study (*Koel et al., 2013*), their findings do not necessarily contradict studies that found broader sets of sites associated with antigenic change. Mutations at other epitope sites may collectively add to the decreased recognition of antibodies or affect viral fitness through alternate mechanisms (e.g. compensatory or permissive mutations) (*Gong et al., 2013*; *Koel et al., 2013*; *Koelle et al., 2006*; *Kryazhimskiy et al., 2011*; *Myers et al., 2013*; *Neher et al., 2014*; *Shih et al., 2007*; *Smith et al., 2004*).

A key result from our study is the direct link between NA antigenic drift and A(H3N2) incidence patterns. Although HA and NA both contribute to antigenicity (*Nelson and Holmes, 2007*; *Webster et al., 1982*) and undergo similar rates of positive selection (*Bhatt et al., 2011*), we expected antigenic change in HA to exhibit stronger associations with seasonal incidence, given its immunodominance relative to NA (*Altman et al., 2015*). H3 and N2 epitope distance were both moderately correlated with epidemic size, peak incidence, and subtype dominance patterns, but, except for subtype dominance, H3 epitope distance had higher variable importance rankings in random forest models and N2 epitope distance was not retained after post-hoc model selection of top ranked random forest features. However, N2 epitope distance but not H3 epitope distance was associated with faster epidemic speed and a greater fraction of ILI cases in adults relative to children. Antigenic changes in H3 and N2 were independent across the 22 seasons of our study, consistent with previous research (*Bhatt et al., 2011*; *Sandbulte et al., 2011*; *Schulman and Kilbourne, 1969*). Thus, the similar predictive performance of HA and NA epitope distance for some epidemic metrics does not necessarily stem from the coevolution of HA and NA.

HI $\log_2$ titer distance was positively correlated with different measures of epidemic impact yet underperformed in comparison to H3 and N2 epitope distances. This outcome was surprising given that we expected our method for generating titer distances would produce more realistic estimates of immune cross-protection between viruses than epitope-based measures. Our computational approach for inferring HI phenotype dynamically incorporates newer titer measurements and assigns antigenic weight to phylogenetic branches rather than fixed sequence positions (*Huddleston et al., 2020*; *Neher et al., 2016*). In contrast, our method for calculating epitope distance assumes that the contributions of specific sites to antigenic drift are constant through time, even though beneficial mutations previously observed at these sites are contingent on historical patterns of viral fitness and host immunity (*Huddleston et al., 2020*; *Koelle et al., 2006*; *Neher et al., 2014*). HI titer measurements have been more useful than epitope substitutions in predicting future A(H3N2) viral populations

(*Huddleston et al., 2020*) and vaccine effectiveness (*Ndifon et al., 2009*), with the caveat that these targets are more proximate to viral evolution than epidemic dynamics.

HI titer measurements may be more immunologically relevant than epitope-based measures, yet several factors could explain why substitutions at epitope sites outperformed HI titer distances in epidemiological predictions. First, epitope distances may capture properties that affect viral fitness (and in turn outbreak intensity) but are unrelated to immune escape, such as intrinsic transmissibility, ability to replicate, or epistatic interactions. A second set of factors concern methodological issues associated with HI assays. The reference anti-sera for HI assays are routinely produced in ferrets recovering from their first influenza virus infection. Most humans are infected by different influenza virus strains over the course of their lifetimes, and one's immune history influences the specificity of antibodies generated against drifted influenza virus strains (*Hensley, 2014*; *Lee et al., 2019*; *Li et al., 2013*; *Miller et al., 2013*). Thus, human influenza virus antibodies, especially those of adults, have more heterogeneous specificities than anti-sera from immunologically naive ferrets (*Hensley, 2014*).

A related methodological issue is that HI assays disproportionately measure anti-HA antibodies that bind near the receptor binding site and, similar to the RBS distance metric, may capture only a partial view of the antigenic change occurring in the HA protein (*Gostic et al., 2019*; *Henry et al., 2019*; *Lam et al., 2020*; *Ranjeva et al., 2019*). A recent study of longitudinal serological data found that HI titers are a good correlate of protective immunity for children, while time since infection is a better predictor of protection for adults (*Ranjeva et al., 2019*). This outcome is consistent with the concept of antigenic seniority, in which an individual's first exposure to influenza virus during child-hood leaves an immunological 'imprint', and exposure to new strains 'back boosts' one's antibody response to strains of the same subtype encountered earlier in life (*Cobey and Hensley, 2017*; *Gostic et al., 2019*; *Zhang et al., 2019*). Ranjeva et al.'s study and others suggest that human influenza virus antibodies shift focus from the HA head to other more conserved epitopes as individuals age (*Gostic et al., 2019*; *Henry et al., 2019*). Given that HI assays primarily target epitopes adjacent to the RBS, HI assays using ferret or human serological data are not necessarily suitable for detecting the broader immune responses of adults. A third explanation for the underperformance of HI titers concerns measurement error. Recent A(H3N2) viruses have reduced binding efficiency in HI assays, which can skew estimates of immune cross-reactivity between viruses (*Zost et al., 2017*). These combined factors could obfuscate the relationship between the antigenic phenotypes inferred from HI assays and population-level estimates of A(H3N2) incidence.

Novel antigenic variants are expected to have higher infectivity in immune populations, leading to earlier epidemics and more rapid geographic spread (*Viboud et al., 2006*), but few studies have quantitatively linked antigenic drift to epidemic timing or geographic synchrony. Previous studies of pneumonia and influenza-associated mortality observed greater severity or geographic synchrony in seasons with major antigenic transitions (*Greene et al., 2006*; *Wiley et al., 1981*). A more recent Australian study of lab-confirmed cases also noted greater spatiotemporal synchrony during seasons when novel H3 antigenic variants emerged, although their assessment was based on virus typing alone (i.e. influenza A or B; *Geoghegan et al., 2018*). A subsequent Australian study with finer-resolution data on subtype incidence and variant circulation determined that more synchronous epidemics were not associated with drifted A(H3N2) strains (*Lam et al., 2020*), and a U.S.-based analysis of ILI data also failed to detect a relationship between HA antigenic cluster transitions and geographic synchrony (*Charu et al., 2017*). In our study, the earliest epidemics tended to occur in seasons with transitions between H3 antigenic clusters (e.g. the emergence of the FU02 cluster in 2003–2004) or vaccine mismatches (e.g. N2 mismatch in 1999–2000, H3 mismatch in 2014–2015; *Sandbulte et al., 2011*; *Smith et al., 2004*; *Xie et al., 2015*), but there was not a statistically significant correlation between antigenic change and earlier epidemic onsets or peaks. Regarding epidemic speed, the length of time from epidemic onset to peak decreased with N2 epitope distance but not H3 epitope distance. The relationship between antigenic drift and epidemic timing may be ambiguous because external seeding events or climatic factors, such as temperature and absolute humidity, are more important in driving influenza seasonality and the onsets of winter epidemics (*Bedford et al., 2015*; *Charu et al., 2017*; *Chattopadhyay et al., 2018*; *Kramer and Shaman, 2019*; *Lee et al., 2018*; *Shaman and Kohn, 2009*; *Shaman et al., 2010*). Alternatively, the resolution of our epidemiological surveillance data (HHS regions) may not be granular enough to detect a signature of antigenic drift in epidemic timing,

though studies of city-level influenza dynamics were also unable to identify a clear relationship (*Charu et al., 2017*; *Lam et al., 2020*).

After exploring individual correlations between evolutionary indicators and annual epidemics, we considered the effects of influenza A(H1N1) incidence and B incidence on A(H3N2) virus circulation within a season. We detected strong negative associations between A(H1N1) incidence and A(H3N2) epidemic size, peak incidence, transmissibility, and excess mortality, consistent with previous animal, epidemiological, phylodynamic, and theoretical studies that found evidence for cross-immunity between IAV subtypes (*Cowling et al., 2010*; *Epstein, 2006*; *Ferguson et al., 2003*; *Gatti et al., 2022*; *Goldstein et al., 2011*; *Sonoguchi et al., 1985*). For example, individuals recently infected with seasonal influenza A viruses are less likely to become infected during subsequent pandemic waves (*Cowling et al., 2010*; *Epstein, 2006*; *Fox et al., 2017*; *Laurie et al., 2015*; *Sridhar et al., 2013*), and the early circulation of one influenza virus type or subtype is associated with a reduced total incidence of the other type/subtypes within a season (*Goldstein et al., 2011*; *Lam et al., 2020*). Due to the shared evolutionary history of their internal genes (*Webster et al., 1992*) and in turn greater T cell-mediated cross-protective immunity, pre-2009 seasonal A(H1N1) viruses may impact A(H3N2) virus circulation to a greater extent than A(H1N1)pdm09 viruses, which have a unique combination of genes that were not identified in animals or humans prior to 2009 (*Garten et al., 2009*; *Smith et al., 2009*). We observed similar relationships between A(H3N2) epidemic metrics and A(H1N1) incidence during pre- and post-2009 pandemic seasons, with slightly stronger correlations observed during the pre-2009 period. However, given the small sample size (12 pre-2009 seasons and 9 post-2009 seasons), we cannot fully answer this question.

In our study, univariate correlations between A(H1N1) and A(H3N2) incidence were more pronounced than those observed between A(H3N2) incidence and evolutionary indicators, and A(H1N1) epidemic size was the highest ranked feature by random forest models predicting epidemic size, peak incidence, and transmissibility (effective $R_t$). Consequently, interference between the two influenza A subtypes may be more impactful than viral evolution in determining the size of annual A(H3N2) outbreaks. Concerning epidemic timing, we did not detect a relationship between A(H3N2) antigenic change and the relative timing of A(H3N2) and A(H1N1) cases; specifically, A(H3N2) incidence did not consistently lead A(H1N1) incidence in seasons with greater H3 or N2 antigenic change. Overall, we did not find any indication that influenza B incidence affects A(H3N2) epidemic burden or timing, which is not unexpected, given that few T and B cell epitopes are shared between the two virus types (*Terajima et al., 2013*).

Lastly, we used random forest models and multivariable linear regression models to assess the relative importance of viral evolution, prior population immunity, co-circulation of other influenza viruses, and vaccine-related parameters in predicting regional A(H3N2) epidemic dynamics. We chose conditional inference random forest models as our primary method of variable selection because several covariates were collinear, relationships between some predictors and target variables were nonlinear, and our goal was inferential rather than predictive. We performed leave-one-season-out cross-validation to tune each model, but, due to the limited number of seasons in our dataset, we were not able to test predictive performance on an independent test set. With the caveat that models were likely overfit to historical data, random forest models produced accurate predictions of regional epidemic size, peak incidence, and subtype dominance patterns, while predictions of epidemic intensity and transmission rates were less exact. The latter two measures could be more closely tied to climatic factors, the timing of influenza case importations from abroad, or mobility patterns (*Bedford et al., 2015*; *Charu et al., 2017*; *Shaman and Kohn, 2009*; *Shaman et al., 2010*) or they may be inherently more difficult to predict because their values are more constrained. Random forest models tended to underpredict epidemic burden in seasons with major antigenic transitions, particularly the SY97 seasons (1998–1999, 1999–2000) and the FU02 season (2003–2004), potentially because antigenic jumps of these magnitudes were infrequent during our 22-season study period. An additional step of post-hoc model selection demonstrated that models with only three covariates could also produce accurate fits to observed epidemiological data.

Our study is subject to several limitations, specifically regarding geographic resolution and data availability. First, our analysis is limited to one country with a temperate climate and its findings concerning interactions between A(H3N2), A(H1N1), and type B viruses may not be applicable to tropical or subtropical countries, which experience sporadic epidemics of all three viruses throughout

the year (*Yang et al., 2020*). Second, our measure of population-level influenza incidence is derived from regional CDC outpatient data because those data are publicly available starting with the 1997–1998 season. State level outpatient data are not available until after the 2009 A(H1N1) pandemic, and finer resolution data from electronic health records are accessible in theory but not in the public domain. Access to ILI cases aggregated at the state or city level, collected over the course of decades, would increase statistical power, and enable us to add more location-specific variables to our analysis, such as climatic and environmental factors. A third limitation is that we measured influenza incidence by multiplying the rate of influenza-like illness by the percentage of tests positive for influenza, which does not completely eliminate the possibility of capturing the activity of other co-circulating respiratory pathogens (*Kramer and Shaman, 2019*). Surveillance data based on more specific diagnosis codes would ensure the exclusion of patients with non-influenza respiratory conditions. Fourth, our data on the age distribution of influenza cases are derived from ILI encounters across four broad age groups and do not include test positivity status, virus type/subtype, or denominator information. Despite the coarseness of these data, we found statistically significant correlations in the expected directions between N2 antigenic change and the fraction of cases in children relative to adults. Lastly, a serological assay exists for NA, but NA titer measurements are not widely available because the assay is labor-intensive and inter-lab variability is high. Thus, we could not test the performance of NA antigenic phenotype in predicting epidemic dynamics.

Beginning in early 2020, non-pharmaceutical interventions (NPIs), including lockdowns, school closures, physical distancing, and masking, were implemented in the United States and globally to slow the spread of severe acute respiratory syndrome coronavirus 2 (SARS-CoV-2), the virus responsible for the COVID-19 pandemic. These mitigation measures disrupted the transmission of seasonal influenza viruses and other directly-transmitted respiratory viruses throughout 2020 and 2021 (*Cowling et al., 2020*; *Huang et al., 2021*; *Olsen et al., 2020*; *Olsen et al., 2021*; *Qi et al., 2021*; *Tempia et al., 2021*), and population immunity to influenza is expected to have decreased substantially during this period of low circulation (*Ali et al., 2022*; *Baker et al., 2020*). COVID-19 NPIs relaxed during 2021 and 2022 and co-circulation of A(H3N2) and A(H1N1)pdm09 viruses in the United States resumed during the 2022–2023 influenza season. Our study concludes with the 2018–2019 season, and thus it is unclear whether our modeling approach would be useful in projecting seasonal burden during the post-pandemic period, without an additional component to account for COVID-19-related perturbations to influenza transmission. Further studies will need to determine whether ecological interactions between influenza viruses have changed or if the effects of viral evolution and subtype interference on seasonal outbreaks are different in the post-pandemic period.

In conclusion, relationships between A(H3N2) antigenic drift, epidemic impact, and age dynamics are moderate, with genetic distances based on broad sets of H3 and N2 epitope sites having greater predictive power than serology-based antigenic distances for the timeframe analyzed. Influenza epidemiological patterns are consistent with increased population susceptibility in seasons with high antigenic novelty, and our study is the first to link NA antigenic drift to epidemic burden, timing, and the age distribution of cases. It is well established that anti-HA and anti-NA antibodies are independent correlates of immunity (*Couch et al., 2013*; *Gaglani et al., 2016*; *Gill and Murphy, 1977*; *Hope-Simpson, 1971*; *Memoli et al., 2016*; *Monto et al., 2015*; *Murphy et al., 1972*), and the influenza research community has advocated for NA-based vaccines (*Eichelberger et al., 2018*; *Krammer et al., 2018*). The connection between NA drift and seasonal incidence further highlights the importance of monitoring evolution in both HA and NA to inform vaccine strain selection and epidemic forecasting efforts. Although antigenic change in both HA and NA was correlated with epidemic dynamics, ecological interactions between influenza A subtypes appear to be more influential than viral evolution in determining the intensity of annual A(H3N2) epidemics. The aim of our study was to retrospectively assess the potential drivers of annual A(H3N2) epidemics, yet we cautiously suggest that one could project the size or intensity of future epidemics based on sequence data and A(H1N1)pdm09 incidence alone (*Goldstein et al., 2011*; *Wolf et al., 2010*).

## Acknowledgements

We thank the Influenza Division at the U.S. Centers for Disease Control and Prevention, the Victorian Infectious Diseases Reference Laboratory at the Australian Peter Doherty Institute for Infection and Immunity, the Influenza Virus Research Center at the Japan National Institute of Infectious Diseases, the

Crick Worldwide Influenza Centre at the UK Francis Crick Institute for sharing HI titer data. We gratefully acknowledge the authors, originating laboratories, and submitting laboratories of the sequences from the GISAID EpiFlu Database on which this research is based (listed in Appendix 1). We thank members of the Fogarty International Center's Division of International Epidemiology and Population Studies (DIEPS) and the Bedford Lab for useful discussions. ACP, CLH, and CV were supported by the in-house research division of the Fogarty International Center, U.S. National Institutes of Health. ACP was supported by the NSF Infectious Disease Evolution Across Scales (IDEAS) Research Collaboration Network. JH was supported by NIH NIAID awards F31 AI140714 and R01 AI165821. The work done at the Crick Worldwide Influenza Centre was supported by the Francis Crick Institute receiving core funding from Cancer Research UK (FC001030), the Medical Research Council (FC001030) and the Wellcome Trust (FC001030). SF, KN, NK, SW and HH were supported by the Ministry of Health, Labour and Welfare, Japan (10110400 and 10111800). SW was supported by the Japan Agency for Medical Research and Development (JP22fk0108118 and JP23fk0108662). The WHO Collaborating Centre for Reference and Research on Influenza is supported by the Australia Government Department of Health and Aged Care. The Melbourne WHO Collaborating Centre for Reference and Research on Influenza is supported by the Australian Government Department of Health. Influenza virus work in the Krammer laboratory was partially supported by the NIAID Centers of Excellence for Influenza Research and Surveillance (CEIRS) contract HHSN272201400008C, NIAID Centers of Excellence for Influenza Research and Response (CEIRR) contract 75N93021C00014 (FK), and NIAID CIVIC contract (75N93019C00051). TB was supported by NIH awards NIGMS R35 GM119774 and NIAID R01 AI127893. TB is an Investigator of the Howard Hughes Medical Institute. Funding sources were not involved in study design, data collection and interpretation, or the decision to submit the work for publication.

## Additional information

### Competing interests

Chelsea L Hansen: Received personal fees from Sanofi outside the submitted work. John W McCauley: Received consulting fees, honoraria, and travel support from Sanofi Pasteur and Sequris. Sheena G Sullivan: The WHO Collaborating Centre for Reference and Research on Influenza in Melbourne has a collaborative research and development agreement (CRADA) with CSL Seqirus for isolation of candidate vaccine viruses in cells and an agreement with IFPMA for isolation of candidate vaccine viruses in eggs. SGS reports honoraria from CSL Seqirus, Moderna, Pfizer, and Evo Health. Ian G Barr, Kanta Subbarao: The WHO Collaborating Centre for Reference and Research on Influenza in Melbourne has a collaborative research and development agreement (CRADA) with CSL Seqirus for isolation of candidate vaccine viruses in cells and an agreement with IFPMA for isolation of candidate vaccine viruses in eggs. Florian Krammer: The Icahn School of Medicine at Mount Sinai has filed patent applications relating to influenza virus vaccines (U.S. patent numbers: 12030928, 11865173, 11266734, 11254733, 10736956, 10583188, 10137189, 10131695, 9968670, 9371366; publication numbers: 20230181715, 20220403358, 20220249652, 20220242935, 20220153873, 20210260179, 20190125859, 20190106461, 20180333479), SARS-CoV-2 serological assays (publication number: 20240210415), and SARS-CoV-2 vaccines (publication numbers: 20230310583, 20230226171), which list FK as co-inventor. FK has consulted for Merck and Pfizer (before 2020), and is currently consulting for Pfizer, Seqirus, 3rd Rock Ventures, GSK and Avimex. The Krammer laboratory is also collaborating with Pfizer on animal models of SARS-CoV-2 and with Dynavax on universal influenza virus vaccines. Cécile Viboud: Received honoraria for serving as an Editor in Chief of the journal Epidemics (Elsevier). The other authors declare that no competing interests exist.

### Funding

| Funder | Grant reference number | Author |
|---|---|---|
| Fogarty International Center | | Amanda C Perofsky<br>Chelsea L Hansen<br>Cécile Viboud |

| Funder | Grant reference number | Author |
| --- | --- | --- |
| National Science Foundation | 1354890 | Amanda C Perofsky |
| National Institutes of Health | F31 AI140714 | John Huddleston |
| National Institutes of Health | R01 AI165821 | John Huddleston |
| Cancer Research UK | FC001030 | Nicola Lewis<br>Lynne Whittaker<br>Burcu Ermetal<br>Ruth Harvey<br>Monica Galiano<br>Rodney Stuart Daniels<br>John W McCauley |
| Medical Research Council | FC001030 | Nicola Lewis<br>Lynne Whittaker<br>Burcu Ermetal<br>Ruth Harvey<br>Monica Galiano<br>Rodney Stuart Daniels<br>John W McCauley |
| Wellcome Trust | FC001030 | Nicola Lewis<br>Lynne Whittaker<br>Burcu Ermetal<br>Ruth Harvey<br>Monica Galiano<br>Rodney Stuart Daniels<br>John W McCauley |
| Ministry of Health, Labour and Welfare | 10110400 | Seiichiro Fujisaki<br>Kazuya Nakamura<br>Noriko Kishida<br>Shinji Watanabe<br>Hideki Hasegawa |
| Ministry of Health, Labour and Welfare | 10111800 | Seiichiro Fujisaki<br>Kazuya Nakamura<br>Noriko Kishida<br>Shinji Watanabe<br>Hideki Hasegawa |
| Japan Agency for Medical Research and Development | JP22fk0108118 | Shinji Watanabe |
| Japan Agency for Medical Research and Development | JP23fk0108662 | Shinji Watanabe |
| Department of Health and Aged Care, Australian Government | | Sheena G Sullivan<br>Ian G Barr<br>Kanta Subbarao |
| Department of Health, Government of Western Australia | | Sheena G Sullivan<br>Ian G Barr<br>Kanta Subbarao |
| National Institutes of Health | HHSN272201400008C | Florian Krammer |
| National Institutes of Health | 75N93021C00014 | Florian Krammer |
| National Institutes of Health | 75N93019C00051 | Florian Krammer |
| National Institutes of Health | R35 GM119774 | Trevor Bedford |
| National Institutes of Health | R01 AI127893 | Trevor Bedford |

| Funder | Grant reference number | Author |
|---|---|---|
| Howard Hughes Medical Institute | | Trevor Bedford |

The funders had no role in study design, data collection and interpretation, or the decision to submit the work for publication. For the purpose of Open Access, the authors have applied a CC BY public copyright license to any Author Accepted Manuscript version arising from this submission.

## Author contributions

Amanda C Perofsky, Conceptualization, Data curation, Software, Formal analysis, Funding acquisition, Validation, Investigation, Visualization, Methodology, Writing – original draft, Project administration, Writing – review and editing; John Huddleston, Data curation, Software, Formal analysis, Validation, Investigation, Visualization, Methodology, Writing – review and editing; Chelsea L Hansen, Resources, Data curation, Software, Formal analysis, Investigation, Writing – review and editing; John R Barnes, Thomas Rowe, Xiyan Xu, Rebecca Kondor, David E Wentworth, Nicola Lewis, Lynne Whittaker, Burcu Ermetal, Ruth Harvey, Monica Galiano, Rodney Stuart Daniels, John W McCauley, Seiichiro Fujisaki, Kazuya Nakamura, Noriko Kishida, Shinji Watanabe, Hideki Hasegawa, Sheena G Sullivan, Ian G Barr, Kanta Subbarao, Resources, Investigation, Methodology, Writing – review and editing; Florian Krammer, Resources, Data curation, Funding acquisition, Investigation, Writing – review and editing; Trevor Bedford, Conceptualization, Resources, Software, Supervision, Funding acquisition, Methodology, Project administration; Cécile Viboud, Conceptualization, Resources, Software, Supervision, Funding acquisition, Methodology, Project administration, Writing – review and editing

## Author ORCIDs

Amanda C Perofsky ⓘ https://orcid.org/0000-0001-7341-9193
John Huddleston ⓘ http://orcid.org/0000-0002-4250-2063
Chelsea L Hansen ⓘ https://orcid.org/0000-0002-4526-6772
Rebecca Kondor ⓘ https://orcid.org/0000-0002-2596-4282
David E Wentworth ⓘ https://orcid.org/0000-0002-5190-980X
Rodney Stuart Daniels ⓘ https://orcid.org/0000-0003-2818-5089
John W McCauley ⓘ https://orcid.org/0000-0002-4744-6347
Kanta Subbarao ⓘ https://orcid.org/0000-0003-1713-3056
Trevor Bedford ⓘ https://orcid.org/0000-0002-4039-5794
Cécile Viboud ⓘ https://orcid.org/0000-0003-3243-4711

## Ethics

The human surveillance data and viral sequence data used in this study are anonymous and were openly available to the public prior to the initiation of this study. Therefore, this research does not constitute human subjects research. Influenza syndromic and virologic surveillance data can be obtained from the U.S. Centers for Disease Control and Prevention (CDC) FluView Interactive dashboard (https://www.cdc.gov/flu/weekly/fluviewinteractive.htm). Influenza viral sequence data can be obtained from the Global Initiative on Sharing All Influenza Data (GISAID) database (https://gisaid.org/). The GISAID Initiative ensures that open access to data in GISAID is provided free-of-charge to all individuals that agreed to identify themselves and agreed to uphold the GISAID sharing mechanism governed through its Database Access Agreement. This study followed the Strengthening the Reporting of Observational Studies in Epidemiology (STROBE) reporting guidelines for cross-sectional studies.

Reviewer #1 (Public review): https://doi.org/10.7554/eLife.91849.3.sa1
Author response https://doi.org/10.7554/eLife.91849.3.sa2

# Additional files

## Supplementary files

• Supplementary file 1. GISAID accessions and metadata for influenza H3 and N2 sequences, including originating labs and submitting labs.
• MDAR checklist

## Data availability

Sequence data are available from GISAID using accession ids provided in *Supplementary file 1*. Source code for phylogenetic analyses, inferred HI titers from serological measurements, and evolutionary fitness measurements are available in the GitHub repository https://github.com/blab/perofsky-ili-antigenicity (copy archived at *Huddleston, 2024*). The five replicate trees for HA and NA can be found at https://nextstrain.org/groups/blab/ under the keyword "perofsky-ili-antigenicity". Epidemiological data, datasets combining seasonal evolutionary fitness measurements and epidemic metrics, and source code for calculating epidemic metrics and performing statistical analyses are available at https://doi.org/10.5281/zenodo.11188848 and https://github.com/aperofsky/H3N2_Antigenic_Epi (copy archived at *Perofsky, 2024*). Raw serological measurements are restricted from public distribution by previous data sharing agreements.

The following dataset was generated:

| Author(s) | Year | Dataset title | Dataset URL | Database and Identifier |
|---|---|---|---|---|
| Perofsky A | 2024 | aperofsky/H3N2_Antigenic_Epi: Initial release (v1.0.0) | https://doi.org/10.5281/zenodo.11188848 | Zenodo, 10.5281/zenodo.11188848 |

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

## Appendix 1

## GISAID Acknowledgements

WHO Collaborating Centre for Reference and Research on Influenza, Victorian Infectious Diseases Reference Laboratory, Australia; WHO Collaborating Centre for Reference and Research on Influenza, Chinese National Influenza Center, China; WHO Collaborating Centre for Reference and Research on Influenza, National Institute of Infectious Diseases, Japan; The Crick Worldwide Influenza Centre, The Francis Crick Institute, United Kingdom; WHO Collaborating Centre for the Surveillance, Epidemiology and Control of Influenza, Centers for Disease Control and Prevention, United States; Aalesund Sjukehus, Norway; ADImmune Corporation, Taiwan; ADPH Bureau of Clinical Laboratories, United States; Aichi Prefectural Institute of Public Health, Japan; Akershus University Hospital, Norway; Akita Research Center for Public Health and Environment, Japan; Alabama State Laboratory, United States; Alaska State Public Health Laboratory, United States; Alaska State Virology Lab, United States; Alfred Hospital, Australia; Aomori Prefectural Institute of Public Health and Environment, Japan; Aristotelian University of Thessaloniki, Greece; Arizona Department of Health Services, United States; Arkansas Department of Health, United States; Atlanta VA Medical Center, United States; Auckland Healthcare, New Zealand; Auckland Hospital, New Zealand; Austin Health, Australia; Baylor College of Medicine, United States; Baylor Scott and White Health, United States; California Department of Health Services, United States; Canberra Hospital, Australia; Cantacuzino Institute, Romania; Canterbury Health Services, New Zealand; Caribbean Epidemiology Center, Trinidad and Tobago; CDC Central Asia Office; CDC GAP Nigeria, Nigeria; CDC Kenya, Kenya; CEMIC University Hospital, Argentina; CENETROP, Bolivia; Center For Medical Microbiology, College of Public Health, University of the Philippines, Philippines; Center for Public Health and Environment, Hiroshima Prefectural Technology Research Institute, Japan; Center of Hygiene And Epidemiology, Kirov Oblast, Russian Federation; Center of Hygiene And Epidemiology, Yamalo-Nenets Autonomous Okrug, Russian Federation; Center of Hygiene And Epidemiology, The Republic Of Dagestan, Russian Federation; Central Health Laboratory, Mauritius; Central Laboratory of Public Health, Paraguay; Central Public Health Laboratory, Ministry of Health, Oman; Central Public Health Laboratory, Palestinian Territory; Central Public Health Laboratory, Papua New Guinea; Central Public Health Reference Laboratory, Sierra Leone; Central Research Institute for Epidemiology, Russian Federation; Central Virology Laboratory, Israel; Centre de Recherche Médicale et Sanitaire, Niger; Centre for Diseases Control and Prevention, Armenia; Centre for Infections, Health Protection Agency, United Kingdom; Centre National de Référence des Virus des Infections Respiratoires, France; Centre National de Référence du Virus Influenza Région Sud, France; Centre Pasteur du Cameroun, Cameroon; Centro de Investigación Regional Dr. Hideyo Noguchi, Mexico; Chiba City Institute of Health and Environment, Japan; Chiba Prefectural Institute of Public Health, Japan; Children's Mercy Hospital, United States; Children's Hospital Westmead, Australia; Chuuk State Hospital, Micronesia, Federated States of; City of El Paso Dept of Public Health, United States; City of Milwaukee Health Department, United States; Clinical Virology Unit, CDIM, Australia; Colorado Department of Health Lab, United States; Connecticut Department of Public Health, United States; Contiguo a Hospital Rosales, El Salvador; CSL Ltd, United States; Dallas County Health and Human Services, United States; DC Public Health Lab, United States; Delaware Public Health Lab, United States; Departamento de Laboratorio de Salud Publica, Uruguay; Department of Clinical Virology, University College London Hospitals NHS Foundation Trust, United Kingdom; Department of Health, Hong Kong; Department of Public Health, Niigata City, Japan; Department of Virology, Medical University Vienna, Austria; Disease Investigation Centre Wates (BBVW), Australia; Dorevitch Pathology, Australia; Drammen Hospital/Vestreviken HF, Norway; Ehime Prefecture Institute of Public Health and Environmental Science, Japan; Erasmus Medical Center, Netherlands; Erasmus University of Rotterdam, Netherlands; Ethiopian Health and Nutrition Research Institute (EHNRI), Ethiopia; Ethiopian Public Health Institute, Ethiopia; Evandro Chagas Institute, Brazil; Facultad de Medicina, Spain; FBUZ Center for Hygiene and Epidemiology, Russian Federation; Florida Department of Health, United States; Fred Hutchinson Cancer Research Center, United States; Fukui Prefectural Institute of Public Health, Japan; Fukuoka City Institute for Hygiene and the Environment, Japan; Fukuoka Institute of Public Health and Environmental Sciences, Japan; Fukushima Prefectural Institute of Public Health, Japan; Gart Naval General Hospital, United Kingdom; Georgia Public Health Laboratory, United States; Gifu Municipal Institute of Public Health, Japan; Gifu Prefectural Institute of Health and Environmental Sciences, Japan; Government Virus Unit, Hong Kong; Gunma Prefectural Institute of Public Health and Environmental Sciences, Japan; Hackensack University Medical Center, United States; Hamamatsu City Health Environment Research Center, Japan; Haukeland University Hospital, Dept. of Microbiology, Norway; Health Forde, Department of

Microbiology, Norway; Health Protection Agency, United Kingdom; Health Protection Inspectorate, Estonia; Hellenic Pasteur Institute, Greece; Helsinki University Central Hospital, Finland; Hiroshima City Institute of Public Health, Japan; Hobart Pathology, Australia; Hokkaido Institute of Public Health, Japan; Hôpital Cantonal Universitaire de Geneves, Switzerland; Hôpital Charles Nicolle, Tunisia; Hôpital Georges L. Dumont, Canada; Hospital Clinic de Barcelona, Spain; Houston Department of Health and Human Services, United States; Hyogo Prefectural Institute of Public Health and Consumer Sciences, Japan; Ibaraki Prefectural Institute of Public Health, Japan; International Centre For Diarrhoeal Disease Research, Bangladesh; Illinois Department of Public Health, United States; Indiana State Department of Health Laboratories, United States; Infectology Center of Latvia, Latvia; Innlandet Hospital Trust, Division Lillehammer, Department for Microbiology, Norway; INRB Service De Virologie, Democratic Republic of the Congo; Institut Fédératif de Recherche Lyon, France; Institut Louis Malardé Clinical Laboratory, French Polynesia; Institut National d'Hygiène, Morocco; Instituto Nacional de Investigación en Salud Pública, Ecuador; Institut National de Recherches en Sante Publique, Mauritania;; Institut de Recherche en Sciences de la Santé, Burkina Faso; Institut Pasteur d'Algerie, Algeria; Institut Pasteur de Bangui, Central African Republic; Institut Pasteur de Dakar, Senegal; Institut Pasteur de Madagascar, Madagascar; Institut Pasteur in Cambodia, Cambodia; Institut Pasteur New Caledonia, New Caledonia; Institut Pasteur, France; Institut Penyelidikan Perubatan, Malaysia; Institute National D'Hygiene, Togo; Institute of Environmental Science and Research, New Zealand; Institute of Environmental Science and Research, Tonga; Institute For Biomedical Sciences, Suriname; Institute of Environmental Science & Research, New Zealand; Institute of Epidemiology and Infectious Diseases, Ukraine; Institute of Epidemiology Disease Control and Research, Bangladesh; Institute of Immunology and Virology Torlak, Serbia; Institute of Medical and Veterinary Science (IMVS), Australia; Institute of Public Health, Serbia; Institute of Public Health, Albania; Institute of Public Health, Montenegro; Institute Pasteur du Cambodia, Cambodia; Instituto Adolfo Lutz, Brazil; Instituto Conmemorativo Gorgas de Estudios de la Salud, Panama; Instituto De Diagnostico Y Referencia Epidemiologicos, Mexico;Instituto de Salud Carlos III, Spain; Instituto de Salud Publica de Chile, Chile; Instituto Nacional de Enfermedades Infecciosas, Argentina; Instituto Nacional de Higiene Rafael Rangel, Venezuela, Bolivia; Instituto Nacional de Laboratoriosde Salud (INLASA), Bolivia; Instituto Nacional de Salud de Columbia, Colombia; Instituto Nacional de Saude, Portugal; Iowa State Hygienic Laboratory, United States; IRSS, Burkina Faso; Ishikawa Prefectural Institute of Public Health and Environmental Science, Japan; ISS, Italy; Istanbul University, Turkey; Istituto Di Igiene, Italy; Istituto Superiore di Sanità, Italy; Ivanovsky Research Institute of Virology RAMS, Russian Federation; Jiangsu Provincial Center for Disease Control and Prevention, China; John Hunter Hospital, Australia; Kagawa Prefectural Research Institute for Environmental Sciences and Public Health, Japan; Kagoshima Prefectural Institute for Environmental Research and Public Health, Japan; Kanagawa Prefectural Institute of Public Health, Japan; Kansas Department of Health and Environment, United States; Kawasaki City Institute of Public Health, Japan; KEMRI Wellcome Trust Research Programme, Kenya; Kentucky Division of Laboratory Services, United States; Kitakyusyu City Institute of Environmental Sciences, Japan; Klinisk Mikrobiologi, Hallands Sjukhus Halmstad, Sweden; Klinisk Mikrobiologi, Karolinska Universitetslaboratoriet, Karolinska Universitetssjukhuset Solna, Sweden; Klinisk Mikrobiologi, Laboratoriemedicin, Norrlands Universitetssjukhus Umea, Sweden; Klinisk Mikrobiologi, Sahlgrenska Universitetssjukhuset Goteborg, Sweden; Kobe Institute of Health, Japan; Kochi Public Health and Sanitation Institute, Japan; Kumamoto City Environmental Research Center, Japan; Kumamoto Prefectural Institute of Public Health and Environmental Science, Japan; Kyoto City Institute of Health and Environmental Sciences, Japan; Kyoto Prefectural Institute of Public Health and Environment, Japan; Laboratoire De Santé Publique Du Québec, Canada; Laboratoire National de Sante Publique, Haiti; Laboratoire National de Sante, Luxembourg; Laboratório Central do Estado do Paraná, Brazil; Laboratorio Central do Estado do Rio de Janeiro, Brazil; Laboratorio de Investigacion/Centro de Educacion Medica y Amistad Dominico Japones (CEMADOJA), Dominican Republic; Laboratorio De Isolamento Viral, Mozambique; Laboratorio De Referencia Nacional Virus Respiratorios, Instituto Nacional De Salud, Peru; Laboratorio De Saude Publico, Macao; Laboratorio de Virologia, Direccion de Microbiologia, Nicaragua; Laboratorio de Virus Respiratorio, Mexico; Laboratorio Di Virologia, Azienda Ospedaliero Universitaria Ospedali Riuniti Ancona, Italy; Laboratorio Nacional de Influenza, Costa Rica; Laboratorio Nacional De Salud Guatemala, Guatemala; Laboratorio Nacional de Virologia, Honduras; Laboratory Directorate, Jordan; Laboratory for Virology, National Institute of Public Health, Slovenia; Laboratory of Influenza and ILI, Belarus; LACEN/ES Laboratório Central de Saúde Pública do Estado do Espirito Santo, Brazil; LACEN/RS - Laboratório Central de Saúde Pública do Rio Grande do Sul, Brazil; LACEN-SC - Laboratório Central de Saúde Pública do Estado de Santa Catarina; Landspitali - University Hospital, Iceland; Lismore

Base Hospital, Australia; Lithuanian AIDS Center Laboratory, Lithuania; Los Angeles Quarantine Station, CDC Quarantine Epidemiology and Surveillance Team, United States; Louisiana Department of Health and Hospitals, United States; Maine Health and Environmental Testing Laboratory, United States; Marshfield Clinic Research Foundation, United States; Maryland Department of Health and Mental Hygiene, United States; Massachusetts Department of Public Health, United States; Mater Dei Hospital, Malta; Medical Research Institute, Sri Lanka; Medical University Vienna, Austria; Melbourne Pathology, Australia; Michigan Department of Community Health, United States; Microbiology Services Colindale, Public Health England, United Kingdom; Mie Prefecture Health and Environment Research Institute, Japan; Mikrobiologisk laboratorium, Sykehuset i Vestfold, Norway; Ministry of Health and Population, Egypt; Ministry of Health of Ukraine, Ukraine; Ministry of Health, Bahrain; Ministry of Health, Kiribati; Ministry of Health, Lao, People's Democratic Republic; Ministry of Health, NIHRD, Indonesia; Ministry of Health, Maldives; Ministry of Health, Oman; Ministry of Health Riyadh, Saudi Arabia; Ministry of Health, Singapore; Ministry of Health, Thailand; Minnesota Department of Health, United States; Mississippi Public Health Laboratory, United States; Missouri Department of Health and Senior Services, United States; Miyagi Prefectural Institute of Public Health and Environment, Japan; Miyazaki Prefectural Institute for Public Health and Environment, Japan; Molde Hospital, Laboratory for Medical Microbiology, Norway; Monash Medical Centre, Australia; Montana Laboratory Services Bureau, United States; Montana Public Health Laboratory, United States; Nagano City Health Center, Japan; Nagano Environmental Conservation Research Institute, Japan; Nagasaki Prefectural Institute For Environment Research and Public Health, Japan; Nagoya City Public Health Research Institute, Japan; NAMRU-2 U.S. Naval Medical Research Unit-2, Cambodia; NAMRU-2 U.S. Naval Medical Research Unit-2, Indonesia; NAMRU-6 U.S. Naval Medical Research Unit-6, Peru; Nara Prefectural Institute for Hygiene and Environment, Japan; National Center for Communicable Diseases, Mongolia; National Center For Epidemiology, National Influenza Center, Hungary; National Center for Laboratory and Epidemiology, Laos; National Centre for Disease Control and Public Health, Georgia; National Centre for Preventive Medicine, Moldova, Republic of; National Centre for Scientific Services for Virology and Vector Borne Diseases, Fiji; National Health Laboratory, Japan; National Health Laboratory, Myanmar; National Influenza Center CVD-Mali, Mali; National Influenza Center French Guiana and French Indies, French Guiana; National Influenza Center, Brazil; National Influenza Center, Mongolia; National Influenza Centre for Northern Greece, Greece; National Influenza Centre of Iraq, Iraq; National Influenza Lab, Tanzania, United Republic of; National Influenza Reference Laboratory, Nigeria; National Institute for Communicable Disease, South Africa; National Institute for Health and Welfare, Finland; National Institute For Medical Research, United Kingdom; National Institute For Public Health and The Environment (RIVM), Netherlands; National Institute of Health, Korea, Republic of; National Institute of Health, Pakistan; National Institute of Hygiene and Epidemiology, Vietnam; National Institute of Infectious Diseases (NIID), Japan; National Institute of Public Health of Kosova, Kosovo; National Institute of Public Health - National Institute of Hygiene, Poland; National Institute of Public Health, Czech Republic; National Institute of Virology, India; National Microbiology Laboratory, Health Canada, Canada; National Public Health Institute of Slovakia, Slovakia; National Public Health Laboratory, Cambodia; National Public Health Laboratory, Ministry of Health, Singapore, Singapore; National Public Health Laboratory, Nepal; National Public Health Laboratory, Singapore; National Reference Laboratory, Kazakhstan; National Referral Hospital, Solomon Islands; National University Hospital, Singapore; National Virology Laboratory, Center Microbiological Investigations, Kyrgyzstan; National Veterinary Institute, Sri Lanka; National Virus Reference Laboratory, Ireland; Naval Health Research Center, United States; NCDC Public Health Reference Laboratory, Nigeria; Nebraska Public Health Lab, United States; Nevada State Health Laboratory, United States; New Hampshire Public Health Laboratories, United States; New Jersey Department of Health and Senior Services, United States; New Mexico Department of Health, United States; New York City Department of Health, United States; New York Presbyterian Hospital Columbia University Medical Center, Microbiology Department, United States; New York State Department of Health, United States; Nicosia General Hospital, Cyprus; Niigata City Institute of Public Health and Environment, Japan; Niigata Prefectural Institute of Public Health and Environmental Sciences, Japan; Niigata University, Japan; N ingbo International Travel Healthcare Center, China; North Carolina State Laboratory of Public Health, United States; North Dakota Department of Health, United States; Norwegian Institute of Public Health, Norway; Ohio Department of Health Laboratories, United States; Oita Prefectural Institute of Health and Environment, Japan; Okayama Prefectural Institute for Environmental Science and Public Health, Japan; Okinawa Prefectural Institute of Health and Environment, Japan; Oklahoma State Department of Health, United States; Ontario Agency for Health Protection and Promotion (OAHPP), Canada; Oregon Public Health Laboratory, United States; Osaka City Institute of Public

Health and Environmental Sciences, Japan; Osaka Prefectural Institute of Public Health, Japan; Oslo University Hospital, Ulleval Hospital, Dept. of Microbiology, Norway; Ostfold Hospital - Fredrikstad, Dept. of Microbiology, Norway; Oswaldo Cruz Institute - FIOCRUZ - Laboratory of Respiratory Viruses and Measles (LVRS), Brazil; Papua New Guinea Institute of Medical Research, Papua New Guinea; Pasteur Institut of Côte D'ivoire, Côte D'ivoire; Pasteur Institute of Ho Chi Minh City, Vietnam; Pasteur Institute, Influenza Laboratory, Vietnam; Pathwest QE II Medical Centre, Australia; Pennsylvania Department of Health, United States; Prince of Wales Hospital, Australia; Princess Margaret Hospital for Children, Australia; Provincial Laboratory For Public Health For Northern Alberta, Canada; Provincial Laboratory for Public Health, Alberta Health Services, Canada; Provincial Laboratory of Public Health For Southern Alberta, Canada; Public Health Agency of Sweden, Sweden; Public Health Laboratory Services Branch, Centre for Health Protection, Hong Kong; Public Health Laboratory, Barbados; Public Health Laboratory, Virology Unit, Kuwait; Public Health Ontario, Canada; Public Health Wales Microbiology, United Kingdom; Puerto Rico Department of Health, Puerto Rico; Queensland Health Forensic and Scientific Services, Australia; Queensland Health Scientific Services, Australia; Rafic Hariri University Hospital, Lebanon; Refik Saydam National Public Health Agency, Turkey; Regent Seven Seas Cruises, United States; Republic Institute For Health Protection, Macedonia; Republic of Nauru Hospital, Nauru; Republican Anti Plague Station, Azerbaijan, Republic of; Research Institute for Environmental Sciences and Public Health of Iwate Prefecture, Japan; Research Institute of Health Sciences (IRSS), Burkina Faso; Research Institute of Tropical Medicine, Philippines; Rhode Island Department of Health, United States; Robert-Koch-Institute, Germany; Roy Romanow Provincial Laboratory, Canada; Royal Centre For Disease Control, Bhutan; Royal Children's Hospital, Australia; Royal Darwin Hospital, Australia; Royal Hobart Hospital, Australia; Royal Melbourne Hospital, Australia; Russian Academy of Medical Sciences, Russian Federation; Rwanda Biomedical Center, National Reference Laboratory, Rwanda; Saga Prefectural Institute of Public Health and Pharmaceutical Research, Japan; Sagamihara City Laboratory of Public Health, Japan; Saitama City Institute of Health Science and Research, Japan; Saitama Institute of Public Health, Japan; Saitama Medical University, Japan; Sakai City Institute of Public Health, Japan; San Antonio Metropolitan Health, United States; Sandringham, National Institute for Communicable Disease, South Africa; Sapporo City Institute of Public Health, Japan; Sciensano, Scientific Institute of Public Health, Belgium; Scientific Institute of Public Health, Belgium; Seattle and King County Public Health Lab, United States; Sendai City Institute of Public Health, Japan; Servicio de Microbiología Complejo Hospitalario de Navarra, Spain; Servicio de Microbiología Hospital Donostia, Spain; Servicio de Microbiología Hospital Meixoeiro, Spain; Servicio de Microbiología Hospital Miguel Servet, Spain; Servicio de Microbiología Hospital Ramón y Cajal, Spain; Servicio de Microbiología Hospital San Pedro de Alcántara, Spain; Servicio de Microbiología Hospital Universitario de Gran Canaria Doctor Negrín, Spain; Servicio de Microbiología Hospital Universitario Son Espases, Spain; Servicio de Microbiología Hospital Virgen de las Nieves, Spain; Servicio de Virosis Respiratorias INEI-ANLIS Carlos G. Malbran, Argentina; Seychelles Public Health Laboratory, Seychelles; Sheikh Khalifa Medical City, United Arab Emirates; Shanghai International Travel Healthcare Center, China; Shiga Prefectural Institute of Public Health, Japan; Shimane Prefectural Institute of Public Health and Environmental Science, Japan; Shizuoka City Institute of Environmental Sciences and Public Health, Japan; Shizuoka Institute of Environment and Hygiene, Japan; Singapore General Hospital, Singapore; Sorlandet Sykehus HF, Dept. of Medical Microbiology, Norway; South Carolina Department of Health, United States; South Dakota Public Health Lab, United States; Southern Nevada Public Health Lab, United States; Spokane Regional Health District, United States; St. Jude's Children's Research Hospital, United States; St. Olavs Hospital HF, Dept. of Medical Microbiology, Norway; State Agency, Infectology Center of Latvia, Latvia; State of Hawaii Department of Health, United States; State of Idaho Bureau of Laboratories, United States; State Research Center of Virology and Biotechnology Vector, Russian Federation; Statens Serum Institute, Denmark; Stavanger Universitetssykehus, Avd. for Medisinsk Mikrobiologi, Norway; Supreme Health Council, Qatar; Swedish Institute for Infectious Disease Control, Sweden; Taiwan CDC, Taiwan; Tan Tock Seng Hospital, Singapore; Tarrant County Public Health, Texas, United States; Tehran University of Medical Sciences, Iran; Tennessee Department of Health Laboratory-Nashville, United States; Texas Children's Hospital, United States; Texas Department of State Health Services, United States; Texas Department of State Health Services, South Texas Laboratory, United States; Thai National Influenza Center, Thailand; Thailand MOPH-U.S. CDC Collaboration (IEIP), Thailand; The Nebraska Medical Center, United States; The NIAID Influenza Genome Sequencing Consortium, United States; Thüringer Landesamt für Verbraucherschutz, Germany; Tochigi Prefectural Institute of Public Health and Environmental Science, Japan; Tokushima Prefectural Centre for Public Health and Environmental Sciences, Japan; Tokyo Metropolitan Institute of Public Health, Japan; Tottori Prefectural Institute of

Public Health and Environmental Science, Japan; Toyama Institute of Health, Japan; U.S. Air Force School of Aerospace Medicine, United States; U.S. AMC AFRIMS Department of Virology, Thailand; U.S. Army Medical Research Unit, Kenya (USAMRU-K), Geis Human Influenza Program, Kenya; Uganda Virus Research Institute (UVRI), National Influenza Center, Uganda; Unilabs Laboratoriemedicin Stockholm Solna, Sweden; Unilabs Laboratoriemedicin Vastra Gotaland, Skaraborgs Sjukhus Skovde, Sweden; Universidad de Valladolid, Spain; Universitetssykehuset Nord-Norge HF, Norway; University Malaya, Malaysia; University of Genoa, Italy; University of Ghana, Ghana; University of Michigan SPH EPID, United States; University of The West Indies, Jamaica; University of Vienna, Austria; University of Virginia, Medical Labs/Microbiology, United States; University Teaching Hospital, Zambia; Uoc Policlinico Di Bari Dimo, Italy; UPMC-CLB Dept of Microbiology, United States; Utah Department of Health, United States; Utah Public Health Laboratory, United States; Utsunomiya City Institute of Public Health and Environment Science, Japan; VACSERA, Egypt; Vanderbilt University Medical Center, United States; Vefa Center, Tajikistan; Vermont Department of Health Laboratory, United States; Victorian Infectious Diseases Reference Laboratory, Australia; Virginia Division of Consolidated Laboratories, United States; Virus Research Center, Sendai Medical Center, Japan; Wakayama City Institute of Public Health, Japan; Wakayama Prefectural Research Center of Environment and Public Health, Japan; Walter Reed Army Institute of Research, United States; Washington State Public Health Laboratory, United States; West Virginia Office of Laboratory Services, United States; Westchester County Department of Laboratories and Research, United States; Westmead Hospital, Australia; National Influenza Centre Russian Federation, Russian Federation; WHO National Influenza Centre, National Institute of Medical Research (NIMR), Thailand; WHO National Influenza Centre, Norway; Wisconsin State Laboratory of Hygiene, United States; Wyoming Public Health Laboratory, United States; Yamagata Prefectural Institute of Public Health, Japan; Yamaguchi Prefectural Institute of Public Health and Environment, Japan; Yamanashi Institute for Public Health, Japan; Yap State Hospital, Micronesia; Yokohama City Institute of Health, Japan; Yokosuka Institute of Public Health, Japan

