## [Editor Report · eLife assessment]

This paper explores the relationships among evolutionary and epidemiological quantities in influenza, and presents **fundamental** findings that substantially advance our understanding of the drivers of influenza epidemics. The authors use a rich set of data sources to gather and analyze **compelling** evidence on the roles of genetic distance, other influenza dynamics and epidemiological indicators in predicting influenza epidemics. The central findings highlight the significant influence of genetic distance on A(H3N2) virus epidemiology and emphasize the role of A(H1N1) virus incidence in shaping A(H3N2) epidemics, suggesting subtype interference as a key factor. This paper also makes relevant data available to the research community.

---

## [Referee Report · Reviewer #1 (Public review)]

Summary:

The authors aimed to investigate the contribution of antigenic drift in the HA and NA genes of seasonal influenza A(H3N2) virus to their epidemic dynamics. Analyzing 22 influenza seasons before the COVID-19 pandemic, the study explored various antigenic and genetic markers, comparing them against indicators characterizing the epidemiology of annual outbreaks. The central findings highlight the significant influence of genetic distance on A(H3N2) virus epidemiology and emphasize the role of A(H1N1) virus incidence in shaping A(H3N2) epidemics, suggesting subtype interference as a key factor.

Major Strengths:

The paper is well-organized, written with clarity, and presents a comprehensive analysis. The study design, incorporating a span of 22 seasons, provides a robust foundation for understanding influenza dynamics. The inclusion of diverse antigenic and genetic markers enhances the depth of the investigation, and the exploration of subtype interference adds valuable insights.

Major Weaknesses:

While the analysis is thorough, some aspects require deeper interpretation, particularly in the discussion of certain results. Clarity and depth could be improved in the presentation of findings, and minor adjustments are suggested. Furthermore, the evolving dynamics of H3N2 predominance post-2009 need better elucidation.

Comments on revised version:

The authors have addressed each of the comments well. I have no further comments.

---

## [Author Response]

The following is the authors’ response to the original reviews.

**Public Reviews:**

**Reviewer #1 (Public Review):**
Summary:The authors aimed to investigate the contribution of antigenic drift in the HA and NA genes of seasonal influenza A(H3N2) virus to their epidemic dynamics. Analyzing 22 influenza seasons before the COVID-19 pandemic, the study explored various antigenic and genetic markers, comparing them against indicators characterizing the epidemiology of annual outbreaks. The central findings highlight the significant influence of genetic distance on A(H3N2) virus epidemiology and emphasize the role of A(H1N1) virus incidence in shaping A(H3N2) epidemics, suggesting subtype interference as a key factor.Major Strengths:The paper is well-organized, written with clarity, and presents a comprehensive analysis. The study design, incorporating a span of 22 seasons, provides a robust foundation for understanding influenza dynamics. The inclusion of diverse antigenic and genetic markers enhances the depth of the investigation, and the exploration of subtype interference adds valuable insights.Major Weaknesses:While the analysis is thorough, some aspects require deeper interpretation, particularly in the discussion of certain results. Clarity and depth could be improved in the presentation of findings. Furthermore, the evolving dynamics of H3N2 predominance post-2009 need better elucidation.
**Reviewer #2 (Public Review):**
Summary: This paper aims to achieve a better understanding of how the antigenic or genetic compositions of the dominant influenza A viruses in circulation at a given time are related to key features of seasonal influenza epidemics in the US. To this end, the authors analyze an extensive dataset with a range of statistical, data science and machine learning methods. They find that the key drivers of influenza A epidemiological dynamics are interference between influenza A subtypes and genetic divergence, relative to the previous one or two seasons, in a broader range of antigenically related sites than previously thought.Strengths: A thorough investigation of a large and complex dataset.Weaknesses: The dataset covers a 21 year period which is substantial by epidemiological standards, but quite small from a statistical or machine learning perspective. In particular, it was not possible to follow the usual process and test predictive performance of the random forest model with an independent dataset.
**Reviewer #3 (Public Review):**
Summary:This paper explores the relationships among evolutionary and epidemiological quantities in influenza, using a wide range of datasets and features, and using both correlations and random forests to examine, primarily, what are the drivers of influenza epidemics. It's a strong paper representing a thorough and fascinating exploration of potential drivers, and it makes a trove of relevant data readily available to the community.Strengths:This paper makes links between epidemiological and evolutionary data for influenza. Placing each in the context of the other is crucial for understanding influenza dynamics and evolution and this paper does a thorough job of this, with many analyses and nuances. The results on the extent to which evolutionary factors relate to epidemic burden, and on interference among influenza types, are particularly interesting. The github repository associated with the paper is clear, comprehensive, and well-documented.Weaknesses:The format of the results section can be hard to follow, and we suggest improving readability by restructuring and simplifying in some areas. There are a range of choices made about data preparation and scaling; the authors could explore sensitivity of the results to some of these.

Response to public reviews

We appreciate the positive comments from the reviewers and have implemented or responded to all of the reviewers’ recommendations.

In response to Reviewer 1, we expand on the potential drivers and biological implications of the findings pointed out in their specific recommendations. For example, we now explicitly mention that antigenically distinct 3c.2a and 3c.3a viruses began to co-circulate in 2012 and underwent further diversification during subsequent seasons in our study. We note that, after the 2009 A(H1N1) pandemic, the mean fraction of influenza positive cases typed as A(H3N2) in A(H3N2) dominant seasons is lower compared to A(H3N2) dominant seasons prior to 2009. We propose that the weakening of A(H3N2) predominance may be linked to the diversification of A(H3N2) viruses during the 2010s, wherein multiple antigenically distinct clades with similar fitness circulated in each season, as opposed to a single variant with high fitness.

In response to Reviewer 2, we agree that it would be ideal and best practice to measure model performance with an independent test set, but our dataset includes only ~20 seasons. Predictions of independent test sets of 2-3 seasons had unstable performance, which indicates we do not have sufficient power to measure model performance with a test set this small. In the revised manuscript, we provide more justification and clarification of our methodology. Instead of testing model performance on an independent test set, we use leave-one-season-out cross-validation to train models and measure model performance, wherein each “assessment” set contains one season of data (predicted by the model), and the corresponding “analysis” set (“fold”) contains the remaining seasons. This approach is roughly analogous to splitting data into training and test sets, but all seasons are used at some point in the training of the model (Kuhn & Johnson, 2019).

In response to Reviewer 3, we follow the reviewer’s advice to put the Methods section before the Results section. Concerning Reviewer 3’s question about the sensitivity of our results to data preparation and rescaling, we provide more justification and clarification of our methodology in the revised manuscript. In our study, we adjust influenza type/subtype incidences for differences in reporting between the pre- and post-2009 pandemic periods and across HHS regions. We adjust for differences in reporting between the pre- and post-2009 periods because the US CDC and WHO increased laboratory testing capacity in response to the 2009 A(H1N1) pandemic, which led to substantial, long-lasting improvements to influenza surveillance that are still in place today. Figure 1 - figure supplement 2 shows systematic increases in influenza test volume in all HHS regions after the 2009 pandemic. Given the substantial increase in test volume after 2009, we opted to keep the time trend adjustment for the pre- and post-2009 pandemic periods and evaluate whether adjusting for regional reporting differences affects our results. When estimating univariate correlations between various A(H3N2) epidemic metrics and evolutionary indicators, we found qualitatively equivalent results when adjusting for both pre- and post-2009 pandemic reporting and regional reporting versus only adjusting for the pre- and post-2009 pandemic reporting.

**Reviewer #1 (Recommendations For The Authors):**
Specific comments:(1) Line 155-156. Request for a reference for: "Given that protective immunity wanes after 1-4 years"

We now include two references (He et al. 2015 and Wraith et al. 2022), which were cited at the beginning of the introduction when referring to the duration of protective immunity for antigenically homologous viruses. (Lines 640-642 in revised manuscript)

(2) Line 162-163: Request a further explanation of the negative correlation between seasonal diversity of HA and NA LBI values and NA epitope distance. Clarify biological implications to aid reader understanding.

In the revised manuscript we expand on the biological implications of A(H3N2) virus populations characterized by high antigenic novelty and low LBI diversity.

Lines 649-653:

“The seasonal diversity of HA and NA LBI values was negatively correlated with NA epitope distance (Figure 2 – figure supplements 5 – 6), with high antigenic novelty coinciding with low genealogical diversity. This association suggests that selective sweeps tend to follow the emergence of drifted variants with high fitness, resulting in seasons dominated by a single A(H3N2) variant rather than multiple cocirculating clades.”

(3) Figure S3 legend t-2 may be marked as t-1.

Thank you for catching this. We have fixed this typo. Note: Figure S3 is now Figure 2 – figure supplement 5.

(4) Lines 201-214. The key takeaways from the analysis of subtype dominance are ultimately not clear. It also misses the underlying dynamics that H3N2 predominance following an evolutionary change has waned since 2009.

In the revised manuscript we elaborate on key takeaways concerning the relationship between antigenic drift and A(H3N2) dominance. We also add a caveat noting that A(H3N2) predominance is weaker during the post-2009 period, which may be linked to the diversification of A(H3N2) lineages after 2012. We do not know of a reference that links the diversification of A(H3N2) viruses in the 2010s to a particular evolutionary change. Therefore, we do not attribute the diversification of A(H3N2) viruses to a specific evolutionary change in A(H3N2) variants circulating at the time (A/Perth/16/2009-like strains (PE09)). Instead, we allude to the potential role of A(H3N2) diversification in creating multiple co-circulating lineages that may have less of a fitness advantage.

Lines 681-703:

“We explored whether evolutionary changes in A(H3N2) may predispose this subtype to dominate influenza virus circulation in a given season. A(H3N2) subtype dominance – the proportion of influenza positive samples typed as A(H3N2) – increased with H3 epitope distance (t – 2) (R^2^ = 0.32, P = 0.05) and N2 epitope distance (t – 1) (R^2^ = 0.34, P = 0.03) (regression results: Figure 4; Spearman correlations: Figure 3 – figure supplement 1). Figure 4 illustrates this relationship at the regional level across two seasons in which A(H3N2) was nationally dominant, but where antigenic change differed. In 2003-2004, we observed widespread dominance of A(H3N2) viruses after the emergence of the novel antigenic cluster, FU02 (A/Fujian/411/2002-like strains). In contrast, there was substantial regional heterogeneity in subtype circulation during 2007-2008, a season in which A(H3N2) viruses were antigenically similar to those circulating in the previous season. Patterns in type/subtype circulation across all influenza seasons in our study period are shown in Figure 4 – figure supplement 1. As observed for the 2003-2004 season, widespread A(H3N2) dominance tended to coincide with major antigenic transitions (e.g., A/Sydney/5/1997 (SY97) seasons, 1997-1998 to 1999-2000; A/California/7/2004 (CA04) season, 20042005), though this was not universally the case (e.g., A/Perth/16/2009 (PE09) season, 2010-2011).

After the 2009 A(H1N1) pandemic, A(H3N2) dominant seasons still occurred more frequently than A(H1N1) dominant seasons, but the mean fraction of influenza positive cases typed as A(H3N2) in A(H3N2) dominant seasons was lower compared to A(H3N2) dominant seasons prior to 2009. Antigenically distinct 3c.2a and 3c.3a viruses began to co-circulate in 2012 and underwent further diversification during subsequent seasons in our study (https://nextstrain.org/seasonal-flu/h3n2/ha/12y@2024-05-13)(Dhanasekaran et al., 2022; Huddleston et al., 2020; Yan et al., 2019). The decline in A(H3N2) predominance during the post-2009 period may be linked to the genetic and antigenic diversification of A(H3N2) viruses, wherein multiple lineages with similar fitness co-circulated in each season.”

(5) Line 253-255: It would be beneficial to provide a more detailed interpretation of the statement that "pre-2009 seasonal A(H1N1) viruses may limit the circulation of A(H3N2) viruses to a greater extent than A(H1N1)pdm09 viruses." Elaborate on the cause-and-effect relationship within this statement.

In the revised manuscript we suggest that seasonal A(H1N1) viruses may interfere with the circulation of A(H3N2) viruses to a greater extent than A(H1N1)pdm09 viruses, because seasonal A(H1N1) viruses and A(H3N2) are more closely related, and thus may elicit stronger cross-reactive T cell responses.

Lines 738-745:

“The internal gene segments NS, M, NP, PA, and PB2 of A(H3N2) viruses and pre-2009 seasonal A(H1N1) viruses share a common ancestor (Webster et al., 1992) whereas A(H1N1)pdm09 viruses have a combination of gene segments derived from swine and avian reservoirs that were not reported prior to the 2009 pandemic (Garten et al., 2009; Smith et al., 2009). Non-glycoprotein genes are highly conserved between influenza A viruses and elicit cross-reactive antibody and T cell responses (Grebe et al., 2008; Sridhar, 2016). Because pre-2009 seasonal A(H1N1) viruses and A(H3N2) are more closely related, we hypothesized that seasonal A(H1N1) viruses could potentially limit the circulation of A(H3N2) viruses to a greater extent than A(H1N1)pdm09 viruses, due to greater T cell-mediated cross-protective immunity.”

(6) In the results section, many statements report statistical results of correlation analyses. Consider providing further interpretations of these results, such as the implications of nonsignificant correlations and how they support or contradict the hypothesis or previous studies. For example, the statement on line 248 regarding the lack of significant correlation between influenza B epidemic size and A(H3N2) epidemic metrics would benefit from additional discussion on what this non-significant correlation signifies and how it relates to the hypothesis or previous research.

In the Discussion section, we suggest that the lack of an association between influenza B circulation and A(H3N2) epidemic metrics is due to few T and B cell epitopes shared between influenza A and B viruses (Terajima et al., 2013).

Lines 1005-1007 in revised manuscript (Lines 513-515 in original manuscript):

“Overall, we did not find any indication that influenza B incidence affects A(H3N2) epidemic burden or timing, which is not unexpected, given that few T and B cell epitopes are shared between the two virus types (Terajima et al., 2013).”

Minor comments:(1) Line 116-122: Include a summary statistical description of all collected data sets, detailing the number of HA and NA sequence data and their sources. Briefly describe subsampled data sets, specifying preferences (e.g., the number of HA or NA sequence data collected from each region).

In our revised manuscript we now include supplementary tables that summarize the number of A/H3 and

A/N2 sequences in each subsampled dataset, aggregated by world region, for all seasons combined (Figure 2 - table supplements 1 - 2). We also include supplementary figures showing the number of sequences collected in each month and each season in North America versus the other nine world regions combined (Figure 2 - figure supplements 1 - 2). Subsampled datasets are plotted individually in the figures below but individual time series are difficult to discern due to minor differences in sequence counts across the datasets.

(2) Figure 7A: Due to space limitations, consider rounding numbers on the x-axis to whole numbers for clarity.

Thank you for this suggestion. In the revised manuscript we round numbers in the axes of Figure 7A (Figure 9A in the revised manuscript) so that the axes are less crowded.

(3) Figure 4C & Figure 4D: Note that Region 10 (purple) data were unavailable for seasons before 2009 (lines 1483-1484). Label each region on the map with its respective region number (1 to 10) and indicate this in the legend for easy identification.

In our original submission, the legend for Figure 4 included “Data for Region 10 (purple) were not available for seasons prior to 2009” at the end of the caption. We have moved this sentence, as well as other descriptions that apply to both C and D, so that they follow the sentence “C-D. Regional patterns of influenza type and subtype incidence during two seasons when A(H3N2) was nationally dominant.”

In our revised manuscript, Figure 4, and Figure 4 - figure supplement 1 (Figure S10 in original submission) include labels for each HHS region.

We did not receive specific recommendations from Reviewer #2. However, our responses to Reviewer #3 addresses the study’s weaknesses mentioned by Reviewer #2.

**Reviewer #3 (Recommendations For The Authors):**
This paper explores the relationships among evolutionary and epidemiological quantities in influenza, using a wide range of datasets and features, and using both correlations and random forests to examine, primarily, what are the drivers of influenza epidemics.This is a work horse of paper, in the volumes of data that are analyzed and the extensive analysis that is done. The data that are provided are a treasure trove resource for influenza modelers and for anyone interested in seeing influenza surveillance data in the context of evolution, and evolutionary information in the context of epidemiology.L53 - end of sentence "and antigenic drift": not sure this fits, explain? I thought this sentence was in contrast to antigenic drift.

Thank you for catching this. We did not intend to include “and antigenic drift” at the end of this sentence and have removed it (Line 59).

Para around L115: would using primarily US data be a limitation, because it's global immunity that shapes success of strains? Or, how much does each country's immunity and vaccination and so on actually shape what strains succeed there, compared to global/international factors?

The HA and NA phylogenetic trees in our study are enriched with U.S. sequences because our study focuses on epidemiological dynamics in the U.S., and we wanted to prioritize A(H3N2) viruses that the U.S. human population encountered in each season. We agree with the reviewer that the world population may be the right scale to understand how immunity, acquired by vaccination or natural infection, may shape the emergence and success of new lineages that will go on to circulate globally. However, our study assesses the overall impact of antigenic drift on regional A(H3N2) epidemic dynamics in the U.S. In other words, our driving question is whether we can predict the population-level impact of an A(H3N2) variant in the U.S., conditional on this particular lineage having established in the U.S. and circulating at relatively high levels. We do not assess the global or population-level factors that may influence which A(H3N2) virus lineages are successful in a given location or season.

We have added a clarifying sentence to the end of the Introduction to narrow the scope of the paper for the reader.

Line 114-116: “Rather than characterize in situ evolution of A(H3N2) lineages circulating in the U.S., we study the epidemiological impacts of antigenic drift once A(H3N2) variants have arrived on U.S. soil and managed to establish and circulate at relatively high levels.”

In the Results section, I found the format hard to follow, because of the extensive methodological details, numbers with CIs and long sentences. Sentences sometimes included the question, definitions of variables, and lists. For example at line 215 we have: "Next, we tested for associations between A(H3N2) evolution and epidemic timing, including onset week, defined as the winter changepoint in incidence [16], and peak week, defined as the first week of maximum incidence; spatiotemporal synchrony, measured as the variation (standard deviation, s.d.) in regional onset and peak timing; and epidemic speed, including seasonal duration and the number of weeks from onset to peak (Table 2, Figure S11)". I would suggest putting the methods section first, using shorter sentences, separating lists from the question being asked, and stating what was found without also putting in all the extra detail. Putting the methods section before the results might reduce the sense that you have to explain what you did and how in the results section too.

Thank you for suggesting how to improve the readability of the Results section. In the revised manuscript, we follow the reviewer’s advice to put the Methods section before the Results section. Although eLife formatting requirements specify the order: Introduction, Results, Discussion, and Methods, the journal allows for the Methods section to follow the Introduction when it makes sense to do so. We agree with the reviewer that putting the Methods section before the Results section makes our results easier to follow because we no longer need to introduce methodological details at the beginning of each set of results.

L285 in the RF you remove variables without significant correlations with the target variables, but isn't one of the aims of RF to uncover relationships where a correlation might not be evident, and in part to reveal combinations of features that give the targeted outcome? Also with the RF, I am a bit concerned that you could not use the leave-one-out approach because it was "unstable" - presumably that means that you obtain quite different results if you leave out a season. How robust are these results, and what are the most sensitive aspects? Are the same variables typically high in importance if you leave out a season, for example? What does the scatterplot of observed vs predicted epidemic size (as in Fig 7) look like if each prediction is for the one that was left out (i.e. from a model trained on all the rest)? In my experience, where the RF is "unstable", that can look pretty terrible even if the model trained on all the data looks great (as does Figure 7). In any case I think it's worth discussing sensitivity.(1) In response to the reviewer’s first question, we explain our rationale for not including all candidate predictors in random forest and penalized regression models.

Models trained with different combinations of predictors can have similar performance, and these combinations of predictors can include variables that do not necessarily have strong univariate associations with the target variable. The performance of random forest and LASSO regression models are not sensitive to redundant or irrelevant predictors (see Figure 10.2 in Kuhn & Johnson, 2019). However, if our goal is variable selection rather than strictly model performance, it is considered best practice to remove collinear, redundant, and/or irrelevant variables prior to training models (see section 11.3 in Kuhn & Johnson, 2019). In both random forest and LASSO regression models, if there are highly collinear variables that are useful for predicting the target variable, the predictor chosen by the model becomes a random selection. In random forest models, these highly collinear variables will be used in all splits across the forest of decision trees, and this redundancy dilutes variable importance scores. Thus, failing to minimize multicollinearity prior to model training could result in some variables having low rankings and the appearance of being unimportant, because their importance scores are overshadowed by those of the highly correlated variables. Our rationale for preprocessing predictor data follows the philosophy of Kuhn & Johnson, 2019, who recommend including the minimum possible set of variables that does not compromise model performance. Even if a particular model is insensitive to extra predictors, Kuhn and John explain that “removing predictors can reduce the cost of acquiring data or improve the throughput of the software used to make predictions.”

In the revised manuscript, we include more details about our steps for preprocessing predictor data. We also follow the reviewer’s suggestion to include all evolutionary predictors in variable selection analyses, regardless of whether they have strong univariate correlations with target outcomes, because the performance of random forest and LASSO regression models is not affected by redundant predictors.

Including additional predictors in our variable selection analyses does not change our conclusions. As reported in our original manuscript, predictors with strong univariate correlations with various epidemic metrics were the highest ranked features in both random forest and LASSO regression models.

Lines 523-563:

“Preprocessing of predictor data: The starting set of candidate predictors included all viral fitness metrics: genetic and antigenic distances between current and previously circulating strains and the standard deviation and Shannon diversity of H3 and N2 LBI values in the current season. To account for potential type or subtype interference, we included A(H1N1) or A(H1N1)pdm09 epidemic size and B epidemic size in the current and prior season and the dominant IAV subtype in the prior season (Lee et al., 2018). We included A(H3N2) epidemic size in the prior season as a proxy for prior natural immunity to A(H3N2). To account for vaccine-induced immunity, we considered four categories of predictors and included estimates for the current and prior seasons: national vaccination coverage among adults (18-49 years coverage × ≥ 65 years coverage), adjusted A(H3N2) vaccine effectiveness (VE), a combined metric of vaccination coverage and A(H3N2) VE (18-49 years coverage × ≥ 65 years coverage × VE), and H3 and N2 epitope distances between naturally circulating A(H3N2) viruses and the U.S. A(H3N2) vaccine strain in each season. We could not include a predictor for vaccination coverage in children or consider cladespecific VE estimates, because these data were not available for most seasons in our study.

Random forest and LASSO regression models are not sensitive to redundant (highly collinear) features (Kuhn & Johnson, 2019), but we chose to downsize the original set of candidate predictors to minimize the impact of multicollinearity on variable importance scores. For both types of models, if there are highly collinear variables that are useful for predicting the target variable, the predictor chosen by the model becomes a random selection (Kuhn & Johnson, 2019). In random forest models, these highly collinear variables will be used in all splits across the forest of decision trees, and this redundancy dilutes variable importance scores (Kuhn & Johnson, 2019). We first confirmed that none of the candidate predictors had zero variance or near-zero variance. Because seasonal lags of each viral fitness metric are highly collinear, we included only one lag of each evolutionary predictor, with a preference for the lag that had the strongest univariate correlations with various epidemic metrics. We checked for multicollinearity among the remaining predictors by examining Spearman’s rank correlation coefficients between all pairs of predictors. If a particular pair of predictors was highly correlated (Spearman’s 𝜌 >0.8), we retained only one predictor from that pair, with a preference for the predictor that had the strongest univariate correlations with various epidemic metrics. Lastly, we performed QR decomposition of the matrix of remaining predictors to determine if the matrix is full rank and identify sets of columns involved in linear dependencies. This step did not eliminate any additional predictors, given that we had already removed pairs of highly collinear variables based on Spearman correlation coefficients.

After these preprocessing steps, our final set of model predictors included 21 variables, including 8 viral evolutionary indicators: H3 epitope distance (t – 2), HI log2 titer distance (t – 2), H3 RBS distance (t – 2), H3 non-epitope distance (t – 2), N2 epitope distance (t – 1), N2 non-epitope distance (t – 1), and H3 and N2 LBI diversity (s.d.) in the current season; 6 proxies for type/subtype interference and prior immunity: A(H1N1) and B epidemic sizes in the current and prior season, A(H3N2) epidemic size in the prior season, and the dominant IAV subtype in the prior season; and 7 proxies for vaccine-induced immunity: A(H3N2) VE in the current and prior season, H3 and N2 epitope distances between circulating strains and the vaccine strain in each season, the combined metric of adult vaccination coverage × VE in the current and prior season, and adult vaccination coverage in the prior season.”

(2) Next, we clarify our model training methodology to address the reviewer’s second point about using a leave-one-out cross-validation approach.

We believe the reviewer is mistaken; we use a leave-one-season-out validation approach which lends some robustness to the predictions. In our original submission, we stated “We created each forest by generating 3,000 regression trees from 10 repeats of a leave-one-season-out (jackknife) cross-validated sample of the data. Due to the small size of our dataset, evaluating the predictive accuracy of random forest models on a quasi-independent test set produced unstable estimates.” (Lines 813-816 in the original manuscript)

To clarify, we use leave-one-season-out cross-validation to train models and measure model performance, wherein each “assessment” set contains one season of data (predicted by the model), and the corresponding “analysis” set (“fold”) contains the remaining seasons. This approach is roughly analogous to splitting data into training and test sets, but all seasons are used at some point in the training of the model (see Section 3.4 in Kuhn & Johnson, 2019). To reduce noise, we generated 10 bootstrap resamples of each fold and averaged the RMSE and R^2^ values of model predictions from resamples.

Although it would be ideal and best practice to measure model performance with an independent test set, our dataset includes only ~20 seasons. We found that predictions of independent test sets of 2-3 seasons had unstable performance, which indicates we do not have sufficient power to measure model performance with a test set this small. Further, we suspect that large antigenic jumps in a small subset of seasons further contribute to variation in prediction accuracy across randomly selected test sets. Our rationale for using cross-validation instead of an independent test set is best described in Section 4.3 of Kuhn and Johnson’s book “Applied Predictive Modeling” (Kuhn & Johnson, 2013):

“When the number of samples is not large, a strong case can be made that a test set should be avoided because every sample may be needed for model building. Additionally, the size of the test set may not have sufficient power or precision to make reasonable judgements. Several researchers (Molinaro 2005; Martin and Hirschberg 1996; Hawkins et al. 2003) show that validation using a single test set can be a poor choice. Hawkins et al. (2003) concisely summarize this point: “holdout samples of tolerable size [...] do not match the cross-validation itself for reliability in assessing model fit and are hard to motivate. “Resampling methods, such as cross-validation, can be used to produce appropriate estimates of model performance using the training set. These are discussed in length in Sect.4.4. Although resampling techniques can be misapplied, such as the example shown in Ambroise and McLachlan (2002), they often produce performance estimates superior to a single test set because they evaluate many alternate versions of the data.”

In our revised manuscript, we provide additional clarification of our methods (Lines 574-590):

“We created each forest by generating 3,000 regression trees. To determine the best performing model for each epidemic metric, we used leave-one-season-out (jackknife) cross-validation to train models and measure model performance, wherein each “assessment” set is one season of data predicted by the model, and the corresponding “analysis” set contains the remaining seasons. This approach is roughly analogous to splitting data into training and test sets, but all seasons are used at some point in the training of each model (Kuhn & Johnson, 2019). Due to the small size of our dataset (~20 seasons), evaluating the predictive accuracy of random forest models on a quasi-independent test set of 2-3 seasons produced unstable estimates. Instead of testing model performance on an independent test set, we generated 10 bootstrap resamples (“repeats”) of each analysis set (“fold”) and averaged the predictions of models trained on resamples (Kuhn & Johnson, 2013, 2019). For each epidemic metric, we report the mean root mean squared error (RMSE) and R^2^ of predictions from the best tuned model. We used permutation importance (N=50 permutations) to estimate the relative importance of each predictor in determining target outcomes. Permutation importance is the decrease in prediction accuracy when a single feature (predictor) is randomly permuted, with larger values indicating more important variables. Because many features were collinear, we used conditional permutation importance to compute feature importance scores, rather than the standard marginal procedure (Altmann et al., 2010; Debeer & Strobl, 2020; Strobl et al., 2008; Strobl et al., 2007).”

(3) In response to the reviewer’s question about the sensitivity of results when one season is left out, we clarify that the variable importance scores in Figure 8 and model predictions in Figure 9 were generated by models tuned using leave-one-season-out cross-validation.

As explained above, in our leave-one-season-out cross-validation approach, each “assessment” set contains one season of data predicted by the model, and the corresponding “analysis” set (“fold”) contains the remaining seasons. We generated predictions of epidemic metrics and variable importance rankings by averaging the model output of 10 bootstrap resamples of each cross-validation fold.

In Lines 791-806, we describe which epidemic metrics have the highest prediction accuracy and report that random forest models tend to underpredict most epidemic metrics in seasons with high antigenic novelty:

“We measured correlations between observed values and model-predicted values at the HHS region level. Among the various epidemic metrics, random forest models produced the most accurate predictions of A(H3N2) subtype dominance (Spearman’s 𝜌 = 0.95, regional range = 0.85 – 0.97), peak incidence (𝜌 = 0.91, regional range = 0.72 – 0.95), and epidemic size (𝜌 = 0.9, regional range = 0.74 – 0.95), while predictions of effective Rt and epidemic intensity were less accurate (𝜌 = 0.81, regional range = 0.65 – 0.91; 𝜌 = 0.78, regional range = 0.63 – 0.92, respectively) (Figure 9). Random forest models tended to underpredict most epidemic targets in seasons with substantial H3 antigenic transitions, in particular the SY97 cluster seasons (1998-1999, 1999-2000) and the FU02 cluster season (2003-2004) (Figure 9).

For epidemic size and peak incidence, seasonal predictive error – the root-mean-square error (RMSE) across all regional predictions in a season – increased with H3 epitope distance (epidemic size, Spearman’s 𝜌 = 0.51, P = 0.02; peak incidence, 𝜌 = 0.63, P = 0.004) and N2 epitope distance (epidemic size, 𝜌 = 0.48, P = 0.04; peak incidence, 𝜌 = 0.48, P = 0.03) (Figure 9 – figure supplements 1 – 2). For models of epidemic intensity, seasonal RMSE increased with N2 epitope distance (𝜌 = 0.64, P = 0.004) but not H3 epitope distance (𝜌 = 0.06, P = 0.8) (Figure 9 – figure supplements 1 – 2). Seasonal RMSE of effective Rt and subtype dominance predictions did not correlate with H3 or N2 epitope distance (Figure 9 – figure supplements 1 – 2).”

I think the competition (interference) results are really interesting, perhaps among the most interesting aspects of this work.

Thank you! We agree that our finding that subtype interference has a greater impact than viral evolution on A(H3N2) epidemics is one of the more interesting results in the study.

Have you seen the paper by Barrat-Charlaix et al? They found that LBI was not good predicting frequency dynamics (see https://pubmed.ncbi.nlm.nih.gov/33749787/); instead, LBI was high for sequences like the consensus sequence, which was near to future strains. LBI also was not positively correlated with epidemic impact in Figure S7.

The local branching index (LBI) measures the rate of recent phylogenetic branching and approximates relative fitness among viral clades, with high LBI values representing greater fitness (Neher et al. 2014).

Two of this study’s co-authors (John Huddleston and Trevor Bedford) are also co-authors of BarratCharlaix et al. 2021. Barrat-Charlaix et al. 2021 assessed the performance of LBI in predicting the frequency dynamics and fixation of individual amino acid substitutions in A(H3N2) viruses. Our study is not focused on predicting the future success of A(H3N2) clades or the frequency dynamics or probability of fixation of individual substitutions. Instead, we use the standard deviation and Shannon diversity of LBI values in each season as a proxy for genealogical (clade-level) diversity. We find that, at a seasonal level, low diversity of H3 or N2 LBI values in the current season correlates with greater epidemic intensity, higher transmission rates, and shorter seasonal duration.

In the Discussion we provide an explanation for these correlation results (Lines 848-857):

“The local branching index (LBI) is traditionally used to predict the success of individual clades, with high LBI values indicating high viral fitness (Huddleston et al., 2020; Neher et al., 2014). In our epidemiological analysis, low diversity of H3 or N2 LBI in the current season correlated with greater epidemic intensity, higher transmission rates, and shorter seasonal duration. These associations suggest that low LBI diversity is indicative of a rapid selective sweep by one successful clade, while high LBI diversity is indicative of multiple co-circulating clades with variable seeding and establishment times over the course of an epidemic. A caveat is that LBI estimation is more sensitive to sequence sub-sampling schemes than strain-level measures. If an epidemic is short and intense (e.g., 1-2 months), a phylogenetic tree with our sub-sampling scheme (50 sequences per month) may not incorporate enough sequences to capture the true diversity of LBI values in that season.”

Figure 1 - LBI goes up over time. Is that partly to do with sampling? Overall how do higher sampling volumes in later years impact this analysis? (though you choose a fixed number of sequences so I guess you downsample to cope with that). I note that LBI is likely to be sensitive to sequencing density.

Thank you for pointing this out. We realized that increasing LBI Shannon diversity over the course of the study period was indeed an artefact of increasing sequence volume over time. Our sequence subsampling scheme involves selecting a random sample of up to 50 viruses per month, with up to 25 viruses selected from North America (if available) and the remaining sequences evenly divided across nine other global regions. In early seasons of the study (late 1990s/early 2000s), sampling was often too sparse to meet the 25 viruses/month threshold for North America or for the other global regions combined (H3: Figure 2 - figure supplement 1; N2: Figure 2 - figure supplement 2). Ecological diversity metrics are sensitive to sample size, which explains why LBI Shannon diversity appeared to steadily increase over time in our original submission. In our revised manuscript, we correct for uneven sample sizes across seasons before estimating Shannon diversity and clarify our methodology.

Lines 443-482:

“Clade growth: The local branching index (LBI) measures the relative fitness of co-circulating clades, with high LBI values indicating recent rapid phylogenetic branching (Huddleston et al., 2020; Neher et al., 2014). To calculate LBI for each H3 and N2 sequence, we applied the LBI heuristic algorithm as originally described by Neher et al., 2014 to H3 and N2 phylogenetic trees, respectively. We set the neighborhood parameter 𝜏 to 0.4 and only considered viruses sampled between the current season 𝑡 and the previous season 𝑡 – 1 as contributing to recent clade growth in the current season 𝑡.

Variation in the phylogenetic branching rates of co-circulating A(H3N2) clades may affect the magnitude, intensity, onset, or duration of seasonal epidemics. For example, we expected that seasons dominated by a single variant with high fitness might have different epidemiological dynamics than seasons with multiple co-circulating clades with varying seeding and establishment times. We measured the diversity of clade growth rates of viruses circulating in each season by measuring the standard deviation (s.d.) and Shannon diversity of LBI values in each season. Given that LBI measures relative fitness among cocirculating clades, we did not compare overall clade growth rates (e.g., mean LBI) across seasons.

Each season’s distribution of LBI values is right-skewed and does not follow a normal distribution. We therefore bootstrapped the LBI values of each season in each replicate dataset 1000 times (1000 samples with replacement) and estimated the seasonal standard deviation of LBI from resamples, rather than directly from observed LBI values. We also tested the seasonal standard deviation of LBI from log transformed LBI values, which produced qualitatively equivalent results to bootstrapped LBI values in downstream analyses.

As an alternative measure of seasonal LBI diversity, we binned raw H3 and N2 LBI values into categories based on their integer values (e.g. an LBI value of 0.5 is assigned to the (0,1] bin) and estimated the exponential of the Shannon entropy (Shannon diversity) of LBI categories (Hill, 1973; Shannon, 1948). The Shannon diversity of LBI considers both the richness and relative abundance of viral clades with different growth rates in each season and is calculated as follows:P1D=exp⁡(−∑i=1Rpiln⁡pi)

where PqD is the effective number of categories or Hill numbers of order 𝑞 (here, clades with different growth rates), with 𝑞 defining the sensitivity of the true diversity to rare versus abundant categories (Hill, 1973). exp is the exponential function, pi is the proportion of LBI values belonging to the 𝑖th category, and 𝑅 is richness (the total number of categories). Shannon diversity P1D (𝑞 = 1) estimates the effective number of categories in an assemblage using the geometric mean of their proportional abundances pi (Hill, 1973).

Because ecological diversity metrics are sensitive to sampling effort, we rarefied H3 and N2 sequence datasets prior to estimating Shannon diversity so that seasons had the same sample size. For each season in each replicate dataset, we constructed rarefaction and extrapolation curves of LBI Shannon diversity and extracted the Shannon diversity estimate of the sample size that was twice the size of the reference sample size (the smallest number of sequences obtained in any season during the study) (iNEXT R package) (Chao et al., 2014). Chao et al. found that their diversity estimators work well for rarefaction and short-range extrapolation when the extrapolated sample size is up to twice the reference sample size. For H3, we estimated seasonal diversity using replicate datasets subsampled to 360 sequences/season; For N2, datasets were subsampled to 230 sequences/season.”

Estimating the Shannon diversity of LBI from datasets with even sampling across seasons removes the previous secular trend of increasing LBI diversity over time (Figure 2 in revised manuscript).

Figure 3 - I wondered what about the co-dominant times?

In Figure 3, orange points correspond to seasons in which A(H3N2) and A(H1N1) were codominant. We are not sure of the reviewer’s specific question concerning codominant seasons, but if it concerns whether antigenic drift is linked to epidemic magnitude among codominant seasons alone, we cannot perform separate regression analyses for these seasons because there are only two codominant seasons during the 22 season study period.

Figure 4 - Related to drift and epidemic size, dominance, etc. -- when is drift measured, and (if it's measured in season t), would larger populations create more drift, simply by having access to more opportunity (via a larger viral population size)? This is a bit 'devil's advocate' but what if some epidemiological/behavioural process causes a larger and/or later peak, and those gave rise to higher drift?

Seasonal drift is measured as the genetic or antigenic distance between viruses circulating during season t and viruses circulating in the prior season (𝑡 – 1) or two seasons ago (𝑡 – 2).

Concerning the question about whether larger human populations lead to greater rates of antigenic drift, phylogeographic studies have repeatedly found that East-South-Southeast Asia are the source populations for A(H3N2) viruses (Bedford et al., 2015; Lemey et al., 2014), in part because these regions have tropical or subtropical climates and larger human populations, which enable year-round circulation and higher background infection rates. Larger viral populations (via larger host population sizes) and uninterrupted transmission may increase the efficiency of selection and the probability of strain survival and global spread (Wen et al., 2016). After A(H3N2) variants emerge in East-South-Southeast Asia and spread to other parts of the world, A(H3N2) viruses circulate via overlapping epidemics rather than local persistence (Bedford et al., 2015; Rambaut et al., 2008). Each season, A(H3N2) outbreaks in the US (and other temperate regions) are seeded by case importations from outside the US, genetic diversity peaks during the winter, and a strong genetic bottleneck typically occurs at the end of the season (Rambaut et al., 2008).

Due to their faster rates of antigenic evolution, A(H3N2) viruses undergo more rapid clade turnover and dissemination than A(H1N1) and B viruses, despite similar global migration networks across A(H3N2), A(H1N1), and B viruses (Bedford et al., 2015). Bedford et al. speculate that there is typically little geographic differentiation in A(H3N2) viruses circulating in each season because A(H3N2) viruses tend to infect adults, and adults are more mobile than children. Compared to A(H3N2) viruses, A(H1N1) and B viruses tend to have greater genealogical diversity, geographic differentiation, and longer local persistence times (Bedford et al., 2015; Rambaut et al., 2008). Thus, some A(H1N1) and B epidemics are reseeded by viruses that have persisted locally since prior epidemics (Bedford et al., 2015).

Theoretical models have shown that epidemiological processes can influence rates of antigenic evolution (Recker et al., 2007; Wen et al., 2016; Zinder et al., 2013), though the impact of flu epidemiology on viral evolution is likely constrained by the virus’s intrinsic mutation rate.

In conclusion, larger host population sizes and flu epidemiology can indeed influence rates of antigenic evolution. However, given that our study is US-centric and focuses on A(H3N2) viruses, these factors are likely not at play in our study, due to intrinsic biological characteristics of A(H3N2) viruses and the geographic location of our study.

We have added a clarifying sentence to the end of the Introduction to narrow the scope of the paper for the reader.

Line 114-116: “Rather than characterize in situ evolution of A(H3N2) lineages circulating in the U.S., we study the epidemiological impacts of antigenic drift once A(H3N2) variants have arrived on U.S. soil and managed to establish and circulate at relatively high levels.”

Methods --L 620 about rescaling and pre- vs post-pandemic times : tell us more - how has reporting changed? could any of this not be because of reporting but because of NPIs or otherwise? Overall there is a lot of rescaling going on. How sensitive are the results to it?it would be unreasonable to ask for a sensitivity analysis for all the results for all the choices around data preparation, but some idea where there is a reason to think there might be a dependence on one of these choices would be great.

In response to the 2009 A(H1N1) pandemic, the US CDC and WHO increased laboratory testing capacity and strengthened epidemiological networks, leading to substantial, long-lasting improvements to influenza surveillance that are still in place today (https://www.cdc.gov/flu/weekly/overview.htm). At the beginning of the COVID-19 pandemic, influenza surveillance networks were quickly adapted to detect and understand the spread of SARS-CoV-2. The 2009 pandemic occurred over a time span of less than one year, and strict non-pharmaceutical interventions (NPIs), such as lockdowns and mask mandates, were not implemented. Thus, we attribute increases in test volume during the post-2009 period to improved virologic surveillance and laboratory testing capacity rather than changes in care-seeking behavior. In the revised manuscript, we include a figure (Figure 1 - figure supplement 2) that shows systematic increases in test volume in all HHS regions after the 2009 pandemic.

Given the substantial increase in influenza test volume after 2009, we opted to keep the time trend adjustment for the pre- and post-2009 pandemic periods and evaluate whether adjusting for regional reporting differences affects our results. When estimating univariate correlations between various

A(H3N2) epidemic metrics and evolutionary indicators, we found qualitatively equivalent results for Spearman correlations and regression models, when adjusting for the pre- and post-2009 pandemic time periods and regional reporting versus only adjusting for the pre-/post-2009 pandemic time periods. Below, we share adjusted versions of Figure 3 (regression results) and Figure 3 - figure supplement 1 (Spearman correlations). Each figure only adjusts for differences in pre- and post-2009 pandemic reporting.

**Author response image 1. sa2fig1:** Adjustment for pre- and post-2009 pandemic only.

**Author response image 2. sa2fig2:** Adjustment for pre- and post-2009 pandemic only.

L635 - Why discretize the continuous LBI distribution and then use Shannon entropy when you could just use the variance and/or higher moments? (or quantiles)? Similarly, why not use the duration of the peak, rather than Shannon entropy? (though there, because presumably data are already binned weekly, and using duration would involve defining start and stop times, it's more natural than with LBI)

We realize that we failed to mention in the methods that we calculated the standard deviation of LBI in each season, in addition to the exponential of the Shannon entropy (Shannon diversity) of LBI. Both the Shannon diversity of LBI values and the standard deviation of LBI values were negatively correlated with effective Rt and epidemic intensity and positively correlated with seasonal duration. The two measures were similarly correlated with effective Rt and epidemic intensity (Figure 3 - figure supplements 2 - 3), while the Shannon diversity of LBI had slightly stronger correlations with seasonal duration than s.d. LBI (Figure 5). Thus, both measures of LBI diversity appear to capture potentially biologically important heterogeneities in clade growth rates.

Separately, we use the inverse Shannon entropy of the incidence distribution to measure the spread of an A(H3N2) epidemic during the season, following the methods of Dalziel et al. 2018. The peak of an epidemic is a single time point at which the maximum incidence occurs. We have not encountered “the duration of the peak” before in epidemiology terminology, and, to our knowledge, there is not a robust way to measure the “duration of a peak,” unless one were to measure the time span between multiple points of maximum incidence or designate an arbitrary threshold for peak incidence that is not strictly the maximum incidence. Given that Shannon entropy is based on the normalized incidence distribution over the course of the entire influenza season (week 40 to week 20), it does not require designating an arbitrary threshold to describe epidemic intensity.

L642 - again why normalize epidemic intensities, and how sensitive are the results to this? I would imagine given that the RF results were unstable under leave-one-out analysis that some of those results could be quite sensitive to choices of normalization and scaling.

Epidemic intensity, defined as the inverse Shannon entropy of the incidence distribution, measures the spread of influenza cases across the weeks in a season. Following Dalziel et al. 2018, we estimated epidemic intensity from normalized incidence distributions rather than raw incidences so that epidemic intensity is invariant under differences in reporting rates and/or attack rates across regions and seasons. If we were to use raw incidences instead, HHS regions or seasons could have the appearance of greater or lower epidemic intensity (i.e., incidence concentrated within a few weeks or spread out over several weeks), due to differences in attack rates or test volume, rather than fundamental differences in the shapes of their epidemic curves. In other words, epidemic intensity is intended to measure the shape and spread of an epidemic, regardless of the actual volume of cases in a given region or season.

In the methods section, we provide further clarification for why epidemic intensities are based on normalized incidence distributions rather than raw incidences.

Lines 206-209: “Epidemic intensity is intended to measure the shape and spread of an epidemic, regardless of the actual volume of cases in a given region or season. Following the methodology of Dalziel et al. 2018, epidemic intensity values were normalized to fall between 0 and 1 so that epidemic intensity is invariant to differences in reporting rates and/or attack rates across regions and seasons.”

L643 - more information about what goes into Epidemia (variables, priors) such that it's replicable/understandable without the code would be good.

We now include additional information concerning the epidemic models used to estimate Rt, including all model equations, variables, and priors (Lines 210-276 in Methods).

L667 did you do breakpoint detection? Why linear models? Was log(incidence) used?

In our original submission, we estimated epidemic onsets using piecewise regression models (Lines 666674 in original manuscript), which model non-linear relationships with breakpoints by iteratively fitting linear models (Muggeo, 2003). Piecewise regression falls under the umbrella of parametric methods for breakpoint detection.

We did not include results from linear models fit to log(incidence) or GLMs with Gaussian error distributions and log links, due to two reasons. First, models fit to log-transformed data require non-zero values as inputs. Although breakpoint detection does not necessarily require weeks of zero incidence leading up to the start of an outbreak, limiting the time period for breakpoint detection to weeks with nonzero incidence (so that we could use log transformed incidence) substantially pushed back previous more biologically plausible estimates of epidemic onset weeks. Second, as an alternative to limiting the dataset to weeks with non-zero incidence, we tried adding a small positive number to weekly incidences so that we could fit models to log transformed incidence for the whole time period spanning epidemic week 40 (the start of the influenza season) to the first week of maximum incidence. Fitting models to log transformed incidences produced unrealistic breakpoint locations, potentially because log transformations (1) linearize data, and (2) stabilize variance by reducing the impact of extreme values. Due to the short time span used for breakpoint detection, log transforming incidence diminishes abrupt changes in incidence at the beginning of outbreaks, making it difficult for models to estimate biologically plausible breakpoint locations. Log transformations of incidence may be more useful when analyzing time series spanning multiple seasons, rather than short time spans with sharp changes in incidence (i.e., the exponential growth phase of a single flu outbreak).

As an alternative to piecewise regression, our revised manuscript also estimates epidemic onsets using a Bayesian ensemble algorithm that accounts for the time series nature of incidence data and allows for complex, non-linear trajectories interspersed with change points (BEAST - a Bayesian estimator of Abrupt change, Seasonal change, and Trend; Zhao et al., 2019). Although a few regional onset time times differed across the two methods, our conclusions did not change concerning correlations between viral fitness and epidemic onset timing.

We have rewritten the methods section for estimating epidemic onsets to clarify our methodology and to include the BEAST method (Lines 292-308):

“We estimated the regional onsets of A(H3N2) virus epidemics by detecting breakpoints in A(H3N2) incidence curves at the beginning of each season. The timing of the breakpoint in incidence represents epidemic establishment (i.e., sustained transmission) rather than the timing of influenza introduction or arrival (Charu et al., 2017). We used two methods to estimate epidemic onsets: (1) piecewise regression, which models non-linear relationships with break points by iteratively fitting linear models to each segment (segmented R package) (Muggeo, 2008; Muggeo, 2003), and (2) a Bayesian ensemble algorithm (BEAST – a Bayesian estimator of Abrupt change, Seasonal change, and Trend) that explicitly accounts for the time series nature of incidence data and allows for complex, non-linear trajectories interspersed with change points (Rbeast R package) (Zhao et al., 2019). For each region in each season, we limited the time period of breakpoint detection to epidemic week 40 to the first week of maximum incidence and did not estimate epidemic onsets for regions with insufficient signal, which we defined as fewer than three weeks of consecutive incidence and/or greater than 30% of weeks with missing data. We successfully estimated A(H3N2) onset timing for most seasons, except for three A(H1N1) dominant seasons: 20002001 (0 regions), 2002-2003 (3 regions), and 2009-2010 (0 regions). Estimates of epidemic onset weeks were similar when using piecewise regression versus the BEAST method, and downstream analyses of correlations between viral fitness indicators and onset timing produced equivalent results. We therefore report results from onsets estimated via piecewise regression.”

L773 national indicators -- presumably this is because you don't have regional-level information, but it might be worth saying that earlier so it doesn't read like there are other indicators now, called national indicators, that we should have heard of

In the revised manuscript, we move a paragraph that was at the beginning of the Results to the beginning of the Methods.

Lines 123-132:

“Our study focuses on the impact of A(H3N2) virus evolution on seasonal epidemics from seasons 1997-1998 to 2018-2019 in the U.S.; whenever possible, we make use of regionally disaggregated indicators and analyses. We start by identifying multiple indicators of influenza evolution each season based on changes in HA and NA. Next, we compile influenza virus subtype-specific incidence time series for U.S. Department of Health and Human Service (HHS) regions and estimate multiple indicators characterizing influenza A(H3N2) epidemic dynamics each season, including epidemic burden, severity, type/subtype dominance, timing, and the age distribution of cases. We then assess univariate relationships between national indicators of evolution and regional epidemic characteristics. Lastly, we use multivariable regression models and random forest models to measure the relative importance of viral evolution, heterosubtypic interference, and prior immunity in predicting regional A(H3N2) epidemic dynamics.”

In Lines 484-487 in the Methods, we now mention that measures of seasonal antigenic and genetic distance are at the national level.

“For each replicate dataset, we estimated national-level genetic and antigenic distances between influenza viruses circulating in consecutive seasons by calculating the mean distance between viruses circulating in the current season 𝑡 and viruses circulating during the prior season (𝑡 – 1 year; one season lag) or two prior seasons ago (𝑡 – 2 years; two season lag).”

L782 Why Beta regression and what is "the resampled dataset" ?

Beta regression is appropriate for models of subtype dominance, epidemic intensity, and age-specific proportions of ILI cases because these data are continuous and restricted to the interval (0, 1) (Ferrari & Cribari-Neto, 2004). “The resampled dataset” refers to the “1000 bootstrap replicates of the original dataset (1000 samples with replacement)” mentioned in Lines 777-778 of the original manuscript.

In the revised manuscript, we include more background information about Beta regression models, and explicitly mention that regression models were fit to 1000 bootstrap replicates of the original dataset.

Lines 503-507:

“For subtype dominance, epidemic intensity, and age-specific proportions of ILI cases, we fit Beta regression models with logit links. Beta regression models are appropriate when the variable of interest is continuous and restricted to the interval (0, 1) (Ferrari & Cribari-Neto, 2004). For each epidemic metric, we fit the best-performing regression model to 1000 bootstrap replicates of the original dataset.”

The github is clear, comprehensive and well-documented, at least at a brief glance.

Thank you! At the time of resubmission, our GitHub repository is updated to incorporate feedback from the reviewers.

References

Altmann, A., Tolosi, L., Sander, O., & Lengauer, T. (2010). Permutation importance: a corrected feature importance measure. Bioinformatics, 26(10), 1340-1347. https://doi.org/10.1093/bioinformatics/btq134

Barrat-Charlaix, P., Huddleston, J., Bedford, T., & Neher, R. A. (2021). Limited Predictability of Amino Acid Substitutions in Seasonal Influenza Viruses. Mol Biol Evol, 38(7), 2767-2777. https://doi.org/10.1093/molbev/msab065

Bedford, T., Riley, S., Barr, I. G., Broor, S., Chadha, M., Cox, N. J., Daniels, R. S., Gunasekaran, C. P., Hurt, A. C., Kelso, A., Klimov, A., Lewis, N. S., Li, X., McCauley, J. W., Odagiri, T., Potdar, V., Rambaut, A., Shu, Y., Skepner, E., . . . Russell, C. A. (2015). Global circulation patterns of seasonal influenza viruses vary with antigenic drift. Nature, 523(7559), 217-220. https://doi.org/10.1038/nature14460

Chao, A., Gotelli, N. J., Hsieh, T. C., Sander, E. L., Ma, K. H., Colwell, R. K., & Ellison, A. M. (2014). Rarefaction and extrapolation with Hill numbers: a framework for sampling and estimation in species diversity studies. Ecological Monographs, 84(1), 45-67. https://doi.org/10.1890/13-0133.1

Charu, V., Zeger, S., Gog, J., Bjornstad, O. N., Kissler, S., Simonsen, L., Grenfell, B. T., & Viboud, C. (2017). Human mobility and the spatial transmission of influenza in the United States. PLOS Comput Biol, 13(2), e1005382. https://doi.org/10.1371/journal.pcbi.1005382

Dalziel, B. D., Kissler, S., Gog, J. R., Viboud, C., Bjornstad, O. N., Metcalf, C. J. E., & Grenfell, B. T.(2018). Urbanization and humidity shape the intensity of influenza epidemics in U.S. cities. Science, 362(6410), 75-79. https://doi.org/10.1126/science.aat6030

Debeer, D., & Strobl, C. (2020). Conditional permutation importance revisited. BMC Bioinformatics, 21(1), 307. https://doi.org/10.1186/s12859-020-03622-2

Dhanasekaran, V., Sullivan, S., Edwards, K. M., Xie, R., Khvorov, A., Valkenburg, S. A., Cowling, B. J., & Barr, I. G. (2022). Human seasonal influenza under COVID-19 and the potential consequences of influenza lineage elimination. Nat Commun, 13(1), 1721. https://doi.org/10.1038/s41467-022-29402-5

Ferrari, S., & Cribari-Neto, F. (2004). Beta Regression for Modelling Rates and Proportions. Journal of Applied Statistics, 31(7), 799-815. https://doi.org/10.1080/0266476042000214501

Garten, R. J., Davis, C. T., Russell, C. A., Shu, B., Lindstrom, S., Balish, A., Sessions, W. M., Xu, X., Skepner, E., Deyde, V., Okomo-Adhiambo, M., Gubareva, L., Barnes, J., Smith, C. B., Emery, S. L., Hillman, M. J., Rivailler, P., Smagala, J., de Graaf, M., . . . Cox, N. J. (2009). Antigenic and genetic characteristics of swine-origin 2009 A(H1N1) influenza viruses circulating in humans. Science, 325(5937), 197-201. https://doi.org/10.1126/science.1176225

Grebe, K. M., Yewdell, J. W., & Bennink, J. R. (2008). Heterosubtypic immunity to influenza A virus: where do we stand? Microbes Infect, 10(9), 1024-1029. https://doi.org/10.1016/j.micinf.2008.07.002

Hill, M. O. (1973). Diversity and Evenness: A Unifying Notation and Its Consequences. Ecology, 54(2), 427-432. https://doi.org/10.2307/1934352

Huddleston, J., Barnes, J. R., Rowe, T., Xu, X., Kondor, R., Wentworth, D. E., Whittaker, L., Ermetal, B., Daniels, R. S., McCauley, J. W., Fujisaki, S., Nakamura, K., Kishida, N., Watanabe, S., Hasegawa, H., Barr, I., Subbarao, K., Barrat-Charlaix, P., Neher, R. A., & Bedford, T. (2020). Integrating genotypes and phenotypes improves long-term forecasts of seasonal influenza A/H3N2 evolution. Elife, 9, e60067. https://doi.org/10.7554/eLife.60067

Kuhn, M., & Johnson, K. (2013). Applied predictive modeling (Vol. 26). Springer.

Kuhn, M., & Johnson, K. (2019). Feature engineering and selection: A practical approach for predictive models. Chapman and Hall/CRC.

Lee, E. C., Arab, A., Goldlust, S. M., Viboud, C., Grenfell, B. T., & Bansal, S. (2018). Deploying digital health data to optimize influenza surveillance at national and local scales. PLoS Comput Biol, 14(3), e1006020. https://doi.org/10.1371/journal.pcbi.1006020

Lemey, P., Rambaut, A., Bedford, T., Faria, N., Bielejec, F., Baele, G., Russell, C. A., Smith, D. J., Pybus, O. G., Brockmann, D., & Suchard, M. A. (2014). Unifying viral genetics and human transportation data to predict the global transmission dynamics of human influenza H3N2. PLOS Pathog, 10(2), e1003932. https://doi.org/10.1371/journal.ppat.1003932

Muggeo, V. (2008). Segmented: An R Package to Fit Regression Models With Broken-Line Relationships. R News, 8, 20-25.

Muggeo, V. M. (2003). Estimating regression models with unknown break-points. Stat Med, 22(19), 30553071. https://doi.org/10.1002/sim.1545

Neher, R. A., Russell, C. A., & Shraiman, B. I. (2014). Predicting evolution from the shape of genealogical trees. Elife, 3, e03568. https://doi.org/10.7554/eLife.03568

Rambaut, A., Pybus, O. G., Nelson, M. I., Viboud, C., Taubenberger, J. K., & Holmes, E. C. (2008). The genomic and epidemiological dynamics of human influenza A virus. Nature, 453(7195), 615-619. https://doi.org/10.1038/nature06945

Recker, M., Pybus, O. G., Nee, S., & Gupta, S. (2007). The generation of influenza outbreaks by a network of host immune responses against a limited set of antigenic types. PNAS, 104(18), 7711-7716. https://doi.org/10.1073/pnas.0702154104

Shannon, C. E. (1948). A mathematical theory of communication. The Bell system technical journal, 27(3), 379-423. https://doi.org/10.1002/j.1538-7305.1948.tb01338.x

Smith, G. J., Vijaykrishna, D., Bahl, J., Lycett, S. J., Worobey, M., Pybus, O. G., Ma, S. K., Cheung, C. L., Raghwani, J., Bhatt, S., Peiris, J. S., Guan, Y., & Rambaut, A. (2009). Origins and evolutionary genomics of the 2009 swine-origin H1N1 influenza A epidemic. Nature, 459(7250), 1122-1125. https://doi.org/10.1038/nature08182

Sridhar, S. (2016). Heterosubtypic T-Cell Immunity to Influenza in Humans: Challenges for Universal T-Cell Influenza Vaccines. Front Immunol, 7, 195. https://doi.org/10.3389/fimmu.2016.00195

Strobl, C., Boulesteix, A. L., Kneib, T., Augustin, T., & Zeileis, A. (2008). Conditional variable importance for random forests. BMC Bioinformatics, 9, 307. https://doi.org/10.1186/1471-2105-9-307

Strobl, C., Boulesteix, A. L., Zeileis, A., & Hothorn, T. (2007). Bias in random forest variable importance measures: illustrations, sources and a solution. BMC Bioinformatics, 8, 25. https://doi.org/10.1186/1471-2105-8-25

Terajima, M., Babon, J. A., Co, M. D., & Ennis, F. A. (2013). Cross-reactive human B cell and T cell epitopes between influenza A and B viruses. Virol J, 10, 244. https://doi.org/10.1186/1743-422x-10-244

Webster, R. G., Bean, W. J., Gorman, O. T., Chambers, T. M., & Kawaoka, Y. (1992). Evolution and ecology of influenza A viruses. Microbiological Reviews, 56(1), 152-179. https://doi.org/10.1128/mr.56.1.152-179.1992

Wen, F., Bedford, T., & Cobey, S. (2016). Explaining the geographical origins of seasonal influenza A(H3N2). Proc Biol Sci, 283(1838). https://doi.org/10.1098/rspb.2016.1312

Yan, L., Neher, R. A., & Shraiman, B. I. (2019). Phylodynamic theory of persistence, extinction and speciation of rapidly adapting pathogens. Elife, 8. https://doi.org/10.7554/eLife.44205

Zhao, K., Wulder, M. A., Hu, T., Bright, R., Wu, Q., Qin, H., Li, Y., Toman, E., Mallick, B., Zhang, X., & Brown, M. (2019). Detecting change-point, trend, and seasonality in satellite time series data to track abrupt changes and nonlinear dynamics: A Bayesian ensemble algorithm. Remote Sensing of Environment, 232, 111181. https://doi.org/10.1016/j.rse.2019.04.034

Zinder, D., Bedford, T., Gupta, S., & Pascual, M. (2013). The Roles of Competition and Mutation in Shaping Antigenic and Genetic Diversity in Influenza. PLOS Pathogens, 9(1). https://doi.org/10.1371/journal.ppat.1003104